# An Evidential Route to Asymptotic Bayes Optimality under Sparsity

Qiaoyu Liang [1]   Haohua Chen [2]   Zihan Zhu [3]   Michael Evans [1]

## Abstract

From a statistical evidence perspective, we establish some asymptotic optimality properties of certain multiple testing rules based on the relative belief ratio (Evans, 2015). Under the two-groups model with an additive 0-1 loss and within a Bayesian decision theoretic asymptotic framework of Bogdan et al. (2011), we show that relative belief multiple testing rules induced by a simple one-group light-tailed normal prior with a single hyperparameter achieve the same asymptotic Bayes risk as the Bayes oracle benchmark. This risk is the minimum achievable in this asymptotic framework. Despite originating from a different starting point, the evidential relative belief approach enjoys oracle properties. The relative belief multiple testing approach is fundamentally different from existing Bayesian multiple testing procedures, virtually all induced by more complex heavy-tailed one-group global-local shrinkage priors using purely posterior-based inferences (Datta & Ghosh, 2013; Ghosh et al., 2016; Bhadra et al., 2017; Ghosh & Chakrabarti, 2017; Qin & Ghosh, 2025). By measuring statistical evidence via both the prior and posterior, the relative belief approach reveals an alternative new inferential paradigm for attaining asymptotic Bayes optimality under sparsity, one that does not rely on developing increasingly elaborate priors.

## 1. Introduction

### 1.1. Background

In a wide range of modern scientific studies, practitioners are routinely confronted with the problem of simultaneously testing a large number of null hypotheses. Over the past several decades, a rich literature on multiple testing procedures has emerged, with methods typically differing in their inferential objectives and error metrics. A particularly influential and theoretically grounded approach to control the number of false rejections is provided notably by the control of the false discovery rate (FDR), defined as the average proportion of errors among the rejections, as formalized by the seminal Benjamini–Hochberg (BH) procedure (Benjamini & Hochberg, 1995).

In recent years, considerable attention has been devoted to understanding the fundamental limits and the optimality properties of multiple testing procedures under sparsity, namely, settings in which the proportion of true alternatives is small (e.g., Donoho & Jin, 2004; Abramovich et al., 2006; Meinshausen & Rice, 2006; Jin & Cai, 2007; Bogdan et al., 2011). Among the literature on multiple testing under sparsity, Bogdan et al. (2011) study the properties of multiple testing rules through the lens of Bayesian decision theory, wherein a decision maker seeks to minimize expected risk. This perspective leads to a notion of optimality different from FDR; instead, Bogdan et al. (2011) investigate Bayesian multiple testing procedures that are optimal with respect to the minimization of Bayes risk.

Specifically, under the two-groups formulation and the notion of an additive 0–1 loss function, Bogdan et al. (2011) introduced the Bayes oracle as the optimal Bayes rule in the context of Bayesian multiple testing. Carvalho et al. (2009) also commented that a carefully chosen two-groups model can be considered a "gold standard" for sparse problems. Bogdan et al. (2011) then further established conditions under which the optimal Bayes risk can be asymptotically achieved under sparsity, by a multiple testing procedure as the number of hypotheses tends to infinity. This property was termed Asymptotic Bayes Optimality under Sparsity (ABOS). The asymptotic framework developed in Bogdan et al. (2011) is an important theoretical breakthrough, providing a principled benchmark for comparing different multiple testing procedures.

In Bayesian multiple testing, two broad classes of priors are commonly employed: point mass spike-and-slab two-groups priors (e.g., Mitchell & Beauchamp, 1988; George & McCulloch, 1993; Johnstone & Silverman, 2004; Ishwaran

[1] Department of Statistical Sciences, University of Toronto, Toronto, ON, Canada [2] Academy of Mathematics and Systems Science, Chinese Academy of Sciences, Beijing, China [3] Department of Biostatistics, Yale School of Public Health, Yale University, New Haven, CT, USA . Correspondence to: Qiaoyu Liang <qiaoyu.liang@mail.utoronto.ca>.

*Proceedings of the 43rd International Conference on Machine Learning*, Seoul, South Korea. PMLR 306, 2026. Copyright 2026 by the author(s).

& Rao, 2005), and continuous global–local shrinkage priors (e.g., Carvalho et al., 2009; 2010; Armagan et al., 2011; Bhadra et al., 2017). While spike-and-slab formulations provide a direct probabilistic representation of sparsity, global-local shrinkage priors are often computationally more appealing in large-scale problems (e.g., Bhattacharya et al., 2016; Makalic & Schmidt, 2016; Johndrow et al., 2020) due to their flexible and favorable scalability structure.

Adopting the asymptotic framework of Bogdan et al. (2011), Datta & Ghosh (2013), Bhadra et al. (2017), and Ghosh et al. (2016) investigated asymptotic risk properties of multiple testing procedures induced by the horseshoe prior, horseshoe+ prior, and a general class of one-group global-local shrinkage priors with heavy tails, respectively, in the two-groups setup. Under an additive 0-1 loss function, they theoretically established the asymptotic optimality by showing that the ratio of the Bayes risk for these multiple testing procedures to that of the Bayes oracle under the two-groups model is within a constant factor asymptotically, where the constant can be close to one (i.e., nearly ABOS). These results imply that the appropriate use of one-group global–local shrinkage priors can closely approximate the optimal two-groups performance in large-scale high-dimensional Bayesian multiple testing problems.

A natural and compelling follow-up question is whether this multiplicative factor can be exactly one. In other words, this is equivalent to asking whether these induced multiple testing procedures can be ABOS exactly. This question was investigated in Ghosh & Chakrabarti (2017), in which they demonstrated that the question of exact asymptotic Bayes optimality under sparsity has an affirmative answer in the case of horseshoe-type priors. Unlike the previous papers dealing with the univariate normal means problems, Qin & Ghosh (2025) first extended the results of Ghosh & Chakrabarti (2017) to the multivariate normal means problem, and showed that a general class of multivariate global-local priors can achieve ABOS.

Compared with one-group global–local shrinkage priors, the one-group Gaussian prior has an even simpler and more analytically tractable structure, being characterized by a single scale parameter and light tails. However, such a prior is often regarded as insufficient for simultaneously identifying signals and noise in Bayesian multiple testing problems. Notice, though, the one-group Gaussian prior is typically employed under a purely posterior-based inference paradigm, where only the posterior distribution is used. The performance of a multiple testing procedure based on the one-group Gaussian prior, utilizing both prior and posterior, rather than relying solely on the posterior, remains unexplored.

A natural, intuitive, and principled way to make use of both the prior and posterior in the inference is through the statistical evidence paradigm, specifically via the relative belief inference (Evans, 2015). This motivates an investigation into the performance of an evidence-based multiple testing procedure using the one-group Gaussian prior in the problem of simultaneously testing for the means of independent normal observations.

Simply put, the principle of statistical evidence is: evidence in favor of a specific value of an unknown parameter occurs when the posterior probability of the value after observing the data exceeds its prior probability before observing the data, evidence against occurs when the posterior probability of the value after observing the data is less than its prior probability before observing the data, and there is no evidence either way when these two probabilities are equal. The major part of what is discussed here depends only on this simple principle. A numerical measure realizing this principle is the relative belief ratio, defined as the ratio of the posterior probability to the prior probability.

Using the evidential relative belief inference with a single-parameter normal prior, we make three main contributions:

1. Under the asymptotic framework of Bogdan et al. (2011), we are the first to achieve exact ABOS in both univariate and multivariate normal means problems using a single-parameter, light-tailed one-group normal prior. This prior has arguably the simplest and most analytical structure compared with all one-group global–local heavy-tailed priors proposed in the existing literature. This is also the first time one shows that a light-tailed prior can be used to attain ABOS exactly.

2. Within the asymptotic framework of Bogdan et al. (2011), we are the first to move beyond the classical posterior-only Bayesian paradigm by conducting inference through careful measurement of statistical evidence via both prior and posterior.

3. By adopting the relative belief multiple testing rule, we can establish the exact ABOS using arguably the simplest and most direct proof techniques, avoiding reliance on intricate posterior concentration inequalities and advanced real analysis tools that are prevalent in the existing global–local priors literature.

### 1.2. Problem Setting

Suppose our data are modeled as $n$ independent observations $\boldsymbol{X} = (X_1, \ldots, X_n)$, with each $X_i \sim N(\mu_i, \sigma_0^2)$ for $i = 1, \ldots, n$ where the unknown parameters $\mu_1, \ldots, \mu_n$ represent the effects under investigation, and $\sigma_0^2$ is commonly assumed to be known in the multiple testing literature (see, e.g., Abramovich et al., 2006; Bogdan et al., 2008; 2011). Typically, $n$ is large so that we have many tests of normal means (see Johnstone, 2019, for a comprehensive treatment of the normal means model). Without loss of generality,

we let $\sigma_0^2 = 1$. To identify the true signals within $\boldsymbol{X}$, we perform simultaneous testing for $i = 1, \ldots, n$:

$$H_{0i} : \mu_i = 0 \quad \text{versus} \quad H_{1i} : \mu_i \neq 0. \qquad (1)$$

For each $i$, $\mu_i$ is assumed to be truly generated by the following mixture distribution

$$\mu_i \overset{\text{i.i.d.}}{\sim} (1 - p)\delta_{\{0\}} + pN(0, \sigma^2), \quad i = 1, \ldots, n, \qquad (2)$$

where $\sigma^2 > 0$ is typically assumed to be large enough to identify the true signals, $\delta_{\{0\}}$ is the degenerate distribution at zero, and $p \in (0, 1)$ is the theoretical proportion of non-nulls in the population.

The marginal distributions of the $X_i$'s are then given by the following two-groups model for $i = 1, \ldots, n$:

$$X_i \overset{\text{i.i.d.}}{\sim} (1 - p)N(0, 1) + pN(0, 1 + \sigma^2). \qquad (3)$$

## 2. Two-Groups Model and the Asymptotic Framework

Recall the two-groups model (3), the two groups being modeled are $N(0, 1)$ and $N(0, 1 + \sigma^2)$. Such a model is particularly natural when one has a small number of potentially large signals among a large pool of noise, and it has become a popular choice in the literature. Notice that we now have only two unknown parameters $p, \sigma^2$ rather than dealing with $n$ unknown parameters $\mu_i, i = 1, \ldots, n$. In sparse settings, where most of the $\mu_i$'s are zero or negligible in magnitude, it is natural and reasonable to assume that $p$ is small and tends to 0 as the number of hypotheses $n$ grows to infinity (i.e., $p \to 0$ as $n \to \infty$).

Empirical Bayes approaches using the two-groups structure have been considered, for example, in Storey (2003), Genovese & Wasserman (2004), and Efron (2008). Fully Bayesian approaches towards multiple testing based on the two-groups model, obtained by placing hyperpriors on the underlying model parameters, are also available in the literature, such as Scott & Berger (2006; 2010).

Now, in the asymptotic framework of Bogdan et al. (2011), they consider the multiple testing problem (1) with the two-groups structure in (2) and (3). A symmetric 0-1 loss function is assumed for each individual test such that the type I error loss and the type II error loss are both equal to 1. We then further assume that the total loss of a multiple testing procedure is the sum of individual losses across all tests. Hence, this approach is based on the notion of an additive loss function, which is often implicit in most of the current formulations. Let $t_{1i}$ and $t_{2i}$ denote the probabilities of type I error and type II error for the $i$-th test, respectively. Under the two-groups model and an additive 0-1 loss function, the Bayes risk of a multiple testing procedure is

$$R = \sum_{i=1}^{n} \left\{ (1 - p)t_{1i} + pt_{2i} \right\}.$$

It was shown in Bogdan et al. (2011) that the multiple testing rule which minimizes the Bayes risk $R$ is the procedure that applies the Bayes classifier. Specifically, for each $i = 1, \ldots, n$, we reject $H_{0i}$ if $\frac{h_1(X_i)}{h_0(X_i)} > \frac{1-p}{p}$, where $h_1$ denotes the density of $X_i$ under $H_{1i}$ (i.e., the density of $N(0, 1 + \sigma^2)$), while $h_0$ denotes that under $H_{0i}$ (i.e., the density of $N(0, 1)$). After some simplifications, we have the Bayes oracle rule: we reject $H_{0i}$ if

$$X_i^2 > c_{\sigma,f}^2 = \frac{1 + \sigma^2}{\sigma^2}(\log(1 + \sigma^2) + 2\log(f)), \qquad (4)$$

where $f = \frac{1-p}{p}$. Thus, we can now define the Bayes oracle as follows.

**Definition 2.1.** The Bayes oracle is the optimal multiple testing rule, which we reject $H_{0i}$ if $X_i^2 > c_{\sigma,f}^2$, where $c_{\sigma,f}^2$ is defined in (4), and this multiple testing rule minimizes the Bayes risk $R$.

*Remark* 2.2. The Bayes oracle makes use of the unknown parameters $\sigma$ and $p$, so it is not achievable in finite samples.

Bogdan et al. (2011) further reparametrized the parameters as $u = \sigma^2$ and $v = uf^2$, then the threshold for the Bayes oracle (4) becomes $c_{\sigma,f}^2 \equiv c_{u,v}^2 = \left(1 + \frac{1}{u}\right)\left(\log v + \log\left(1 + \frac{1}{u}\right)\right)$. Then, the asymptotic framework in Bogdan et al. (2011) is naturally defined as follows where we consider the sequence $u_n = \sigma_n^2$ and $v_n = u_n f_n^2 = \sigma_n^2 \left(\frac{1-p_n}{p_n}\right)^2$.

**Assumption 2.3.** The sequence of parameters $(\sigma_n, p_n)$ satisfies the following three conditions: (1) $p_n \to 0$ as $n \to \infty$. (2) $u_n = \sigma_n^2 \to \infty$ as $n \to \infty$. (3) $\frac{\log v_n}{u_n} \to C \in (0, \infty)$ as $n \to \infty$.

*Remark* 2.4. The asymptotic framework provides a natural way to systematically characterize the asymptotic Bayes risk as the parameter vector $(\sigma_n, p_n)$ defining the Bayes oracle in (4) varies through an infinite sequence indexed by the number of tests $n$ tending to infinity.

Condition (1) in Assumption 2.3 characterizes the regime of "asymptotically vanishing sparsity" (see Donoho & Jin, 2004; Abramovich et al., 2006), implying that the signal vector $\boldsymbol{\mu}$ becomes increasingly sparse as $n \to \infty$. Condition (2) ensures that the true signals are sufficiently strong to be identified asymptotically. Regarding condition (3), Bogdan et al. (2011) provided detailed insights on the threshold $C$. In summary, condition (3) precludes the limiting power of an individual test from collapsing to zero or saturating at one. Under Assumption 2.3, Bogdan et al. (2011) established the asymptotic type I and type II error probabilities for the Bayes oracle as: $t_{1i}^{AsyBO} = o(p_n), t_{2i}^{AsyBO} = (2\Phi(\sqrt{C}) - 1)(1 + o(1))$ where the $o(1)$ terms vanish as $n \to \infty$. Thus, under Assumption 2.3, the corresponding

optimal asymptotic Bayes risk of the Bayes oracle procedure

$$R_{Opt}^{AsyBO} = n((1-p_n)t_{1i}^{AsyBO} + p_n t_{2i}^{AsyBO})$$
$$= np_n(2\Phi(\sqrt{C}) - 1)(1 + o(1)),$$

where $\Phi(\cdot)$ denotes the standard normal cumulative distribution function (cdf).

**Definition 2.5.** Under Assumption 2.3, a multiple testing procedure with asymptotic Bayes risk $R^{Asy}$ is asymptotically Bayes optimal under sparsity (ABOS) if $R^{Asy}/R_{Opt}^{AsyBO} \to 1$ as $n \to \infty$.

# 3. One-Group Shrinkage Prior and Review of Existing Nearly ABOS and ABOS Results

Priors such as the horseshoe prior are often categorized as one-group shrinkage priors, which can be expressed as global-local scale-mixtures of normals, so one-group shrinkage priors are also often called global-local (GL) shrinkage priors. GL priors take the form

$$\mu_i \mid \left(\lambda_i^2, \xi^2\right) \sim N\left(0, \lambda_i^2\xi^2\right), \ \lambda_i^2 \sim \pi_1, \ \xi^2 \sim \pi_2, \quad (5)$$

where GL priors encourage sparsity by placing large probabilities on means near zero, while their tails are heavy enough to accommodate large signals at the same time through appropriately chosen heavy-tailed densities. Here, $\xi$ is the global parameter, and $\lambda$'s are local parameters. More examples of GL priors can be found in Appendix D.

For global-local shrinkage priors (5), if $X_i \mid \mu_i \sim N(\mu_i, 1)$ for $i = 1, \ldots, n$, then

$$\mu_i \mid (X_i, \kappa_i, \xi) \sim N\left((1 - \kappa_i) X_i, 1 - \kappa_i\right),$$

where $\kappa_i = 1/\left(1 + \lambda_i^2\xi^2\right)$ denotes the $i$-th shrinkage coefficient. Thus, for $i = 1, \ldots, n$, we have $\mathbb{E}\left(\mu_i \mid X_i, \xi\right) = (1 - \mathbb{E}\left(\kappa_i \mid X_i, \xi\right)) X_i$.

Relying on sophisticated posterior concentration inequalities and carefully applying real analysis techniques, especially properties about slowly varying functions (see Appendix D), various multiple testing procedures via one-group shrinkage priors can be shown to be nearly ABOS or even ABOS. We now briefly review these results.

Those nearly ABOS and ABOS results are studied under such setting and assumption: We now suppose $X_1, \cdots, X_n$, are i.i.d. observations generated from the two-groups model (3), and we wish to test $H_{0i} : \mu_i = 0$ vs $H_{1i} : \mu_i \neq 0$, for $i = 1, \ldots, n$, simultaneously. Suppose Assumption 2.3 holds (i.e., under the asymptotic framework of Bogdan et al. (2011)). The global parameter $\xi_n$ of GL priors is always treated as a tuning parameter.

Carvalho et al. (2010) modeled the unknown $\mu_i$'s in the normal means model using the horseshoe prior. They observed that the posterior probability of $H_{1i}$ being true (i.e.,

$\mathbb{P}\left(\mu_i \neq 0 \mid X_i\right)$) under the discrete mixture (2) can be reasonably well approximated by $1 - \widehat{\kappa}_i$, where $\widehat{\kappa}_i$ denotes the $i$-th posterior shrinkage coefficient (i.e., posterior mean of $\kappa_i$) based on the horseshoe prior.

Datta & Ghosh (2013) conducted a formal asymptotic analysis for the horseshoe multiple testing procedure. The hierarchical structure for the horseshoe prior is

$$(X_i \mid \mu_i) \sim N(\mu_i, 1), \ (\mu_i \mid \lambda_i, \xi) \sim N(0, \lambda_i^2\xi_n^2),$$
$$\lambda_i \sim C^+(0, 1),$$

independently for $i = 1, \ldots, n$ where $\xi_n$ is treated as the tuning parameter. Equipped with the horseshoe prior, a half-thresholding rule is considered in Datta & Ghosh (2013), which formally is given by

$$\text{reject } H_{0i} \text{ if } 1 - \mathbb{E}\left(\kappa_i \mid X_i, \xi_n\right) > \frac{1}{2}, i = 1, \ldots, n. \quad (6)$$

Then, they showed the horseshoe-based procedure using the half-thresholding rule (6) attains nearly ABOS if the global shrinkage parameter $\xi_n$ in the horseshoe prior is chosen to be of the same asymptotic order as the proportion of signals $p_n$ in the two-groups model (3) (i.e., $\lim_{n \to \infty} \xi_n/p_n \in (0, \infty)$). Specifically, the asymptotic Bayes risk of the horseshoe multiple testing procedure is $R^{AsyHS} = np_n(2\Phi(3\sqrt{C}) - 1)(1 + o(1))$.

As an extension of the horseshoe prior, the horseshoe+ prior (Bhadra et al., 2017) has an additional layer of local shrinkage parameters $\eta$'s. The hierarchy of the horseshoe+ is

$$X_i \mid \mu_i \sim N(\mu_i, 1), \quad \mu_i \mid \lambda_i, \xi_n \sim N\left(0, \lambda_i^2\xi_n^2\right),$$
$$\lambda_i \mid \eta_i \sim C^+(0, \eta_i), \quad \eta_i \sim C^+(0, 1),$$

independently for $i = 1, \ldots, n$. Taking the same half-thresholding rule (6) as the horseshoe-based procedure. Assuming $\lim_{n \to \infty} \xi_n/p_n \in (0, \infty)$, Bhadra et al. (2017) showed that such a horseshoe+ procedure attains nearly ABOS, where the asymptotic Bayes risk of the multiple testing procedure based on the horseshoe+ prior is $R^{AsyHS+} = np_n\left[2\Phi\left(\sqrt{\frac{2}{\zeta_{HS+}(1-\nu_{HS+})}}\sqrt{C}\right) - 1\right](1 + o(1))$, for all $\zeta_{HS+} \in (0, 1)$ and $\nu_{HS+} \in \left(0, 1/\left[\zeta_{HS+}\left(1 + \xi_n^2\right)\right]\right)$.

Ghosh et al. (2016) considered and proved that a general class of one-group shrinkage priors attain nearly ABOS using the half-thresholding rule (6). The chosen class of one-group shrinkage priors in Ghosh et al. (2016) can be represented as

$$\mu_i \mid \left(\lambda_i^2, \xi_n^2\right) \sim N\left(0, \lambda_i^2\xi_n^2\right), \quad (7)$$
$$\lambda_i^2 \sim \pi_1\left(\lambda_i^2\right) = K\left(\lambda_i^2\right)^{-a-1} L\left(\lambda_i^2\right), \quad (8)$$

independently for $i = 1, \ldots, n$. Here, $K > 0$ is a constant of proportionality, and $a$ is a positive real number. The non-constant slowly varying component $L$ in (8) satisfies (i) and

(ii): (i) There exists some constant $0 < M_0 < \infty$ such that $\sup_{t\in(0,\infty)} L(t) \leq M_0$. (ii) $\lim_{t\to\infty} L(t) \in (0, \infty)$.

This chosen class is sufficiently broad to encompass a wide range of one-group priors, including the horseshoe prior, the three parameter beta normal mixtures priors, the generalized double Pareto priors, and many more. In particular, the class of priors with $a = 0.5$ in (7) and (8) is referred to as the horseshoe-type priors.

Let $\pi_1$ satisfy either (I) or (II): (I) $\frac{1}{2} < a < 1$, (II) $a = \frac{1}{2}$ and $L(t)/\sqrt{\log(t)} \to 0$ as $t \to \infty$. Using the half-thresholding rule (6) induced by the general class of one-group priors (7) and (8), and let $\lim_{n\to\infty} \xi_n/p_n \in (0, \infty)$. Ghosh et al. (2016) showed the asymptotic Bayes risk of this procedure, denoted $R^{AsyOG}$, satisfies

$$
np_n\big[2\Phi\big(\sqrt{2a}\sqrt{C}\big) - 1\big]\big(1 + o(1)\big) \leq R^{AsyOG}
$$
$$
\leq np_n\left[2\Phi\left(\sqrt{\frac{2a}{\zeta_{OG}(1-\nu_{OG})}}\sqrt{C}\right) - 1\right]\big(1 + o(1)\big)
$$

for arbitrary fixed $\zeta_{OG} \in (0, \frac{1}{2})$ and $\nu_{OG} \in (0, 1)$. The $o(1)$ term tends to zero as $n \to \infty$, and depends on the choice of $\zeta_{OG} \in (0, \frac{1}{2})$ and $\nu_{OG} \in (0, 1)$.

Now, suppose $\lim_{n\to\infty} \xi_n/p_n^\gamma \in (0, \infty)$, for arbitrary fixed $\gamma \geq 1$. Then, using the half-thresholding rule (6) induced by the general class of one-group priors (7) and (8) with $a \in [0.5, 1)$, Ghosh & Chakrabarti (2017) showed the asymptotic Bayes risk is $R^{AsyOG(GC)} = np_n[2\Phi(\sqrt{2a\gamma}\sqrt{C}) - 1](1 + o(1))$. In particular, for $a = 0.5$ and $\gamma = 1$, $\lim_{n\to\infty} R^{AsyOG(GC)}/R_{Opt}^{AsyBO} = 1$, which attains ABOS exactly.

Inspired by the empirical Bayes construction for $p$ used in the minimax estimation (van der Pas et al., 2014), Ghosh et al. (2016) use $\widehat{\xi}_n = \max\left\{\frac{1}{n}, \frac{1}{d_3 n}\sum_{i=1}^{n} \mathbf{1}\left\{|X_i| > \sqrt{d_4 \log n}\right\}\right\}$ in practice, where $d_3 \geq 1$ and $d_4 \geq 2$ are some predetermined finite positive constants, and this empirical Bayes procedure would reject $H_{0i}$ if $1 - \mathbb{E}\left(\kappa_i \mid X_i, \widehat{\xi}_n\right) > \frac{1}{2}, i = 1, \ldots, n$. In addition to Assumption 2.3, Ghosh et al. (2016) further assumed $p_n \propto n^{-\beta}$ for some unknown $0 < \beta < 1$. They then proved that this empirical Bayes procedure using a general class of one-group shrinkage priors (7) and (8) attains nearly ABOS. Still, under the same assumptions, Ghosh & Chakrabarti (2017) proved empirical Bayes procedure using horseshoe-type priors attains exact ABOS. The horseshoe prior is certainly notable among those global-local priors; we denote the empirical Bayes procedure using the horseshoe prior as EBHS.

## 4. Preliminaries on Relative Belief Inference

The core of relative belief inference is the relative belief ratio, which is intuitive to understand. To deal with the continuous case in this paper, we can define the relative belief ratio as the ratio of the posterior density $\pi(\theta \mid X)$ to the prior density $\pi(\theta)$: $\mathrm{RB}(\theta \mid X) = \pi(\theta \mid X)/\pi(\theta)$, where $\theta$ is the parameter of interest. There are several variants of the relative belief ratio; see Evans (2015). Now, for the hypothesis testing problem $H_0 : \theta = \theta_0$ versus $H_1 : \theta_i \neq \theta_0$, there is evidence in favour of (against, no evidence either way of) $\theta_0$ being true when $\mathrm{RB}(\theta_0 \mid X) > (<, =)1$.

Inference based on the relative belief ratio enjoys invariance under smooth reparameterizations, a property that is particularly desirable in high-dimensional Bayesian inference. Equivalent conclusions can be obtained using transformations such as the logarithm of the relative belief ratio. Relative belief inferences were also proven to be optimally robust to the prior on the parameter of interest under linear contamination and geometric contamination (see Appendix E). Furthermore, the expected value of the logarithm of the relative belief ratio under the posterior coincides with the relative entropy, also called the Kullback-Leibler divergence, between the posterior and prior. From the standpoint of measuring evidence, this is an object of considerable interest in and of itself, and can be considered as a measure of how much evidence the observed data are providing about the unknown parameter value under investigation. This aspect is not directly exploited in the present work; however, it highlights a close connection between the measurement of statistical evidence and the concept of entropy. Notably, many commonly used divergence measures involve the relative belief ratio (Nott et al., 2020). All the aforementioned properties, and many other desirable, even optimal, properties of relative belief inferences can be found in Evans (2015). Relative belief inference has also been applied in a range of real-life applications; see, for example, Muthukumarana & Evans (2015); Al-Labadi et al. (2022).

## 5. ABOS via Relative Belief in the Univariate Normal Means Problem

Existing literature on ABOS relies predominantly on complicated one-group shrinkage priors with tails heavier than a normal prior. Inspired by Einstein's well-known remark that "everything should be made as simple as possible, but not simpler," it is natural to ask whether a multiple testing procedure employing a prior structure that is simpler and more analytically tractable than the global–local shrinkage priors can still attain ABOS. Thus, it is natural to wonder whether it is possible to simply use a simple light-tailed normal prior with minimal hyperparameter(s) and assumptions, to achieve nearly ABOS or even ABOS within the asymptotic framework of Bogdan et al. (2011). The answer

is affirmative when one uses relative belief inference instead of pure posterior inference. Note that all the proofs in this section can be found in the Appendix F.

Recall that $X_i \mid \mu_i \sim N(\mu_i, 1)$ for $i = 1, \ldots, n$. We consider a simple conjugate normal prior on $\mu_i$, denoted as $\mu_i \mid \tau^2 \sim N(0, \tau^2)$, where $\tau$ is the only hyperparameter. Based on these specifications, the relative belief ratio for $\mu_i = 0$ can be calculated as stated in Proposition 5.1.

**Proposition 5.1.** *In the high-dimensional normal means problem, the relative belief ratio for $\mu_i = 0$ is given by*

$$\text{RB}(\mu_i = 0 \mid X_i, \tau^2) = \sqrt{1 + \tau^2} \exp\left(-\frac{X_i^2}{2\left(1 + \frac{1}{\tau^2}\right)}\right).$$

*Remark* 5.2. From an evidential perspective (Evans, 2015), the relative belief multiple testing rule rejects $H_{0i}$ if $\text{RB}(\mu_i = 0 \mid X_i, \tau^2) < 1$, indicating evidence against $H_{0i}$. Thus, the rejection criterion becomes, for $i = 1, \ldots, n$,

$$|X_i| > r(\tau) := \sqrt{\left(1 + \frac{1}{\tau^2}\right)\log(1 + \tau^2)}, \quad (9)$$

We now apply the criterion (9) to the high-dimensional normal means problem. The following proposition establishes the explicit forms of these error probabilities.

**Proposition 5.3.** *The type I and type II error rates for the relative belief approach in this setting are given by*

$$t_{1i}^{RB} = 2 - 2\Phi(r(\tau)), \quad t_{2i}^{RB} = 2\Phi\left(\frac{r(\tau)}{\sqrt{1 + \sigma^2}}\right) - 1.$$

Thus, the Bayes risk of the relative belief procedure is $R^{RB}(\tau) = (n - np)[2 - 2\Phi(r(\tau))] + np\left[2\Phi\left(\frac{r(\tau)}{\sqrt{1+\sigma^2}}\right) - 1\right]$. We now seek $\tau$ that minimizes this risk. As shown in Proposition 5.4 below, minimizing $R^{RB}(\tau)$ yields the unique Bayes risk minimizer $\tau^*$ for the relative belief procedure.

**Proposition 5.4.** *There exists a unique $\tau^* \in (0, \infty)$ that minimizes the Bayes risk $R^{RB}(\tau)$, given by*

$$\tau^* = r^{-1}\left(\sqrt{\frac{1 + \sigma^2}{\sigma^2}\left(2\log f + \log(1 + \sigma^2)\right)}\right), \quad (10)$$

*where $f = \frac{1-p}{p}$ and $r^{-1}$ denotes the inverse of the strictly increasing function $r$ on $\mathbb{R}_+$.*

*Remark* 5.5. The minimizer $\tau^*$ depends on unknown parameters $p$ and $\sigma$, so it cannot be attained in finite sample.

We now analyze the asymptotic order of $\tau^*$ under the Assumption 2.3.

**Lemma 5.6.** *Let $\tau^*$ be the Bayes risk minimizer defined in (10). Under Assumption 2.3, $\tau_n^* \to \infty$ as $n \to \infty$ and satisfies $\lim_{n\to\infty} \frac{\tau_n^*}{p_n^{-1}\sqrt{\log(1/p_n)}} \in (0, \infty)$.*

Finally, Theorem 5.7 establishes that the relative belief procedure attains ABOS exactly, as long as the tuning parameter $\tau_n$ is asymptotically of the order of $p_n^{-1}\sqrt{\log(1/p_n)}$.

**Theorem 5.7.** *Suppose $X_1, \cdots, X_n$ are i.i.d. observations from two-groups model in (3), and we wish to test $H_{0i} : \mu_i = 0$ vs $H_{1i} : \mu_i \neq 0$, for $i = 1, \ldots, n$, simultaneously using the relative belief rule (9) induced by $\mu_i \sim N(0, \tau_n^2)$. Suppose Assumption 2.3 holds. If $\tau_n \to \infty$ such that $\lim_{n\to\infty} \frac{\tau_n}{p_n^{-1}\sqrt{\log(1/p_n)}} \in (0, \infty)$, then the asymptotic type I and type II error probabilities are*

$$t_{1i}^{AsyRB} = o(p_n), \quad t_{2i}^{AsyRB} = (2\Phi(\sqrt{C}) - 1)(1 + o(1)).$$

*Consequently, the asymptotic Bayes risk $R^{AsyRB}$ is*

$$R^{AsyRB} = np_n(2\Phi(\sqrt{C}) - 1)(1 + o(1)).$$

To put the relative belief procedure in practice, we develop a fully data-driven empirical Bayes relative belief approach via the normal prior $N(0, \tau_n^2)$. Specifically, we use data to give an estimator $\widehat{\tau}_n = \min\left\{n\sqrt{\log n}, \frac{d_1 n\sqrt{\log n}}{\sum_{i=1}^n \mathbf{1}(|X_i| > \sqrt{d_2 \log n})}\right\}$, where $d_1 \geq 1$ and $d_2 \geq 2$ are some predetermined finite positive constants. This $\widehat{\tau}_n$ is asymptotically of the desired order of $p_n^{-1}\sqrt{\log(1/p_n)}$. Then, for this empirical Bayes relative belief procedure (denoted as $\text{EBRB}_N$), the rejection criterion is to reject $H_{0i}$ when

$$|X_i| > r(\widehat{\tau}_n) \quad \text{for } i = 1, \ldots, n, \quad (11)$$

Under the same assumptions as in Ghosh et al. (2016) and Ghosh & Chakrabarti (2017), we show the $\text{EBRB}_N$ procedure attains exact ABOS (i.e., Theorem 5.8).

**Theorem 5.8.** *Suppose $X_1, \cdots, X_n$ are i.i.d. observations from two-groups model in (3), and we wish to test $H_{0i} : \mu_i = 0$ vs $H_{1i} : \mu_i \neq 0$, for $i = 1, \ldots, n$, simultaneously using the relative belief rule (11) induced by $\mu_i \sim N(0, \tau_n^2)$. Suppose Assumption 2.3 holds. Further assume $p_n \propto n^{-\beta}$ for some unknown $0 < \beta < 1$. The asymptotic Bayes risk $R_{EB}^{AsyRB}$ satisfies $\lim_{n\to\infty} \frac{R_{EB}^{AsyRB}}{R_{Opt}^{AsyBO}} = 1$.*

*Remark* 5.9. The proof of Theorem 5.8 relies on a general random threshold extension (which can verify whether random threshold rules in the form of $X_i^2 > \widehat{c}_{random}^2$ attain exact ABOS) of an existing general fixed threshold result (i.e., Theorem 3.2 in Bogdan et al. (2011), which can verify whether fixed threshold rules in the form of $X_i^2 > c_{fixed}^2$ achieve exact ABOS).

## 6. ABOS in the Multivariate Normal Means Problem

Compared with previous univariate normal means problems, the key difference is that the analysis now involves

chi-squared random variables rather than normal random variables. Note that all the proofs in this section can be found in the Appendix G.

## 6.1. Multivariate Bayes Oracle and the Asymptotic Framework

We consider a $k$-dimensional ($k \geq 2$) multivariate normal means model. For $i = 1, \ldots, n$, we suppose independent $\boldsymbol{X}_i \sim N_k(\boldsymbol{\mu}_i, \boldsymbol{\Sigma})$, where $\boldsymbol{\mu}_i \in \mathbb{R}^k$ are unknown mean vectors and $\boldsymbol{\Sigma}$ is a known general variance-covariance matrix. We perform the simultaneously testing: $H_{0i} : \boldsymbol{\mu}_i = \boldsymbol{0}$ versus $H_{1i} : \boldsymbol{\mu}_i \neq \boldsymbol{0}$, for $i = 1, \ldots, n$.

For each $i$, $\boldsymbol{\mu}_i$ is assumed to be truly generated by the following mixture distribution $\boldsymbol{\mu}_i \sim (1 - p)\delta_{\{\boldsymbol{0}\}} + p \, N_k(\boldsymbol{0}, g\boldsymbol{\Sigma})$, $i = 1, \ldots, n$, for $p \in (0, 1)$ is the theoretical proportion of non-nulls and some $g > 0$. Marginally, each $\boldsymbol{X}_i$ follows the two-groups mixture

$$\boldsymbol{X}_i \sim (1-p)N_k(\boldsymbol{0}, \boldsymbol{\Sigma}) + p \, N_k(\boldsymbol{0}, (1+g)\boldsymbol{\Sigma}). \quad (12)$$

We now consider a Bayesian multiple testing framework similar to Bogdan et al. (2011), and continue to work with an additive 0–1 loss function. Accordingly, the Bayes risk $R$ of a testing rule is again $R = \sum_{i=1}^{n} [(1-p)t_{1i} + pt_{2i}]$. Analogous to Bogdan et al. (2011), the minimization of the Bayes risk $R$ is achieved by applying the multivariate version of the Bayes oracle.

**Definition 6.1.** The multivariate Bayes oracle rejects $H_{0i}$ if $\boldsymbol{X}_i^T \boldsymbol{\Sigma}^{-1} \boldsymbol{X}_i > c_0^2 = \frac{1+g}{g} \left[ 2 \log \left( \frac{1-p}{p} \right) + k \log(1+g) \right]$.

*Remark* 6.2. The multivariate Bayes oracle depends on the unknown parameters $p$ and $g$, which are not attainable in finite sample.

The asymptotic framework in the multivariate normal means setting (Qin & Ghosh, 2025) is then introduced as follows.

**Assumption 6.3.** We make the following three asymptotic and regularity assumptions: (I) Let $p$ depend on the number of hypotheses $n$, under the assumption that $p_n \rightarrow 0$ as $n \rightarrow \infty$. (II) Let $g$ depend on the number of hypotheses $n$, under the assumption that $g_n \rightarrow \infty$, and $\frac{-2 \log p_n}{g_n} \rightarrow C_0 \in (0, \infty)$, as $n \rightarrow \infty$. (III) $0 < \iota_{\min}(\boldsymbol{\Sigma}) \leq \iota_{\max}(\boldsymbol{\Sigma}) < \infty$, where the minimum and maximum eigenvalues of $\boldsymbol{\Sigma}$ are $\iota_{\min}(\boldsymbol{\Sigma})$ and $\iota_{\max}(\boldsymbol{\Sigma})$ respectively.

Under Assumption 6.3, Qin & Ghosh (2025) showed that the asymptotic type I error probability of the multivariate Bayes oracle $t_{1i}^{AsymBO}$ is $t_{1i}^{AsymBO} = o(p_n)$. The asymptotic type II error probability $t_{2i}^{AsymBO}$ is $t_{2i}^{AsymBO} = F_{\chi_k^2}(C_0)(1 + o(1))$, where $F_{\chi_k^2}$ is the cdf of the chi-squared distribution with $k$ degrees of freedom. The asymptotic Bayes risk $R_{Opt}^{AsymBO}$ is $R_{Opt}^{AsymBO} = np_n F_{\chi_k^2}(C_0)(1+o(1))$. Similar to Bogdan et al. (2011), the ABOS in the multivariate normal means problem is defined as follows.

**Definition 6.4.** Under Assumption 6.3, a multiple testing procedure is called the multivariate asymptotic Bayes optimal under sparsity (ABOS) if its asymptotic Bayes risk $R^{Asym}$ satisfies $R^{Asym}/R_{Opt}^{AsymBO} \rightarrow 1$ as $n \rightarrow \infty$.

## 6.2. Review of Existing ABOS Results in the Multivariate Normal Means Problem

Qin & Ghosh (2025) extended the results of Ghosh & Chakrabarti (2017) to the multivariate case. Qin & Ghosh (2025) considered the following model. For $i = 1, \ldots, n$, we have $\boldsymbol{X}_i | \boldsymbol{\mu}_i \sim N_k(\boldsymbol{\mu}_i, \boldsymbol{\Sigma})$,, and

$$\boldsymbol{\mu}_i | \lambda_i^2, \xi_n \sim N_k(\boldsymbol{0}, \lambda_i^2 \xi_n \boldsymbol{\Sigma}), \quad (13a)$$

$$\pi_1\left(\lambda_i^2\right) \propto \left(\lambda_i^2\right)^{-a-1} L\left(\lambda_i^2\right). \quad (13b)$$

Here, $\boldsymbol{\Sigma}$ is a known positive definite covariance matrix. The global parameter $\xi_n \in (0, 1)$ is a tuning parameter with $\xi_n \rightarrow 0$ as $n \rightarrow \infty$. Again, $a > 0$ and $L$ is a slowly varying function. Note that (13a) and (13b) constitute a general class of multivariate global-local priors.

Together with the Assumption 6.3, an additional regularity assumption (IV) is made on $L$: (IV) $L$ is non-decreasing and $0 < m \leq \lim_{t \rightarrow \infty} L(t) \leq M < \infty$.

Now, let $\kappa_i = (1 + \lambda_i^2 \xi)^{-1}$ be the $i$-th shrinkage coefficient/factor. For each $i = 1, \ldots, n$, the null hypothesis $H_{0i} : \boldsymbol{\mu}_i = \boldsymbol{0}$ is rejected if $\mathbb{E}(\kappa_i | \boldsymbol{X}_i, \xi_n) < \frac{1}{2}$. Qin & Ghosh (2025) show that this half-thresholding rule based on the general class of multivariate global-local priors (i.e., (13a) and (13b)) can achieve ABOS under assumptions (I)–(IV). Specifically, they assume that $\xi_n = p_n^{(1+\zeta_n)/a}$ with $\zeta_n = 1/\log(\log(1/p_n))$ in (13a) and (13b), then the asymptotic Bayes risk of this procedure, denoted as $R^{AsymOG}$ satisfies $\lim_{n \rightarrow \infty} R^{AsymOG}/R_{Opt}^{AsymBO} = 1$.

## 6.3. ABOS via Relative Belief in the Multivariate Normal Means Problem

Naturally, we place a multivariate normal prior on the mean vector $\boldsymbol{\mu}_i$, with a single hyperparameter $\tau > 0$, i.e., $\boldsymbol{\mu}_i \sim N_k(\boldsymbol{0}, \tau\boldsymbol{\Sigma})$. Then, it is straightforward to obtain the expression of the relative belief ratio for $\boldsymbol{\mu}_i = \boldsymbol{0}$ as shown in the following Proposition 6.5.

**Proposition 6.5.** *Under the multivariate normal prior* $\boldsymbol{\mu}_i \sim N_k(\boldsymbol{0}, \tau\boldsymbol{\Sigma})$, *the relative belief ratio for* $\boldsymbol{\mu}_i = \boldsymbol{0}$ *in the multivariate normal means problem is*

$$\text{RB}(\boldsymbol{\mu}_i = \boldsymbol{0} | \boldsymbol{X}_i, \tau) = (1 + \tau)^{\frac{k}{2}} \exp \left( -\frac{\tau \boldsymbol{X}_i^\top \boldsymbol{\Sigma}^{-1} \boldsymbol{X}_i}{2(1+\tau)} \right).$$

**Proposition 6.6.** *The rejection criterion of the relative belief procedure is to reject* $H_{0i}, i = 1, \ldots, n$ *when*

$$\boldsymbol{X}_i^\top \boldsymbol{\Sigma}^{-1} \boldsymbol{X}_i > k \frac{1+\tau}{\tau} \log(1+\tau) = r_k(\tau). \quad (14)$$

Recall $F_{\chi_k^2}$ is the cdf of the $\chi_k^2$ distribution. Following the Proposition 6.6, we immediately have the following proposition on type I and type II error probabilities of the relative belief procedure (i.e., $t_{1i}^{mRB}$ and $t_{2i}^{mRB}$ respectively).

**Proposition 6.7.** *The type I and type II error rates for the relative belief approach in this setting are given by*

$$t_{1i}^{mRB}(\tau) = 1 - F_{\chi_k^2}\big(r_k(\tau)\big), t_{2i}^{mRB}(\tau) = F_{\chi_k^2}\left(\frac{r_k(\tau)}{1+g}\right).$$

Thus, the Bayes risk for the relative belief procedure, denoted as $R^{mRB}(\tau)$, is $R^{mRB}(\tau) = n(1-p)t_{1i}^{mRB}(\tau) + npt_{2i}^{mRB}(\tau)$. The central task now is to find the optimal $\tau^*$, that minimizes this Bayes risk, which is shown in the following Proposition 6.8.

**Proposition 6.8.** *There exists a unique $\tau^* \in (0, \infty)$ that minimizes the Bayes risk $R^{mRB}(\tau)$, given by*

$$\tau^* = r_k^{-1}\left(\frac{1+g}{g}\left[2\log\left(\frac{1-p}{p}\right)+k\log(1+g)\right]\right). \quad (15)$$

*Remark* 6.9. The Bayes risk minimizer $\tau^*$ depends on the unknown parameters $p$ and $g$, so it is not attainable in finite sample.

In the following Proposition 6.10, we show the asymptotic order of $\tau^*$ under Assumption 6.3.

**Lemma 6.10.** *Let $\tau^*$ be the Bayes risk minimizer defined in (15). Under Assumption 6.3, $\tau_n^* \to \infty$ as $n \to \infty$ and satisfies $\lim_{n\to\infty} \frac{\tau_n^*}{p_n^{-2/k}\log(1/p_n)} \in (0, \infty)$.*

In the following Theorem 6.11, we show that the relative belief procedure can ABOS exactly, for any $\tau_n$ as long as the tuning parameter $\tau_n$ is asymptotically of the order of $p_n^{-2/k}\log(1/p_n)$.

**Theorem 6.11.** *Suppose $X_1, \cdots, X_n$, are independent observations from the two-groups model in (12), and we wish to test $H_{0i} : \mu_i = 0$ vs $H_{1i} : \mu_i \neq 0$, for $i = 1, \ldots, n$, simultaneously, using the relative belief rule (14) induced by $\mu_i \sim N_k(0, \tau_n \Sigma)$. Suppose Assumption 6.3 holds. Further assume $\tau_n \to \infty$ as $n \to \infty$ such that $\lim_{n\to\infty} \frac{\tau_n}{p_n^{-2/k}\log(1/p_n)} \in (0, \infty)$, then the corresponding asymptotic type I and type II error probabilities $t_{1i}^{AsymRB}$ and $t_{2i}^{AsymRB}$ are*

$$t_{1i}^{AsymRB} = o(p_n), \ t_{2i}^{AsymRB} = F_{\chi_k^2}(C_0)(1 + o(1)).$$

*The asymptotic Bayes risk $R^{AsymRB}$ of the relative belief multiple testing procedure is given by*

$$R^{AsymRB} = np_nF_{\chi_k^2}(C_0)(1 + o(1)).$$

# 7. Conclusion and Discussion

From an evidential perspective, Theorem 5.7, Theorem 5.8, and Theorem 6.11 establish that the relative belief procedures enjoy the ABOS property in both the univariate normal means setting and the multivariate normal means setting. Instead of the comparably "deep" global-local shrinkage prior class, the relative belief procedures achieve ABOS by only employing a "shallow" one-group light-tailed normal prior with a single hyperparameter. This conjugate normal prior structure used in the relative belief inference is by far the simplest prior structure used to achieve ABOS compared with the existing literature. Thus, prior elicitation (Mikkola et al., 2024) is also considerably simpler for the normal prior compared to global–local shrinkage priors. Additionally, in Appendix B, we also establish the ABOS property of relative belief procedures via a simple uniform prior (see Theorem B.3 and Theorem B.4). Through a comprehensive simulation study (see Appendix C), we find our proposed procedures EBRB$_N$ and EBRB$_U$ confirm our theoretical results. With the blessing of better rate of convergence properties, both EBRB$_N$ and EBRB$_U$ uniformly outperform EBHS. The computations of EBRB$_N$ and EBRB$_U$ are also highly efficient.

Based on the fact that both uniform prior and normal prior can be used with relative belief rules to obtain ABOS in the univariate normal means setting, it appears that suitable priors can have different tail behaviors, which is quite different from the requirement for the global-local approaches to ABOS (which has a mandatory requirement on heavy tails). To attain ABOS, the same order $p_n^{-1}\sqrt{\log(1/p_n)}$ is required for the tuning parameters for the uniform prior and the normal prior, which is quite intriguing. Qualitatively, this seems to suggest that the main requirement for the priors used in the relative belief approach is that those priors need to be able to place their masses around the parameter values having relatively high likelihood, which can largely avoid the prior-data conflict. Here, the prior-data conflict (Evans & Moshonov, 2006) refers to situations where the prior places most of its mass on parameter values that are incompatible with the observed data. Gelman et al. (2017) also pointed out that priors often need to be understood in the context of the likelihood. Note, the concept of prior-data conflict is deeply connected with the important idea of weakly informative priors (see Gelman, 2006; Evans & Jang, 2011)

The relative belief approach measures evidence considering both the role of prior and posterior together in the inference, while previous inferential approaches are all pure posterior inferences in the literature. The relative belief approach is arguably more analytically tractable than the global-local shrinkage priors approach, with respect to the prior structure and multiple testing rule. With different features listed

above, it is self-evident that the relative belief approach provides a totally new path to ABOS. This potentially suggests a new Bayesian reasoning mode: instead of pushing for ever more elaborate prior designs, we may reach statistical optimality by revising the way we approach Bayesian inference itself. It seems worth thinking about whether we just keep going deeper and deeper by constructing horseshoe++, horseshoe+++ priors, and so on, or are willing to shift our inferential mindset a little bit, so that we can achieve our statistical targets by possibly a more streamlined structure.

Now, with a new route to achieve ABOS using the relative belief, there are many interesting follow-up questions one can ask and pursue.

First, one can ask what general class of priors is compatible with the relative belief approach to achieve nearly ABOS or even ABOS. One possible natural extension is the sub-Gaussian type of priors. Given the extensive toolkit of concentration inequalities for sub-Gaussian distributions (e.g., Boucheron et al., 2013; Wainwright, 2019), a thorough investigation along these lines appears highly feasible. Also, a broad symmetric scale family centered at zero (including some symmetric heavy-tailed priors) is also highly probable to be compatible with the relative belief approach, given the theoretical explorations we had. We believe various techniques to deal with heavy-tailed priors over the years would be relevant and helpful.

Second, implementing the relative belief approach in practice requires choosing the tuning parameter, in which this tuning parameter depends on the $p$. Since $p$ is an unknown parameter, it is important to develop an empirical Bayes estimator for the associated hyperparameter $\tau$ or to equip the $\tau$ with a hyperprior under mild assumptions. In this paper, we provide an empirical Bayes estimator in the univariate normal means setting and show its validity. Here, for the multivariate normal means problem, we conjecture this empirical Bayes estimator $\widehat{\tau}_{\mathrm{multi}} = \min\left\{ n^{2/k} \log n, \right.$

$\left. \left( \frac{n}{\sum_{i=1}^{n} \mathbf{1}\left( X_i^{\top} \Sigma^{-1} X_i > 2 \log n \right)} \right)^{2/k} \log n \right\}$ would be asymptotically of the desired order of $p_n^{-2/k} \log(1/p_n)$, and with the relative belief rule in the multivariate normal means case would achieve exact ABOS.

Third, it is always relevant to study a finite sample setting. An extension of Bogdan et al. (2011) is Neuvial & Roquain (2012), which studies the Bayes risk of different multiple testing procedures using finite sample oracle inequalities. We hope to utilize those tools for further finite sample studies. With different priors that can be employed to achieve ABOS, it would be an interesting direction for future research to characterize the rates of convergence of these

ratios of risks and to compare the "efficiency" of various ABOS approaches.

Fourth, the assumption that $\sigma_0^2$ is known is commonly made. We expect that, in most cases, $\sigma_0^2$ can be accurately estimated from the replicates, allowing the majority of our results to remain valid even when $\sigma_0^2$ is unknown. Nevertheless, formal proofs of some key results may require nontrivial modifications. Also, the current framework assumes that statistics for different tests are independent. Although the model and methods proposed here could, in principle, be extended to settings with dependent test statistics, such extensions would require a thorough new study.

Although the normal means model (asymptotically equivalent to nonparametric regression under some regularity conditions, revealed by Brown & Low (1996)) is already rich enough to study fundamental problems, practitioners always want to go to more complex model settings (such as graphical models and deep neural networks). The global-local priors literature has developed a wide range of computational tools for those more complicated model settings. Currently, the relative belief inference literature has not touched very complicated model settings too often. Thus, relative belief methodology can benefit from computation techniques in the global-local priors literature, while relative belief inference may potentially achieve various statistical optimality in a much more straightforward and elegant way. All those would be valuable to explore.

In recent years, there has been growing interest in the robustness of priors and Bayesian models under various forms of misspecification (e.g., Grünwald, 2012; Bissiri et al., 2016; Miller & Dunson, 2019; Knoblauch et al., 2022; Matsubara et al., 2022; Dewaskar et al., 2025). These developments open promising directions for systematically evaluating the robustness of relative belief inference, as well as to further enhance its robustness and broaden its applicability in more general modeling contexts.

The flexible structure of the relative belief ratio seems to make it well suited for extensions beyond static inference, including sequential learning (e.g., Berger et al., 1994; Foster & Stine, 2008; Duran-Martin et al., 2025) and online learning settings (e.g., Shalev-Shwartz, 2012). This direction is also worth pursuing.

## Acknowledgements

We would like to thank Xueying Tang, Zikun Qin, Ziang Fu, Luxi Qiu, and Wenting Lin for helpful discussions. Evans is partially funded by the Natural Sciences and Engineering Research Council of Canada, grant RGPIN-2024-03839.

## Impact Statement

This paper presents work whose goal is to advance the field of Machine Learning. There are many potential societal consequences of our work, none which we feel must be specifically highlighted here.

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

## A. Notations

The notations that will be used throughout the paper are defined as follows: Let $\{A_n\}$ and $\{B_n\}$ be sequences of non-negative reals such that $B_n \neq 0$ for all large $n$, then

1. $A_n \sim B_n$ means $\lim_{n\to\infty} \frac{A_n}{B_n} = 1$,

2. $A_n \propto B_n$ means $A_n = \widetilde{C} B_n$ for all $n \geqslant 1$ and some global constant $\widetilde{C} > 0$,

3. $a_n = O(b_n)$ if $\left|\frac{a_n}{b_n}\right| \leq N$ for all $n$, for some positive real number $N$ independent of $n$,

4. $A_n = o(B_n)$ means $\lim_{n\to\infty} \frac{A_n}{B_n} = 0$. In particular, $A_n = o(1)$ means $\lim_{n\to\infty} A_n = 0$.

## B. ABOS of Relative Belief Procedures under a Uniform Prior

Beyond the normal prior we discussed before, we further consider using a bounded and non-conjugate uniform prior $U[-\tau_U, \tau_U]$ (which carries a lot of historical remarks in objective Bayes literature, see, e.g., Kass & Wasserman (1996) and Berger et al. (2024)) with the relative belief procedure to achieve ABOS. Note $U[-\tau_U, \tau_U]$ are not either exponential-tailed or polynomial-tailed priors. Apparently, this type of uniform prior constitutes a new class in the ABOS literature, which is different from the unbounded light-tailed normal prior we proposed or any of the previous unbounded global-local heavy-tailed priors.

We first derive the relative belief ratio under $\mu_i \sim U[-\tau_U, \tau_U]$ where $\tau_U > 0$, and express the corresponding relative belief rule. We consider the same assumptions as in Ghosh et al. (2016) and Ghosh & Chakrabarti (2017). Then, with the fixed threshold ABOS criterion in Theorem 3.2 of Bogdan et al. (2011) and the random threshold ABOS criterion we develop in Theorem F.2, we can prove that the relative belief procedure induced by the uniform prior achieves exact ABOS in both fixed threshold and random threshold perspectives. Surprisingly, we find the relative belief procedure with the uniform prior can achieve ABOS if the tuning parameter $\tau_{U,n}$ is asymptotically of the order of $p_n^{-1}\sqrt{\log(1/p_n)}$, which is also the order required for the tuning parameter $\tau_n$ in the normal prior to attain ABOS. In the random threshold perspective, we use empirical Bayes to implement our relative belief procedure. We denote this empirical Bayes relative belief procedure as $\mathrm{EBRB_U}$.

### B.1. Relative Belief Rule under a Uniform Prior

We now study the relative belief procedure under a uniform prior. Throughout this section, we take

$$\mu_i \sim U[-\tau_U, \tau_U], \qquad \tau_U > 0.$$

Let $\varphi$ and $\Phi$ denote the standard normal density and distribution function, respectively. The relative belief ratio at $\mu_i = 0$ can be expressed as the ratio of posterior density at $\mu_i = 0$ to prior density at $\mu_i = 0$

$$\mathrm{RB}(\mu_i = 0 \mid X_i, \tau_U) := \frac{\pi(\mu_i = 0 \mid X_i, \tau_U)}{\pi(\mu_i = 0 \mid \tau_U)}.$$

The following Proposition B.1 gives the expression of $\mathrm{RB}(\mu_i = 0 \mid X_i, \tau_U)$ under the uniform prior.

**Proposition B.1.** *Under the uniform prior*

$$\mu_i \sim U[-\tau_U, \tau_U], \qquad \tau_U > 0,$$

*the relative belief ratio for $\mu_i = 0$ in the high-dimensional normal means problem is*

$$\mathrm{RB}(\mu_i = 0 \mid X_i, \tau_U) = \frac{2\tau_U\, \varphi(X_i)}{\Phi(X_i + \tau_U) - \Phi(X_i - \tau_U)}, \tag{16}$$

*where $\varphi$ and $\Phi$ denote the standard normal density and distribution function.*

*Proof.* By the definition of the relative belief ratio,

$$\mathrm{RB}(\mu_i = 0 \mid X_i, \tau_U) = \frac{\pi(\mu_i = 0 \mid X_i, \tau_U)}{\pi(\mu_i = 0 \mid \tau_U)}. \tag{17}$$

Here $\pi(\mu_i = 0 \mid X_i, \tau_U)$ and $\pi(\mu_i = 0 \mid \tau_U)$ denote the posterior and prior densities of $\mu_i$ evaluated at 0, respectively. Since $\mu_i \mid \tau_U \sim U[-\tau_U, \tau_U]$, the prior density of $\mu_i$ evaluated at 0 is

$$\pi(\mu_i = 0 \mid \tau_U) = \frac{1}{2\tau_U}. \tag{18}$$

Next, we compute the posterior density of $\mu_i$ evaluated at 0. By Bayes' formula,

$$\pi(\mu_i = 0 \mid X_i, \tau_U) = \frac{\varphi(X_i)\frac{1}{2\tau_U}}{\int_{-\tau_U}^{\tau_U} \varphi(X_i - \mu)\frac{1}{2\tau_U} \, d\mu} = \frac{\varphi(X_i)}{\Phi(X_i + \tau_U) - \Phi(X_i - \tau_U)}. \tag{19}$$

Combining (17), (18), and (19), we obtain

$$\mathrm{RB}(\mu_i = 0 \mid X_i, \tau_U) = \frac{\frac{\varphi(X_i)}{\Phi(X_i + \tau_U) - \Phi(X_i - \tau_U)}}{\frac{1}{2\tau_U}} = \frac{2\tau_U \varphi(X_i)}{\Phi(X_i + \tau_U) - \Phi(X_i - \tau_U)}. \tag{20}$$

$\square$

The preceding proposition gives the relative belief ratio explicitly. Using $\mathrm{RB}(\mu_i = 0 \mid X_i, \tau_U) < 1$, we next provide the rejection criterion in the following Proposition B.2.

**Proposition B.2.** *For the relative belief procedure with the uniform prior $U[-\tau_U, \tau_U]$, the rejection criterion is to reject $H_{0i}$ for $i = 1, \ldots, n$ when*

$$X_i^2 > \tilde{r}_U = y_{\tau_U}^2, \tag{21}$$

*where $y_{\tau_U} > 0$ is the unique solution to*

$$2\tau_U \varphi(y_{\tau_U}) = \Phi(y_{\tau_U} + \tau_U) - \Phi(y_{\tau_U} - \tau_U). \tag{22}$$

*Proof.* By Proposition B.1,

$$\mathrm{RB}(\mu_i = 0 \mid X_i, \tau_U) = \frac{2\tau_U \, \varphi(X_i)}{\Phi(X_i + \tau_U) - \Phi(X_i - \tau_U)}.$$

Therefore, the rejection criterion takes the form of

$$\mathrm{RB}(\mu_i = 0 \mid X_i, \tau_U) < 1 \quad \Longleftrightarrow \quad \frac{\Phi(X_i + \tau_U) - \Phi(X_i - \tau_U)}{2\tau_U \, \varphi(X_i)} > 1.$$

Define

$$M_{\tau_U}(X_i) = \frac{\Phi(X_i + \tau_U) - \Phi(X_i - \tau_U)}{2\tau_U \, \varphi(X_i)}, \qquad X_i \in \mathbb{R}.$$

Then, we have

$$\mathrm{RB}(\mu_i = 0 \mid X_i, \tau_U) = \frac{1}{M_{\tau_U}(X_i)}.$$

And, the rejection criterion is equivalent to

$$M_{\tau_U}(X_i) > 1.$$

Also, $M_{\tau_U}$ is even in $X_i$. Hence, it is enough to study $M_{\tau_U}$ on $[0, \infty)$. We will show that

$$M_{\tau_U} \text{ is continuous and strictly increasing on } [0, \infty), \tag{23}$$
$$M_{\tau_U}(0) < 1, \tag{24}$$
$$\lim_{X_i \to \infty} M_{\tau_U}(X_i) = \infty. \tag{25}$$

These three properties imply that there exists a unique $y_{\tau_U} > 0$ such that

$$M_{\tau_U}(y_{\tau_U}) = 1.$$

Equivalently,

$$2\tau_U \varphi(y_{\tau_U}) = \Phi(y_{\tau_U} + \tau_U) - \Phi(y_{\tau_U} - \tau_U).$$

Since $M_{\tau_U}$ is even and strictly increasing in $|X_i|$,

$$\mathrm{RB}(\mu_i = 0 \mid X_i, \tau_U) < 1 \quad \Longleftrightarrow \quad M_{\tau_U}(X_i) > 1 \quad \Longleftrightarrow \quad |X_i| > y_{\tau_U}.$$

Let $\widetilde{r}_U = y_{\tau_U}^2$, this is equivalent to

$$\mathrm{RB}(\mu_i = 0 \mid X_i, \tau_U) < 1 \quad \Longleftrightarrow \quad X_i^2 > \widetilde{r}_U.$$

It remains to prove (23)–(25) to close the argument leading to (22).

*Proof of* (23). Using

$$\Phi(X_i + \tau_U) - \Phi(X_i - \tau_U) = \int_{-\tau_U}^{\tau_U} \varphi(X_i - u)\, du$$

and

$$\frac{\varphi(X_i - u)}{\varphi(X_i)} = \exp\left\{X_i u - \frac{u^2}{2}\right\},$$

we obtain, for $X_i \geq 0$,

$$
\begin{aligned}
M_{\tau_U}(X_i) &= \frac{1}{2\tau_U} \int_{-\tau_U}^{\tau_U} \exp\left\{X_i u - \frac{u^2}{2}\right\} du \\
&= \frac{1}{\tau_U} \int_0^{\tau_U} e^{-u^2/2} \cosh(X_i u)\, du.
\end{aligned}
\tag{26}
$$

This representation shows that $M_{\tau_U}$ is continuous on $[0, \infty)$. Moreover, differentiating (26), for every $X_i > 0$,

$$M'_{\tau_U}(X_i) = \frac{1}{\tau_U} \int_0^{\tau_U} u e^{-u^2/2} \sinh(X_i u)\, du > 0.$$

Hence, $M_{\tau_U}$ is strictly increasing on $[0, \infty)$. This proves (23).

*Proof of* (24). Evaluating $M_{\tau_U}$ at $X_i = 0$, we have

$$M_{\tau_U}(0) = \frac{2\Phi(\tau_U) - 1}{2\tau_U\, \varphi(0)} = \frac{\int_{-\tau_U}^{\tau_U} \varphi(u)\, du}{2\tau_U\, \varphi(0)} < 1,$$

because $\varphi(u) < \varphi(0)$ for all $u \neq 0$. This proves (24).

*Proof of* (25). By (26),

$$M_{\tau_U}(X_i) \geq \frac{1}{\tau_U} \int_{\tau_U/2}^{\tau_U} e^{-u^2/2} \cosh(X_i u)\, du.$$

For $u \in [\tau_U/2, \tau_U]$ and $X_i \geq 0$,

$$\cosh(X_i u) \geq \frac{1}{2} e^{X_i u} \geq \frac{1}{2} e^{X_i \tau_U/2}, \qquad e^{-u^2/2} \geq e^{-\tau_U^2/2}.$$

Therefore,

$$M_{\tau_U}(X_i) \geq \frac{1}{\tau_U} \cdot \frac{\tau_U}{2} \cdot e^{-\tau_U^2/2} \cdot \frac{1}{2} e^{X_i \tau_U/2} = \frac{1}{4} \exp\left\{\frac{X_i \tau_U}{2} - \frac{\tau_U^2}{2}\right\} \to \infty$$

as $X_i \to \infty$. This proves (25). $\qquad\square$

## B.2. ABOS of the Fixed Threshold Relative Belief Procedure under a Uniform Prior

Here, we surprisingly find the relative belief procedure with a uniform prior can achieve ABOS if the tuning parameter $\tau_{U,n}$ is asymptotically of the order of $p_n^{-1}\sqrt{\log(1/p_n)}$, which is also the order required for the tuning parameter $\tau_n$ in the normal prior to attain ABOS.

**Theorem B.3.** *Suppose $X_1, \ldots, X_n$ are i.i.d. observations from the two-groups model* (3), *and we wish to test $H_{0i} : \mu_i = 0$ vs $H_{1i} : \mu_i \neq 0$, for $i = 1, \ldots, n$, simultaneously. Suppose Assumption 2.3 holds. Further assume that $\tau_{U,n}$ satisfies*

$$\lim_{n \to \infty} \frac{\tau_{U,n}}{p_n^{-1}\sqrt{\log(1/p_n)}} \in (0, \infty). \tag{27}$$

*Consider the relative belief procedure induced by the uniform prior*

$$\mu_i \sim U[-\tau_{U,n}, \tau_{U,n}],$$

*which rejects $H_{0i}$ whenever*

$$X_i^2 > \widetilde{r}_U(\tau_{U,n}),$$

*where $\widetilde{r}_U(\tau_{U,n})$ is defined in Proposition B.2, this relative belief procedure attains ABOS, i.e., $\frac{R^{AsyRB_U}}{R_{Opt}^{AsyBO}} \to 1$. Equivalently, its asymptotic Bayes risk is*

$$R^{AsyRB_U} = np_n\left(2\Phi(\sqrt{C}) - 1\right)(1 + o(1)).$$

*Proof.* We recall the standard parametrization throughout the proof:

$$u_n = \sigma_n^2, \qquad f_n = \frac{1 - p_n}{p_n}, \qquad v_n = u_n f_n^2.$$

By Proposition B.2, it is clear that the relative belief rule induced by $\mu_i \sim U[-\tau_{U,n}, \tau_{U,n}]$ is a fixed threshold rule. Specifically, it rejects $H_{0i}$ when

$$X_i^2 > \widetilde{r}_U(\tau_{U,n}), \qquad \widetilde{r}_U(\tau_{U,n}) = y_{\tau_{U,n}}^2,$$

where $y_{\tau_{U,n}} > 0$ is the unique solution to

$$2\tau_{U,n}\varphi(y_{\tau_{U,n}}) = \Phi(y_{\tau_{U,n}} + \tau_{U,n}) - \Phi(y_{\tau_{U,n}} - \tau_{U,n}). \tag{28}$$

We compare this threshold with the fixed threshold ABOS criterion in Theorem 3.2 of Bogdan et al. (2011) (see also Theorem F.1). To prove our relative belief rule attaining exact ABOS, it suffices to show that, for $\tau_{U,n}$ satisfying (27), there exists a sequence $z_n$ such that

$$\widetilde{r}_U(\tau_{U,n}) = \log v_n + z_n, \qquad z_n = o(\log v_n), \qquad z_n + 2\log\log v_n \to \infty.$$

*The asymptotic expansion of $\widetilde{r}_U(\tau_{U,n})$.* We first want to prove that

$$\widetilde{r}_U(\tau_{U,n}) = \log\left(\frac{2\tau_{U,n}^2}{\pi}\right) + o(1), \qquad \text{when } \tau_{U,n} \to \infty. \tag{29}$$

Fix $\varepsilon > 0$. Define two quantities around the proposed limit:

$$b_-(\tau_{U,n}) = \log\left(\frac{2\tau_{U,n}^2}{\pi}\right) - \varepsilon, \qquad b_+(\tau_{U,n}) = \log\left(\frac{2\tau_{U,n}^2}{\pi}\right) + \varepsilon.$$

For all sufficiently large $\tau_{U,n}$, both $b_-(\tau_{U,n})$ and $b_+(\tau_{U,n})$ are positive. Also,

$$\sqrt{b_\pm(\tau_{U,n})} = O(\sqrt{\log \tau_{U,n}}).$$

Hence, we have:
$$\tau_{U,n} - \sqrt{b_\pm(\tau_{U,n})} \to \infty.$$

It follows that
$$\Phi\left(\sqrt{b_\pm(\tau_{U,n})} + \tau_{U,n}\right) - \Phi\left(\sqrt{b_\pm(\tau_{U,n})} - \tau_{U,n}\right) \to 1. \tag{30}$$

We now evaluate the lefthand side of (28) at $y = \sqrt{b_-(\tau_{U,n})}$,

$$2\tau_{U,n}\,\varphi\left(\sqrt{b_-(\tau_{U,n})}\right) = 2\tau_{U,n}\frac{1}{\sqrt{2\pi}}\exp\left\{-\frac{1}{2}\left(\log\left(\frac{2\tau_{U,n}^2}{\pi}\right) - \varepsilon\right)\right\} = e^{\varepsilon/2}.$$

Similarly, at $y = \sqrt{b_+(\tau_{U,n})}$, we have
$$2\tau_{U,n}\,\varphi\left(\sqrt{b_+(\tau_{U,n})}\right) = e^{-\varepsilon/2}.$$

Combining the above two identities with (30), we obtain, for all sufficiently large $\tau_{U,n}$,

$$2\tau_{U,n}\,\varphi\left(\sqrt{b_-(\tau_{U,n})}\right) > \Phi\left(\sqrt{b_-(\tau_{U,n})} + \tau_{U,n}\right) - \Phi\left(\sqrt{b_-(\tau_{U,n})} - \tau_{U,n}\right),$$

and

$$2\tau_{U,n}\,\varphi\left(\sqrt{b_+(\tau_{U,n})}\right) < \Phi\left(\sqrt{b_+(\tau_{U,n})} + \tau_{U,n}\right) - \Phi\left(\sqrt{b_+(\tau_{U,n})} - \tau_{U,n}\right).$$

Thus, the equality defining $y_{\tau_{U,n}}$ is crossed between $\sqrt{b_-(\tau_{U,n})}$ and $\sqrt{b_+(\tau_{U,n})}$. By the uniqueness of the crossing point in Proposition B.2,
$$b_-(\tau_{U,n}) \le \widetilde{r}_U(\tau_{U,n}) \le b_+(\tau_{U,n})$$

for all sufficiently large $\tau_{U,n}$. Since $\varepsilon > 0$ is arbitrary, (29) follows.

*Comparison of $\widetilde{r}_U(\tau_{U,n})$ with $\log v_n$.* By (27), there exists a constant $\ell_U \in (0, \infty)$ such that

$$\frac{\tau_{U,n}}{p_n^{-1}\sqrt{\log(1/p_n)}} \to \ell_U.$$

Equivalently,
$$\tau_{U,n} = \ell_U\, p_n^{-1}\sqrt{\log(1/p_n)}(1 + o(1)).$$

Since $p_n \to 0$, the above equation implies $\tau_{U,n} \to \infty$. Therefore, (29) gives

$$\widetilde{r}_U(\tau_{U,n}) = \log\left(\frac{2\tau_{U,n}^2}{\pi}\right) + o(1).$$

Using the expansion of $\tau_{U,n}$, we obtain

$$\tau_{U,n}^2 = \ell_U^2 p_n^{-2}\log(1/p_n)(1 + o(1)).$$

Hence,

$$\widetilde{r}_U(\tau_{U,n}) = \log\left(\frac{2\ell_U^2}{\pi}p_n^{-2}\log(1/p_n)(1 + o(1))\right) + o(1)$$

$$= 2\log\frac{1}{p_n} + \log\log\frac{1}{p_n} + \log\left(\frac{2\ell_U^2}{\pi}\right) + o(1). \tag{31}$$

We next expand $\log(v_n)$. Since

$$\log(f_n) = \log\left(\frac{1}{p_n}\right) + o(1), \qquad \log(v_n) = \log(u_n) + 2\log(f_n),$$

we have

$$\log(v_n) = 2\log\left(\frac{1}{p_n}\right) + \log(u_n) + o(1).$$

By Assumptions 2.3,

$$\frac{\log(v_n)}{u_n} \to C \in (0, \infty),$$

so $\log(u_n) = o(\log(v_n))$. Therefore,

$$\log(v_n) = 2\log\left(\frac{1}{p_n}\right)(1 + o(1)).$$

Using again $\frac{\log(v_n)}{u_n} \to C$, we get

$$u_n = \frac{2}{C}\log\left(\frac{1}{p_n}\right)(1 + o(1)).$$

Hence,

$$\log(u_n) = \log\log\left(\frac{1}{p_n}\right) + O(1).$$

Substituting this into the expansion of $\log(v_n)$, we obtain

$$\log(v_n) = 2\log\left(\frac{1}{p_n}\right) + \log\log\left(\frac{1}{p_n}\right) + O(1). \tag{32}$$

Combining (31) and (32), we obtain

$$\widetilde{r}_U(\tau_{U,n}) = \log v_n + z_n, \qquad z_n = O(1).$$

Since $\log v_n \to \infty$, this implies

$$z_n = o(\log v_n), \qquad z_n + 2\log\log v_n \to \infty.$$

By Theorem 3.2 of Bogdan et al. (2011), the relative belief rule induced by $\mu_i \sim U[-\tau_{U,n}, \tau_{U,n}]$ attains exact ABOS. Equivalently,

$$R^{AsyRB_U} = np_n\left(2\Phi(\sqrt{C}) - 1\right)(1 + o(1)).$$

$\square$

## B.3. ABOS of the Empirical Bayes Relative Belief Procedure under a Uniform Prior

The preceding ABOS result relies on a tuning parameter $\tau_{U,n}$ whose choice depends on the unknown parameter $p$, and as a result, it cannot be directly applied in practice. We then turn to an empirical Bayes relative belief rule with a data-dependent estimate. From Theorem B.3, we know the tuning parameter $\tau_{U,n}$ in the uniform prior needs to have the same asymptotic order (i.e., $p_n^{-1}\sqrt{\log(1/p_n)}$) as the tuning parameter $\tau_n$ in the normal prior to achieve ABOS. Thus, we consider using the same empirical Bayes estimator $\widehat{\tau}_n(\boldsymbol{X})$ in the normal prior case. Specifically, we recall that $\widehat{\tau}_n(\boldsymbol{X})$ is

$$\widehat{\tau}_n(\boldsymbol{X}) = \min\left\{n\sqrt{\log n}, \ \frac{d_1 n\sqrt{\log n}}{\sum_{i=1}^n \mathbf{1}\left(|X_i| > \sqrt{d_2\log n}\right)}\right\}, \tag{33}$$

where $d_1 \geq 1$ and $d_2 \geq 2$ are some predetermined finite positive constants

For the empirical Bayes relative belief procedure induced by the uniform prior, $\tau_{U,n}$ is replaced by $\widehat{\tau}_n(\boldsymbol{X})$ in the threshold $\widetilde{r}_U(\tau_{U,n})$. Formally, this empirical Bayes relative belief procedure rejects $H_{0i}$ whenever

$$X_i^2 > \widetilde{r}_U(\widehat{\tau}_n(\boldsymbol{X})), \qquad i = 1, \ldots, n,$$

where $\widetilde{r}_U(\widehat{\tau}_n(\boldsymbol{X}))$ denotes the threshold in Proposition B.2 with $\tau_{U,n}$ replaced by $\widehat{\tau}_n(\boldsymbol{X})$.

The next theorem establishes that, under the same assumptions as in Ghosh et al. (2016) and Ghosh & Chakrabarti (2017), our empirical Bayes relative belief procedure induced by the uniform prior, denoted by $\text{EBRB}_U$, attains exact ABOS.

**Theorem B.4.** *Suppose $X_1, \ldots, X_n$ are i.i.d. observations from the two-groups model* (3), *and we wish to test $H_{0i} : \mu_i = 0$ vs $H_{1i} : \mu_i \neq 0$, for $i = 1, \ldots, n$, simultaneously. Suppose Assumption 2.3 holds. Further assume that, for some unknown $0 < \beta < 1$,*

$$p_n \propto n^{-\beta}.$$

*Let $\widehat{\tau}_n(\boldsymbol{X})$ be defined as in* (33) *and $\widetilde{r}_U(\widehat{\tau}_n(\boldsymbol{X}))$ denote the threshold in Proposition B.2 with $\tau_{U,n}$ replaced by $\widehat{\tau}_n(\boldsymbol{X})$. Then, the empirical Bayes relative belief multiple testing procedure rejects $H_{0i}$ whenever*

$$X_i^2 > \widetilde{r}_U(\widehat{\tau}_n(\boldsymbol{X})), \qquad i = 1, \ldots, n,$$

*attains exact ABOS. Equivalently,*

$$\frac{R_{EB}^{AsyRB_U}}{R_{Opt}^{AsyBO}} \to 1.$$

*Proof.* We apply Theorem F.2 with the random threshold

$$\widehat{c}_n^2 = \widetilde{r}_U(\widehat{\tau}_n).$$

Define

$$q_U = 2 \left\{ 1 - \Phi \left( \sqrt{\frac{d_2 C}{2\beta}} \right) \right\}, \tag{34}$$

and then define the threshold

$$c_U^2 = 2 \log \left( \frac{1}{p_n} \right) + \log \log(n) + \log \left( \frac{2 d_1^2}{\pi q_U^2} \right). \tag{35}$$

We want to prove the following two statements:

(i) For every fixed $\varepsilon > 0$,

$$\mathbb{P} \left( \left| \widetilde{r}_U(\widehat{\tau}_n) - c_U^2 \right| > \varepsilon \right) = o(p_n). \tag{36}$$

(ii) The threshold $c_U^2$ is a fixed threshold of a fixed threshold procedure that achieves ABOS. Equivalently, there exists a sequence $z_n$ such that

$$c_U^2 = \log(v_n) + z_n, \qquad z_n = o(\log(v_n)), \qquad z_n + 2 \log \log(v_n) \to \infty.$$

Combining (i) and (ii), Theorem F.2 would imply that the empirical Bayes relative belief rule attains ABOS.

*Proof of (i).* By the assumption $p_n \propto n^{-\beta}$ in Theorem B.4, there exists a constant $\zeta_U \in (0, \infty)$ such that

$$p_n = \zeta_U n^{-\beta}.$$

Hence,

$$f_n = \frac{1 - p_n}{p_n} = \frac{1}{p_n}(1 + o(1)), \qquad \log(f_n) = \beta \log(n) - \log(\zeta_U) + o(1).$$

Since $v_n = u_n f_n^2$, we have

$$\log(v_n) = \log(u_n) + 2\beta \log(n) + O(1).$$

Also, Assumptions 2.3 imply

$$\frac{\log(v_n)}{u_n} \to C \in (0, \infty).$$

Thus $\log(u_n) = o(\log(v_n))$, and therefore,

$$\log(v_n) = 2\beta \log(n)(1 + o(1)).$$

Using again $\frac{\log(v_n)}{u_n} \to C$, we obtain

$$u_n = \frac{2\beta}{C} \log(n)(1 + o(1)). \tag{37}$$

Define

$$N_U = \sum_{i=1}^{n} \mathbf{1}\left(|X_i| > \sqrt{d_2 \log(n)}\right). \tag{38}$$

Then,

$$\widehat{\tau}_n = \min\left\{n\sqrt{\log(n)}, \frac{d_1 n \sqrt{\log(n)}}{N_U}\right\}.$$

We compute the order of $\mathbb{E}(N_U)$. Since

$$X_i \sim (1 - p_n)N(0, 1) + p_n N(0, 1 + u_n),$$

we have

$$\mathbb{E}(N_U) = \underbrace{n(1 - p_n) \, 2\left\{1 - \Phi\left(\sqrt{d_2 \log(n)}\right)\right\}}_{\text{term I}} + \underbrace{np_n \, 2\left\{1 - \Phi\left(\frac{\sqrt{d_2 \log(n)}}{\sqrt{1 + u_n}}\right)\right\}}_{\text{term II}}. \tag{39}$$

We treat the two terms in (39) separately. For the term I, Mills' ratio gives

$$2\left\{1 - \Phi\left(\sqrt{d_2 \log(n)}\right)\right\} = O\left(\frac{n^{-d_2/2}}{\sqrt{\log(n)}}\right).$$

Therefore,

$$\frac{n(1 - p_n) \, 2\left\{1 - \Phi\left(\sqrt{d_2 \log(n)}\right)\right\}}{np_n} = O\left(\frac{n^{\beta - d_2/2}}{\sqrt{\log(n)}}\right) \to 0,$$

because $d_2 \geq 2$ and $0 < \beta < 1$.

For the term II, (37) gives

$$\frac{d_2 \log(n)}{1 + u_n} \to \frac{d_2 C}{2\beta}.$$

Hence,

$$2\left\{1 - \Phi\left(\frac{\sqrt{d_2 \log(n)}}{\sqrt{1 + u_n}}\right)\right\} \to q_U.$$

Thus, the term II satisfies

$$np_n \, 2\left\{1 - \Phi\left(\frac{\sqrt{d_2 \log(n)}}{\sqrt{1 + u_n}}\right)\right\} = q_U np_n(1 + o(1)).$$

Combining the term I and term II, we obtain

$$\mathbb{E}(N_U) = q_U np_n(1 + o(1)). \tag{40}$$

Define the event

$$G_U = \left\{|N_U - \mathbb{E}(N_U)| \leq (\log(n))^{-1/4}\mathbb{E}(N_U)\right\}. \tag{41}$$

Since $N_U$ is a sum of independent Bernoulli random variables, Chernoff bound (see Corollary 2.3.4 in Vershynin (2026)) gives

$$\mathbb{P}(G_U^c) \leq 2\exp\left\{-\frac{(\log(n))^{-1/2}}{3}\mathbb{E}(N_U)\right\}. \tag{42}$$

It remains to check that the right-hand side of (42) is $o(p_n)$. By $p_n \propto n^{-\beta}$ in Theorem B.4, there exists a constant $\zeta_U \in (0, \infty)$ such that

$$p_n = \zeta_U n^{-\beta}.$$

Hence,

$$\log \frac{1}{p_n} = \beta \log n - \log \zeta_U \sim \beta \log n.$$

Combining this with

$$(\log n)^{-1/2} \mathbb{E}(N_U) \sim q_U \zeta_U \frac{n^{1-\beta}}{\sqrt{\log n}},$$

we obtain

$$\frac{(\log n)^{-1/2} \mathbb{E}(N_U)}{\log(1/p_n)} \sim \frac{q_U \zeta_U \, n^{1-\beta}/\sqrt{\log n}}{\beta \log n} = \frac{q_U \zeta_U}{\beta} \frac{n^{1-\beta}}{(\log n)^{3/2}} \to \infty.$$

Therefore,

$$\frac{1}{3} (\log n)^{-1/2} \mathbb{E}(N_U) - \log \frac{1}{p_n} \to \infty.$$

Equivalently,

$$\exp\left( -\frac{1}{3} (\log n)^{-1/2} \mathbb{E}(N_U) \right) = o(p_n).$$

Consequently,

$$\mathbb{P}\left( |N_U - \mathbb{E}(N_U)| > (\log n)^{-1/4} \mathbb{E}(N_U) \right) = o(p_n). \tag{43}$$

On $G_U$, (40) implies

$$N_U = q_U n p_n (1 + o(1)). \tag{44}$$

Since $q_U > 0$ and $np_n \to \infty$, the second term in the definition of $\widehat{\tau}_n$ is eventually smaller than $n\sqrt{\log(n)}$ on $G_U$. Therefore,

$$\widehat{\tau}_n = \frac{d_1 n \sqrt{\log(n)}}{N_U} = \frac{d_1}{q_U} p_n^{-1} \sqrt{\log(n)}(1 + o(1)) \qquad \text{on } G_U. \tag{45}$$

In particular, $\widehat{\tau}_n \to \infty$ on $G_U$.

By (29), applied with $\tau_U = \widehat{\tau}_n$, we have

$$\widetilde{r}_U(\widehat{\tau}_n) = \log\left( \frac{2\widehat{\tau}_n^2}{\pi} \right) + o(1) \qquad \text{on } G_U.$$

Using (45), we obtain

$$\begin{aligned}
\widetilde{r}_U(\widehat{\tau}_n) &= \log\left( \frac{2d_1^2}{\pi q_U^2} p_n^{-2} \log(n)(1 + o(1)) \right) + o(1) \\
&= 2\log\left( \frac{1}{p_n} \right) + \log\log(n) + \log\left( \frac{2d_1^2}{\pi q_U^2} \right) + o(1) \\
&= c_U^2 + o(1) \qquad \text{on } G_U.
\end{aligned} \tag{46}$$

Thus, for every fixed $\varepsilon > 0$ and all sufficiently large $n$,

$$\left\{ \left| \widetilde{r}_U(\widehat{\tau}_n) - c_U^2 \right| > \varepsilon \right\} \subseteq G_U^c.$$

Together with (43), this proves

$$\mathbb{P}\left( \left| \widetilde{r}_U(\widehat{\tau}_n) - c_U^2 \right| > \varepsilon \right) = o(p_n). \tag{47}$$

*Proof of (ii).* We compare $c_U^2$ with $\log(v_n)$. By $p_n \propto n^{-\beta}$, there exists a constant $\zeta_U \in (0, \infty)$ such that

$$p_n = \zeta_U n^{-\beta}.$$

Hence,

$$f_n = \frac{1}{p_n}(1 + o(1)), \qquad \log(f_n) = \log\left( \frac{1}{p_n} \right) + o(1).$$

From (37), we have

$$\log(u_n) = \log\log(n) + \log\left(\frac{2\beta}{C}\right) + o(1).$$

Therefore,

$$\log(v_n) = \log(u_n) + 2\log(f_n) = 2\log\left(\frac{1}{p_n}\right) + \log\log(n) + \log\left(\frac{2\beta}{C}\right) + o(1). \tag{48}$$

Combining (35) and (48), we obtain

$$c_U^2 = \log(v_n) + z_U,$$

where

$$z_U = \log\left(\frac{2d_1^2}{\pi q_U^2}\right) - \log\left(\frac{2\beta}{C}\right) + o(1).$$

Thus, $z_U = O(1)$. Since $\log(v_n) \to \infty$, we have

$$z_U = o(\log(v_n)), \qquad z_U + 2\log\log(v_n) \to \infty.$$

Hence, $c_U^2$ is an ABOS fixed threshold. $\qquad\square$

## C. Simulation Study

With Bayes oracle (i.e., BO) as reference (which uses true underlying parameters $p$ and $\sigma$), we conduct an extensive simulation study to compare the finite sample performance of $\mathrm{EBRB_N}$, $\mathrm{EBRB_U}$, and EBHS. We adopt the experimental framework in Datta & Ghosh (2013) and Ghosh et al. (2016) as a basis, but greatly extend it to a more comprehensive study. We can compare the simulation averages of the proportion of misclassified hypotheses (as estimates of the misclassification probabilities) of $\mathrm{EBRB_N}$, $\mathrm{EBRB_U}$, and EBHS with that of the Bayes oracle in the context of the two-groups problem, in order to assess how closely these multiple testing procedures actually perform relative to the Bayes oracle. Note that the misclassification probability (MP) is equivalent to the individual Bayes risk $(1 - p)t_{1i} + pt_{2i}$ (see Bogdan et al., 2008). Here, the Bayes oracle establishes the lower bound to the MP, whereas the line MP = p represents the case in which all null hypotheses are rejected irrespective of the data. Recall that $\widehat{\xi}_n = \max\left\{\frac{1}{n}, \frac{1}{d_3 n}\sum_{i=1}^{n}\mathbf{1}\left\{|X_i| > \sqrt{d_4\log n}\right\}\right\}$ where $d_3 \geq 1$ and $d_4 \geq 2$ are some predetermined finite positive constants. In both van der Pas et al. (2014) and Ghosh et al. (2016), they used $d_3 = 1$ and $d_4 = 2$ as default choices in their simulation study. We adopt their choices for the implementation of EBHS (see details in Section 3) here. EBHS is implemented using the well-recognized *horseshoe* package (see van der Pas et al., 2019). Recall that $\widehat{\tau}_n = \min\left\{n\sqrt{\log n}, \frac{d_1 n\sqrt{\log n}}{\sum_{i=1}^{n}\mathbf{1}\left(|X_i| > \sqrt{d_2\log n}\right)}\right\}$, where $d_1 \geq 1$ and $d_2 \geq 2$ are some predetermined finite positive constants. Analogously, we choose $d_1 = 1$ and $d_2 = 2$ as default choices for the implementation of $\mathrm{EBRB_N}$ (see details in Section 5) and $\mathrm{EBRB_U}$ (see details in Section B.3).

The simulated data are generated as follows. We always fix $\sigma_0^2 = 1$. For each chosen fixed $p \in (0, 1)$, we consider generating $n = \{200, 1000, 10000\}$ independent observations $X_1, \ldots, X_n$ from the two groups model $(1 - p)N(0, 1) + pN(0, 1 + \sigma^2)$ where four distinct values of $\sigma^2$, specifically 3, 5, 7, and 9 are considered. To estimate the misclassification probability, the process is repeated 1000 times, and simulation averages of misclassification proportions across these replications are used as estimates of the misclassification probabilities for each multiple testing procedure under consideration.

Figures 1–3 present the estimated misclassification probabilities corresponding to the $\mathrm{EBRB_N}$, $\mathrm{EBRB_U}$, and the EBHS multiple testing procedures, alongside those of the Bayes oracle against values of $p$ in the set $\{0.01, 0.05, 0.1, 0.15, 0.2, 0.25, 0.3, 0.35, 0.4, 0.45, 0.5\}$. It is evident from Figures 1–3 that, when $p$ is small, the estimated misclassification probabilities of both $\mathrm{EBRB_N}$ and $\mathrm{EBRB_U}$ are nearly identical to that of the Bayes oracle, which is consistent with the theoretical results. Also, as shown in the figures, it is clear that both $\mathrm{EBRB_N}$ and $\mathrm{EBRB_U}$ uniformly outperform the EBHS approach in every case (often by a large margin) in terms of achieving lower misclassification probabilities across all combinations of $n$'s, $\sigma^2$'s, and $p$'s. Moreover, for any choice of $n$, as the $\sigma^2$ increases from 3 to 9, the performances of the $\mathrm{EBRB_N}$ and $\mathrm{EBRB_U}$ progressively approach that of the Bayes oracle. Notably, when the $\sigma^2$ gets larger, both $\mathrm{EBRB_N}$ and $\mathrm{EBRB_U}$ nearly match the Bayes oracle's performance exactly in a wide range of $p$'s. The $\mathrm{EBRB_U}$ always does slightly better than $\mathrm{EBRB_N}$, which is as expected by the rate of convergence effects. The computations of $\mathrm{EBRB_N}$ and $\mathrm{EBRB_U}$ are highly efficient, with $\mathrm{EBRB_N}$ being fully analytical.

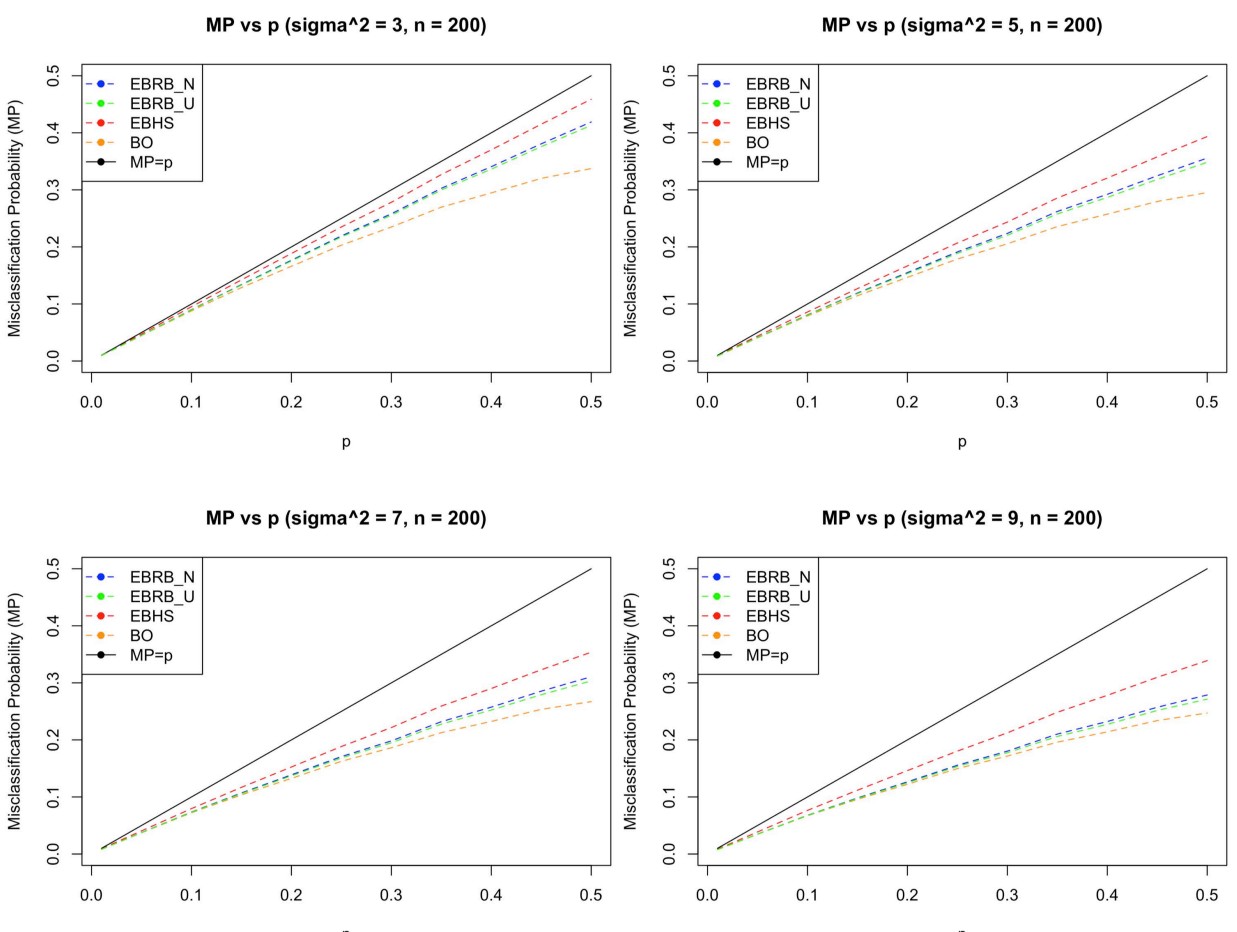

*Figure 1.* Estimated misclassification probabilities for $\mathrm{EBRB_N}$, $\mathrm{EBRB_U}$, EBHS, and BO ($n = 200$).

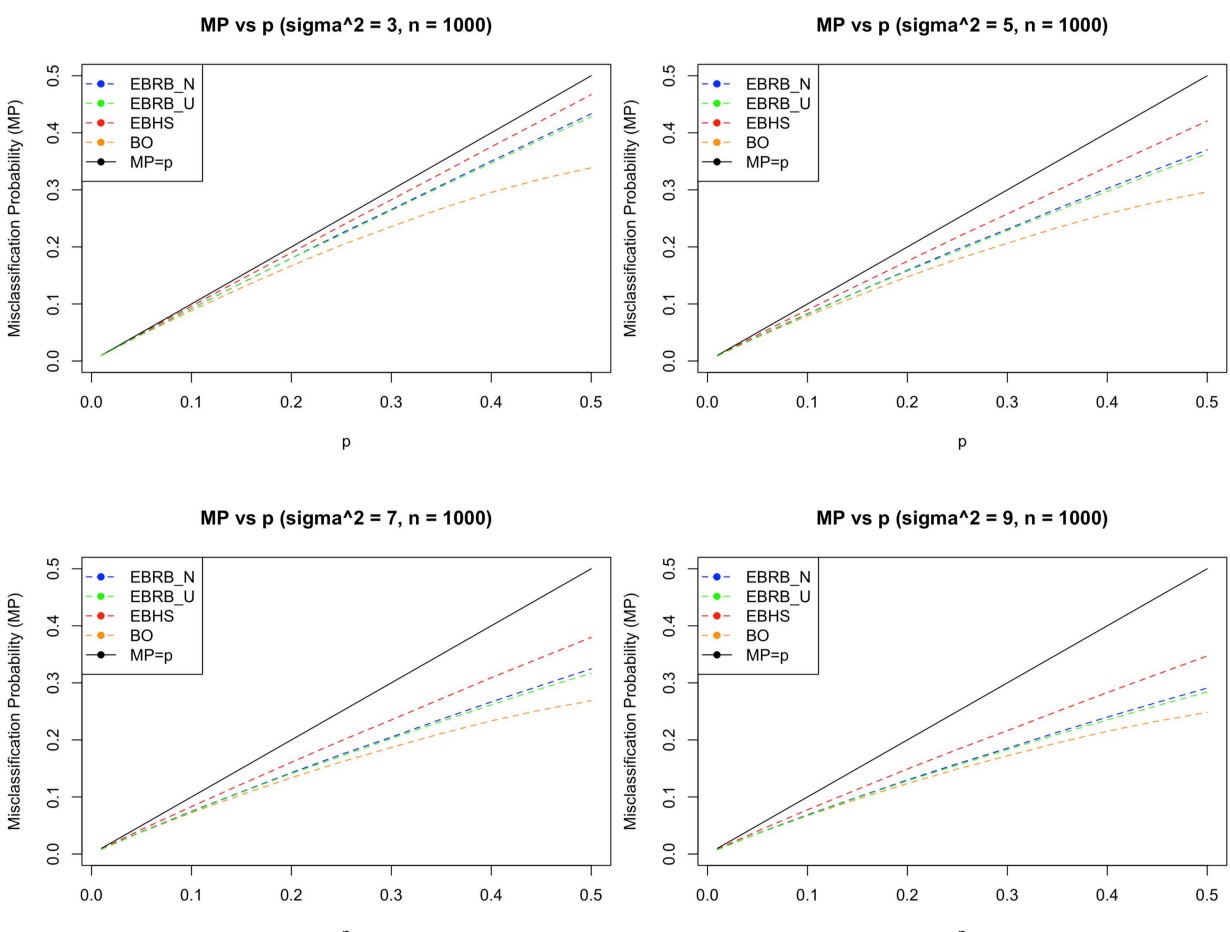

*Figure 2.* Estimated misclassification probabilities for $\mathrm{EBRB_N}$, $\mathrm{EBRB_U}$, EBHS, and BO ($n = 1000$).

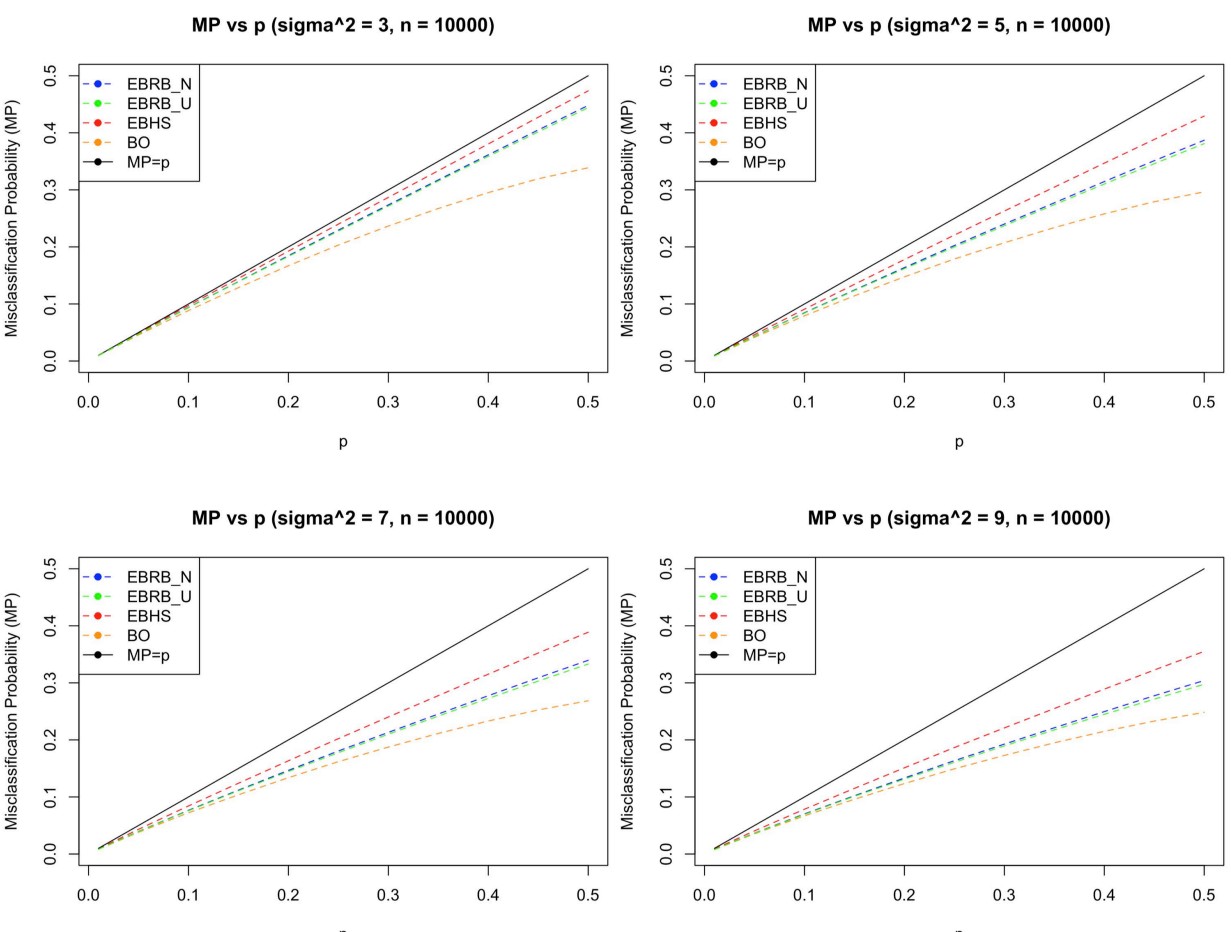

*Figure 3.* Estimated misclassification probabilities for $\mathrm{EBRB_N}$, $\mathrm{EBRB_U}$, EBHS, and BO ($n = 10000$).

# D. Preliminaries on One-Group Shrinkage Priors

The one-group shrinkage priors (i.e., global-local (GL) shrinkage priors) take the form

$$\mu_i \mid \left(\lambda_i^2, \xi^2\right) \sim N\left(0, \lambda_i^2 \xi^2\right), \quad \lambda_i^2 \sim \pi_1, \quad \xi^2 \sim \pi_2.$$

The one-group shrinkage priors encourage sparsity by placing large probabilities on means near zero, while their tails are heavy enough to accommodate large signals at the same time through appropriately chosen heavy-tailed densities. This is accomplished through two levels of parameters to express the prior variances of the $\mu_i$'s, namely, a global shrinkage parameter $\xi$ that is common for all the $\mu_i$'s to cause an overall shrinkage, and the local shrinkage parameters $\lambda_i$'s, which govern the amount of shrinkage at the individual levels. In this sense, we can think that the one-group global-local shrinkage priors approximate a two-groups prior through a continuous density concentrated near zero with heavy tails.

Early instances of one-group shrinkage priors include the Laplace prior in the context of the Bayesian Lasso (Park & Casella, 2008). Since the introduction of the horseshoe prior, a considerable number of new one-group shrinkage priors have been proposed in the literature and used in practice. For example, Armagan et al. (2011) introduced the class of three parameter beta normal mixture priors, and generalized double Pareto priors was introduced in Armagan et al. (2012). The family of three parameter beta normal mixture priors generalizes some well known shrinkage priors including the normal–exponential–gamma prior (Griffin & Brown, 2005) and the horseshoe. The list is quite extensive, and we recommend readers look at Bhadra et al. (2019) for more details.

Connecting to the Bayes oracle we discuss throughout the paper, Carvalho et al. (2010) used the horseshoe prior, proposing the following multiple testing rule under a symmetric $0 - 1$ loss function for the univariate normal means problem

$$\text{reject } H_{0i} \text{ if } 1 - \widehat{\kappa}_i > \frac{1}{2}, \quad i = 1, \ldots, n. \tag{49}$$

Through a series of numerical experiments, Carvalho et al. (2010) demonstrated that, the horseshoe-based procedure using the above half-thresholding rule performs comparably to the Bayes oracle benchmark.

Now, the chosen class of one-group shrinkage priors in Ghosh et al. (2016) can be represented as

$$\mu_i \mid \left(\lambda_i^2, \xi_n^2\right) \sim N\left(0, \lambda_i^2 \xi_n^2\right), \quad \lambda_i^2 \sim \pi_1\left(\lambda_i^2\right) = K\left(\lambda_i^2\right)^{-a-1} L\left(\lambda_i^2\right), \tag{50}$$

independently for $i = 1, \ldots, n$ where $\xi_n$ is treated as the tuning parameter. Here, $K > 0$ is the constant of proportionality, and $a$ is a positive real number. The chosen class of one-group shrinkage priors rely heavily on the slowly varying function (see its definition in Definition D.1 and its properties in Lemma D.2), and the slowly varying component $L$ in (50) is required to satisfy the Assumptions D.3.

**Definition D.1.** A positive measurable function $L$ defined over $(M, \infty)$ for some $M \geq 0$, is said to be slowly varying if $\lim_{t \to \infty} L(\alpha t)/L(t) = 1$ for every fixed $\alpha > 0$.

Below are several important properties for any slowly varying function. A comprehensive treatment of functions of this kind is provided in the classic work (Bingham et al., 1989).

**Lemma D.2.** *If $L$ is any slowly varying function, then*

1. $\lim_{t \to \infty} \frac{\log L(t)}{\log(t)} = 0$,

2. $\lim_{t \to \infty} \frac{L(t)}{t^{\tilde{q}}} = 0, \text{ for all } \tilde{q} > 0$,

3. $L^{\beta}$ *is slowly varying for all $\beta \in \mathbb{R}$.*

**Assumption D.3.** The slowly varying component $L$ in (50) satisfies

(i) There exists some constant $0 < M_0 < \infty$ such that $\sup_{t \in (0,\infty)} L(t) \leq M_0$.

(ii) $\lim_{t \to \infty} L(t) \in (0, \infty)$.

This chosen class in Ghosh et al. (2016) is broad enough to include many one-group priors, such as the horseshoe prior, the three parameter beta normal mixtures priors, the generalized double Pareto priors, and many more. Specifically, the prior of the local shrinkage parameters $\pi_1 \left( \lambda_i^2 \right)$ for the three parameter beta normal mixtures family can be written as

$$\pi_1 \left( \lambda_i^2 \right) = K \left( \lambda_i^2 \right)^{-\beta_0 - 1} L \left( \lambda_i^2 \right)$$

where $L \left( \lambda_i^2 \right) = \left( 1 + \lambda_i^{-2} \right)^{-(\alpha_0 + \beta_0)}$, $K = \frac{\Gamma(\alpha_0 + \beta_0)}{\Gamma(\alpha_0)\Gamma(\beta_0)}$ and $\alpha_0 > 0, \beta_0 > 0$. Here $\Gamma(\alpha_0)$ denotes the gamma function evaluated at $\alpha_0 > 0$. It is clear that $L$ here satisfies Assumption D.3 since $L$ is uniformly bounded by 1 and $\lim_{\lambda_i^2 \to \infty} L \left( \lambda_i^2 \right) = 1$. The three parameter beta normal mixtures family is rich enough to include some well-known one-group shrinkage priors, such as the horseshoe prior ($\alpha_0 = 0.5, \beta_0 = 0.5$) and the normal–exponential–gamma prior ($\alpha_0 = 1, \beta_0 > 0$).

For example, for the generalized double Pareto priors, the prior $\pi_1$ is given by

$$\pi_1 \left( \lambda_i^2 \right) = K \left( \lambda_i^2 \right)^{-\frac{\alpha_1}{2} - 1} L \left( \lambda_i^2 \right),$$

where $L \left( \lambda_i^2 \right) = 2^{\frac{\alpha_1}{2} - 1} \int_0^\infty e^{-\beta_1 \sqrt{2u/\lambda_i^2}} e^{-u} u^{(\frac{\alpha_1}{2} + 1) - 1} du$, $K = \beta_1^{\alpha_1}/\Gamma(\alpha_1)$, with $\alpha_1 > 0, \beta_1 > 0$. By applying the dominated convergence theorem and the monotone convergence theorem, Ghosh et al. (2016) showed $\lim_{\lambda_i^2 \to \infty} L \left( \lambda_i^2 \right) = 2^{\frac{\alpha_1}{2} - 1}\Gamma \left( \frac{\alpha_1}{2} + 1 \right) > 0$ and $L$ is uniformly bounded by $2^{\frac{\alpha_1}{2} - 1}\Gamma \left( \frac{\alpha_1}{2} + 1 \right)$. Thus, $L$ here satisfies Assumption D.3.

In particular, we refer to the class of priors with $a = 0.5$ in (50) as the horseshoe-type priors. This class includes, for example, the three parameter beta normal mixtures with $\alpha_0 = 0.5$ and $\beta_0 > 0$ (e.g., horseshoe), and the generalized double Pareto priors with $\alpha_1 = 1$ (e.g., standard double Pareto), etc. Although not all priors in the horseshoe-type prior class exhibit a spike at zero like the horseshoe prior, they all allocate sufficient mass near the origin, making them effective for handling sparsity. Additionally, some of these priors may have tails that are even heavier than the horseshoe's.

## E. Robustness to the Prior for Relative Belief Inference

L. Al Labadi and M. Evans (2017) consider the sensitivity of the relative belief inference with small perturbations to the prior $\pi$. The discussion is presented for the full parameter $\theta$, and the same arguments apply to a marginal parameter $\psi = \Psi(\theta)$.

Throughout, $\Theta$ is assumed to be finite or countably infinite and fix $\theta_0 \in \Theta$ with $\pi(\theta_0) > 0$. Let $\pi_\epsilon$ be a perturbed prior indexed by $\epsilon \in [0, 1]$, meaning the prior $\pi$ is perturbed by a small amount $\epsilon$. We write $\mathrm{RB}_\epsilon(\theta_0 \mid X)$ as the corresponding relative belief ratio. To assess whether small changes in the prior can materially alter the evidential assessment at $\theta_0$, we measure robustness by the derivative of $\log \mathrm{RB}_\epsilon(\theta_0 \mid X)$ at $\epsilon = 0$, where the size and characteristics of this derivative measure the robustness. Since $\log$ is increasing, the use of $\log \mathrm{RB}_\epsilon(\theta_0 \mid X)$ leads to the same conclusion as using $\mathrm{RB}_\epsilon(\theta_0 \mid X)$.

Two forms of contamination of $\pi$ are considered as follows.

**Definition E.1.** Let $q$ be a probability function on $\Theta$. For $\epsilon \in [0, 1]$, define the contaminated prior $\pi_\epsilon$ by one of the following constructions:

- Linear contamination:
$$\pi_\epsilon(\theta) = (1 - \epsilon)\pi(\theta) + \epsilon q(\theta), \qquad \epsilon \in [0, 1].$$

- Geometric contamination:
$$\pi_\epsilon(\theta) = \frac{\pi^{1-\epsilon}(\theta) \, q^\epsilon(\theta)}{\sum_{\theta \in \Theta} \pi^{1-\epsilon}(\theta) \, q^\epsilon(\theta)}, \qquad \epsilon \in [0, 1].$$

Here, $\sum_{\theta \in \Theta} \pi^{1-\epsilon}(\theta) q^\epsilon(\theta) < \infty$ is assumed for all $\epsilon \in [0, 1]$. Let $V = q/\pi$ and assume $\mathbb{E}_\pi(\log V) < \infty$ and $\mathbb{E}_{\pi(\cdot|x)}(\log V) < \infty$. Further, assume there exists a function $\tilde{V} : \Theta \times [0, 1] \to [0, \infty)$ such that $V^\epsilon(\theta) \leq \tilde{V}(\theta, \epsilon)$ for all $(\theta, \epsilon)$, with $\mathbb{E}_\pi(\tilde{V}) < \infty$ and $\mathbb{E}_{\pi(\cdot|x)}(\tilde{V}) < \infty$.

The following result gives the derivatives of $\log \mathrm{RB}_\epsilon(\theta_0 \mid T(X))$ at $\epsilon = 0$ under the two contaminations.

**Proposition E.2.** *Let $T$ be a minimal sufficient statistic for the model, then $\mathrm{RB}(\theta \mid X) = \mathrm{RB}(\theta \mid T(X))$. Let $m_T$ and $m_T^{(q)}$ denote the prior predictive densities of $T$ under the priors $\pi$ and $q$, respectively, and write $t = T(X)$.*

*(i) Under linear contamination,*

$$\frac{\partial}{\partial \epsilon} \log \mathrm{RB}_\epsilon(\theta_0 \mid t)\Big|_{\epsilon=0} = 1 - \frac{m_T^{(q)}(t)}{m_T(t)}.$$

*(ii) Under geometric contamination, with $V = q/\pi$,*

$$\frac{\partial}{\partial \epsilon} \log \mathrm{RB}_\epsilon(\theta_0 \mid t)\Big|_{\epsilon=0} = \mathbb{E}_\pi(\log V) - \mathbb{E}_{\pi(\cdot \mid t)}(\log V).$$

By Proposition E.2, a convenient check for a priori robustness is whether the a priori expectation of the log-derivative equals zero, which is shown in Corollary E.3 below.

**Corollary E.3.** *Under linear contamination,*

$$\mathbb{E}_{m_T}\left(\frac{\partial}{\partial \epsilon} \log \mathrm{RB}_\epsilon(\theta_0 \mid T(X))\Big|_{\epsilon=0}\right) = \mathbb{E}_{m_T}\left(1 - \frac{m_T^{(q)}(T(X))}{m_T(T(X))}\right) = 0,$$

*and under geometric contamination,*

$$\mathbb{E}_{m_T}\left(\frac{\partial}{\partial \epsilon} \log \mathrm{RB}_\epsilon(\theta_0 \mid T(X))\Big|_{\epsilon=0}\right) = \mathbb{E}_{m_T}\big(\mathbb{E}_\pi(\log V) - \mathbb{E}_{\pi(\cdot \mid T(X))}(\log V)\big) = \mathbb{E}_\pi(\log V) - \mathbb{E}_\pi(\log V) = 0,$$

*where $\mathbb{E}_{m_T}\big(\mathbb{E}_{\pi(\cdot \mid T(X))}(\log V)\big) = \mathbb{E}_\pi(\log V)$ is used. Thus, $\mathrm{RB}(\theta_0 \mid T(X))$ is a priori robust under both contaminations.*

*Remark* E.4. Put a priori robustness in another way: the a priori expectation is that minor perturbations to the prior will not affect the measure of evidence.

We now present the proof of Proposition E.2 below.

*Proof.* Write $t = T(x)$. For each $\theta \in \Theta$, let $f_{\theta,T}(t)$ denote the sampling density (or mass function) of $T$ under $\theta$. Since $\Theta$ is finite or countably infinite and $\pi_\epsilon(\theta_0) > 0$, Bayes' rule gives

$$\pi_\epsilon(\theta_0 \mid t) = \frac{\pi_\epsilon(\theta_0) \, f_{\theta_0,T}(t)}{\sum_{\theta \in \Theta} \pi_\epsilon(\theta) \, f_{\theta,T}(t)} = \frac{\pi_\epsilon(\theta_0) \, f_{\theta_0,T}(t)}{m_{\epsilon,T}(t)}, \qquad m_{\epsilon,T}(t) := \sum_{\theta \in \Theta} \pi_\epsilon(\theta) \, f_{\theta,T}(t). \tag{51}$$

Hence

$$\mathrm{RB}_\epsilon(\theta_0 \mid t) = \frac{\pi_\epsilon(\theta_0 \mid t)}{\pi_\epsilon(\theta_0)} = \frac{f_{\theta_0,T}(t)}{m_{\epsilon,T}(t)}.$$

In particular, the dependence on $\epsilon$ enters only through $m_{\epsilon,T}(t)$.

*(i) Linear contamination.* Under $\pi_\epsilon = (1 - \epsilon)\pi + \epsilon q$, we have

$$m_{\epsilon,T}(t) = \sum_{\theta \in \Theta} \big((1 - \epsilon)\pi(\theta) + \epsilon q(\theta)\big) f_{\theta,T}(t) = (1 - \epsilon)m_T(t) + \epsilon m_T^{(q)}(t),$$

where $m_T(t) = \sum_{\theta \in \Theta} \pi(\theta) f_{\theta,T}(t)$ and $m_T^{(q)}(t) = \sum_{\theta \in \Theta} q(\theta) f_{\theta,T}(t)$. Therefore,

$$\log \mathrm{RB}_\epsilon(\theta_0 \mid t) = \log f_{\theta_0,T}(t) - \log\big((1 - \epsilon)m_T(t) + \epsilon m_T^{(q)}(t)\big).$$

Differentiating and evaluating at $\epsilon = 0$ yields

$$\frac{\partial}{\partial \epsilon} \log \mathrm{RB}_\epsilon(\theta_0 \mid t)\Big|_{\epsilon=0} = -\frac{-m_T(t) + m_T^{(q)}(t)}{(1 - \epsilon)m_T(t) + \epsilon m_T^{(q)}(t)}\Big|_{\epsilon=0} = 1 - \frac{m_T^{(q)}(t)}{m_T(t)}.$$

*(ii) Geometric contamination.* Let $V = q/\pi$. The geometric contamination can be rewritten as

$$\pi_\epsilon(\theta) = \frac{\pi^{1-\epsilon}(\theta)q^\epsilon(\theta)}{\sum_{\theta \in \Theta} \pi^{1-\epsilon}(\theta)q^\epsilon(\theta)} = \frac{\pi(\theta)V^\epsilon(\theta)}{\sum_{\theta \in \Theta} \pi(\theta)V^\epsilon(\theta)} = \frac{\pi(\theta)V^\epsilon(\theta)}{\mathbb{E}_\pi(V^\epsilon)}.$$

Substituting this into the definition of $m_{\epsilon,T}(t)$ in (51) gives

$$m_{\epsilon,T}(t) = \frac{1}{\mathbb{E}_\pi(V^\epsilon)} \sum_{\theta \in \Theta} \pi(\theta) V^\epsilon(\theta) f_{\theta,T}(t).$$

Let $\pi(\cdot \mid t)$ denote the posterior induced by the prior $\pi$, i.e.,

$$\pi(\theta \mid t) = \frac{\pi(\theta) f_{\theta,T}(t)}{m_T(t)}.$$

Then

$$\sum_{\theta \in \Theta} \pi(\theta) V^\epsilon(\theta) f_{\theta,T}(t) = m_T(t) \sum_{\theta \in \Theta} \pi(\theta \mid t) V^\epsilon(\theta) = m_T(t) \, \mathbb{E}_{\pi(\cdot|t)}(V^\epsilon),$$

and hence

$$m_{\epsilon,T}(t) = \frac{m_T(t) \, \mathbb{E}_{\pi(\cdot|t)}(V^\epsilon)}{\mathbb{E}_\pi(V^\epsilon)}.$$

Using $\mathrm{RB}_\epsilon(\theta_0 \mid t) = f_{\theta_0,T}(t)/m_{\epsilon,T}(t)$, we obtain

$$\log \mathrm{RB}_\epsilon(\theta_0 \mid t) = \log \frac{f_{\theta_0,T}(t)}{m_T(t)} - \log \mathbb{E}_{\pi(\cdot|t)}(V^\epsilon) + \log \mathbb{E}_\pi(V^\epsilon).$$

The stated integrability assumptions in Definition E.1 ensure that differentiation may be exchanged with expectation, so that

$$\frac{\partial}{\partial \epsilon} \log \mathbb{E}_\pi(V^\epsilon) \bigg|_{\epsilon=0} = \frac{\mathbb{E}_\pi(V^\epsilon \log V)\big|_{\epsilon=0}}{\mathbb{E}_\pi(V^\epsilon)\big|_{\epsilon=0}} = \mathbb{E}_\pi(\log V),$$

and similarly,

$$\frac{\partial}{\partial \epsilon} \log \mathbb{E}_{\pi(\cdot|t)}(V^\epsilon) \bigg|_{\epsilon=0} = \mathbb{E}_{\pi(\cdot|t)}(\log V).$$

Combining the above derivatives gives

$$\frac{\partial}{\partial \epsilon} \log \mathrm{RB}_\epsilon(\theta_0 \mid t) \bigg|_{\epsilon=0} = \mathbb{E}_\pi(\log V) - \mathbb{E}_{\pi(\cdot|t)}(\log V),$$

as desired. $\qquad \square$

## F. Proofs for ABOS via Relative Belief in the Univariate Normal Means Problem

In this section, we provide detailed proofs for the theoretical results established in Section 5. We begin by deriving the closed-form expression of the relative belief ratio (Proposition 5.1), and the corresponding error probabilities (Proposition 5.3). Subsequently, we prove the existence and uniqueness of the optimal hyperparameter $\tau^*$ that minimizes the Bayes risk (Proposition 5.4). Considering the asymptotic framework of Bogdan et al. (2011), we analyze the asymptotic order of $\tau_n^*$ (Lemma 5.6), and demonstrate that the relative belief multiple testing procedure attains asymptotic Bayes optimality under sparsity when the tuning parameter $\tau_n$ is of this asymptotic order (Theorem 5.7). Finally, we develop a random threshold ABOS criterion (Theorem F.2) by utilizing the fixed threshold criterion of Bogdan et al. (2011), and apply it to prove exact ABOS for the data-driven empirical Bayes relative belief rule $\mathrm{EBRB}_N$ (Theorem 5.8).

### F.1. Proof of Proposition 5.1

*Proof.* The prior density for $\mu_i$ is given by

$$\pi(\mu_i \mid \tau^2) = \frac{1}{\sqrt{2\pi\tau^2}} \exp\left(-\frac{\mu_i^2}{2\tau^2}\right).$$

The associated likelihood function is

$$\mathcal{L}(\mu_i; X_i) = \frac{1}{\sqrt{2\pi}} \exp\left(-\frac{(X_i - \mu_i)^2}{2}\right).$$

By Bayes' theorem, the posterior distribution of $\mu_i$ given $X_i$ is proportional to the product of the likelihood and the prior

$$\pi(\mu_i \mid X_i, \tau^2) \propto \exp\left(-\frac{(X_i - \mu_i)^2}{2}\right) \exp\left(-\frac{\mu_i^2}{2\tau^2}\right).$$

Simplifying the exponent

$$-\frac{(X_i - \mu_i)^2}{2} - \frac{\mu_i^2}{2\tau^2} = -\frac{1}{2}\left(\mu_i^2 - 2X_i\mu_i + X_i^2 + \frac{\mu_i^2}{\tau^2}\right).$$

Combining terms

$$-\frac{1}{2}\left(1 + \frac{1}{\tau^2}\right)\mu_i^2 + X_i\mu_i - \frac{X_i^2}{2}.$$

Completing the square for $\mu_i$ to identify the normal kernel

$$-\frac{1}{2}\left(1 + \frac{1}{\tau^2}\right)\left(\mu_i - \frac{X_i}{1 + \frac{1}{\tau^2}}\right)^2 + \text{constant terms}.$$

Thus, the posterior distribution of $\mu_i$ given $X_i$ is normal

$$\mu_i \mid X_i, \tau^2 \sim N\left(\frac{X_i}{1 + \frac{1}{\tau^2}}, \frac{1}{1 + \frac{1}{\tau^2}}\right).$$

Substituting $\mu_i = 0$ into the posterior density

$$\pi(\mu_i = 0 \mid X_i, \tau^2) = \frac{1}{\sqrt{\frac{2\pi}{1 + \frac{1}{\tau^2}}}} \exp\left(-\frac{X_i^2}{2\left(1 + \frac{1}{\tau^2}\right)}\right).$$

Substituting $\mu_i = 0$ into the prior density

$$\pi(\mu_i = 0) = \frac{1}{\sqrt{2\pi\tau^2}}.$$

Taking the ratio of the posterior to the prior

$$\begin{aligned}
\text{RB}(\mu_i = 0 \mid X_i, \tau^2) &= \frac{\pi(\mu_i = 0 \mid X_i, \tau^2)}{\pi(\mu_i = 0)} \\
&= \frac{\sqrt{2\pi\tau^2}}{\sqrt{\frac{2\pi}{1 + \frac{1}{\tau^2}}}} \exp\left(-\frac{X_i^2}{2\left(1 + \frac{1}{\tau^2}\right)}\right) \\
&= \sqrt{1 + \tau^2} \exp\left(-\frac{X_i^2}{2\left(1 + \frac{1}{\tau^2}\right)}\right).
\end{aligned}$$

$\square$

### F.2. Proof of Proposition 5.3

*Proof.* Recall $t_{1i}^{RB}$ and $t_{2i}^{RB}$ denote the type I and type II error probabilities, respectively, then

$$\begin{aligned}
t_{1i}^{RB} &= \mathbb{P}_{H_{0i}:X_i \sim N(0,1)}(|X_i| > r(\tau)), \\
t_{2i}^{RB} &= \mathbb{P}_{H_{1i}:X_i \sim N(0,1+\sigma^2)}(|X_i| < r(\tau)).
\end{aligned}$$

To calculate the type I error probability $t_{1i}^{RB}$, we proceed as follows. Specifically, the probability of type I error under $H_{0i}$ is defined as

$$t_{1i}^{RB} = \mathbb{P}_{H_{0i}:X_i \sim N(0,1)}\left(\text{RB}(\mu_i = 0 \mid X_i, \tau^2) < 1\right),$$

where $\text{RB}(\mu_i = 0 \mid X_i, \tau^2) < 1$ is equivalent to

$$|X_i| > \sqrt{\left(1 + \frac{1}{\tau^2}\right)\log(1 + \tau^2)}.$$

Under $H_{0i}$, we have
$$X_i \mid \mu_i \sim N(\mu_i, 1),$$
where $\mu_i \sim \delta_{\{0\}}$. Therefore, $X_i$ is Gaussian with
$$\mathbb{E}[X_i] = \mathbb{E}[\mathbb{E}[X_i \mid \mu_i]] = \mathbb{E}[\mu_i] = 0,$$
and
$$\mathrm{Var}(X_i) = \mathbb{E}[\mathrm{Var}(X_i \mid \mu_i)] + \mathrm{Var}(\mathbb{E}[X_i \mid \mu_i]) = \mathbb{E}[1] + \mathrm{Var}(\mu_i) = 1.$$
Thus, $X_i \sim N(0, 1)$. Let the standardized variable $Z_i$ be $Z_i = X_i \sim N(0, 1)$. The type I error probability can be expressed as
$$t_{1i}^{RB} = \mathbb{P}\left( |Z_i| > \sqrt{\left(1 + \frac{1}{\tau^2}\right) \log(1 + \tau^2)} \right).$$
Let $r(\tau) = \sqrt{\left(1 + \frac{1}{\tau^2}\right) \log(1 + \tau^2)}$. This simplifies to
$$t_{1i}^{RB} = \mathbb{P}\left( Z_i > r(\tau) \right) + \mathbb{P}\left( Z_i < -r(\tau) \right).$$
Using the cumulative distribution function (cdf) of the standard normal distribution $\Phi(\cdot)$, we have
$$t_{1i}^{RB} = 1 - \Phi(r(\tau)) + \Phi(-r(\tau)) = 2(1 - \Phi(r(\tau))).$$
Therefore:
$$t_{1i}^{RB} = 2 - 2\Phi(r(\tau)).$$
The type II error probability $t_{2i}^{RB}$ is computed analogously, noting that $X_i \sim N(0, 1 + \sigma^2)$ under $H_{1i}$. $\qquad\square$

### F.3. Proof of Proposition 5.4

*Proof.* Let
$$Q = r(\tau) = \sqrt{\left(1 + \frac{1}{\tau^2}\right) \log(1 + \tau^2)}, \qquad \tau > 0, \tag{52}$$
and write the Bayes risk as a function of $Q$:
$$R^{RB}(Q) = 2(n - np)\left[1 - \Phi(Q)\right] + np\left[2\Phi\left(\frac{Q}{\sqrt{1+\sigma^2}}\right) - 1\right], \tag{53}$$
where $\Phi$ is the standard normal cdf and $\varphi(x) = (2\pi)^{-1/2} e^{-x^2/2}$ is the standard normal pdf.

By the chain rule, we have
$$\frac{dR^{RB}}{d\tau}(\tau) = \frac{dR^{RB}}{dQ}(Q) \cdot \frac{dQ}{d\tau}(\tau).$$
Let $t = \tau^2 > 0$ and define $\widetilde{h}(t) = \left(1 + t^{-1}\right) \log(1 + t)$. Then $Q = \sqrt{\widetilde{h}(t)}$ and
$$\widetilde{h}'(t) = \frac{t - \log(1 + t)}{t^2} > 0, \quad \forall\, t > 0,$$
so $\widetilde{h}$ is strictly increasing and hence $Q'(\tau) > 0$ for all $\tau > 0$. Therefore the sign of $\frac{dR^{RB}}{d\tau}$ is the same as that of $\frac{dR^{RB}}{dQ}$.

Differentiating (53) and factoring $\varphi(Q)$ yields
$$\frac{dR^{RB}}{dQ}(Q) = -2(n - np)\,\varphi(Q) + 2np\,\frac{1}{\sqrt{1+\sigma^2}}\,\varphi\left(\frac{Q}{\sqrt{1+\sigma^2}}\right)$$
$$= 2n\,\varphi(Q)\left[-(1 - p) + p\,\frac{1}{\sqrt{1+\sigma^2}}\,\exp\left(\frac{\sigma^2}{2(1+\sigma^2)}\,Q^2\right)\right]. \tag{54}$$

Solving for $\frac{dR^{RB}}{dQ} = 0$ gives us the only critical point

$$Q_* = \sqrt{\frac{1+\sigma^2}{\sigma^2}\left(2\log f + \log(1+\sigma^2)\right)} \tag{55}$$

Since $Q \in (1, \infty)$ by (52), the equality admits a feasible solution precisely when

$$\frac{1+\sigma^2}{\sigma^2}\left(2\log f + \log(1+\sigma^2)\right) > 1. \tag{56}$$

In this regime (56), we verify (55) is indeed a unique global minimizer in the sequel. Define

$$\widetilde{g}(Q) := -(1-p) + p\,\frac{1}{\sqrt{1+\sigma^2}}\,\exp\!\left(\frac{\sigma^2}{2(1+\sigma^2)}\,Q^2\right). \tag{57}$$

Since $\varphi(Q) > 0$, $\mathrm{sign}\!\left(\frac{dR^{RB}}{dQ}(Q)\right) = \mathrm{sign}\!\left(\widetilde{g}(Q)\right)$. Moreover,

$$\frac{d\widetilde{g}}{dQ}(Q) = p\,\frac{1}{\sqrt{1+\sigma^2}}\,\frac{\sigma^2}{1+\sigma^2}\,Q\,\exp\!\left(\frac{\sigma^2}{2(1+\sigma^2)}\,Q^2\right) > 0 \quad \text{where } Q > 1, \tag{58}$$

so $\widetilde{g}$ is strictly increasing on $(1, \infty)$.

Moreover,

$$\lim_{Q\to\infty}\widetilde{g}(Q) = +\infty. \tag{59}$$

Considering $Q \downarrow 1$ (when $\tau \downarrow 0$), we have

$$\lim_{Q\downarrow 1}\widetilde{g}(Q) = -(1-p) + p\,\frac{1}{\sqrt{1+\sigma^2}}\,\exp\!\left(\frac{\sigma^2}{2(1+\sigma^2)}\right) < 0, \tag{60}$$

based on $\frac{1+\sigma^2}{\sigma^2}\left\{2\log f + \log(1+\sigma^2)\right\} > 1$ in (56).

By continuity and the strict monotonicity in (58), the $Q_* \in (1, \infty)$ in (55) is unique with

$$\widetilde{g}(Q_*) = 0. \tag{61}$$

Now, we first establish $Q_*$ is a strict local minimizer of $R^{RB}(Q)$. Differentiate (54) once more. Using $\varphi'(Q) = -Q\,\varphi(Q)$, write

$$\frac{d^2 R^{RB}}{dQ^2}(Q) = 2n\,\varphi'(Q)\,\widetilde{g}(Q) + 2n\,\varphi(Q)\,\widetilde{g}'(Q).$$

At the critical point $Q_*$ we have $\widetilde{g}(Q_*) = 0$, hence

$$\frac{d^2 R^{RB}}{dQ^2}(Q_*) = 2n\,\varphi(Q_*)\,\widetilde{g}'(Q_*).$$

By (58), $\widetilde{g}'(Q_*) = p\,\frac{1}{\sqrt{1+\sigma^2}}\,\frac{\sigma^2}{1+\sigma^2}\,Q_*\,\exp\!\left(\frac{\sigma^2}{2(1+\sigma^2)}\,Q_*^2\right) > 0$. Since $\varphi(Q_*) > 0$, $\frac{d^2 R^{RB}}{dQ^2}(Q_*) > 0$. Thus $Q_*$ is a strict local minimizer of $R^{RB}(Q)$ on $(1, \infty)$.

By (59), (60), (61), and the fact $\widetilde{g}$ is strictly increasing on $(1, \infty)$, we know that $\frac{dR^{RB}}{dQ} < 0$ on $(1, Q_*)$ and $\frac{dR^{RB}}{dQ} > 0$ on $(Q_*, \infty)$. Hence, $Q_*$ is the unique global minimizer of $R^{RB}(Q)$ over the feasible domain $Q \in (1, \infty)$.

Since $r$ is strictly increasing on its feasible domain, there is a unique $\tau^* > 0$ such that $r(\tau^*) = Q_*$. Using (55), the corresponding unique global minimizer $\tau^*$ for $R^{RB}(\tau)$ is

$$\tau^* = r^{-1}\!\left(\sqrt{\frac{1+\sigma^2}{\sigma^2}\left(2\log f + \log(1+\sigma^2)\right)}\right),$$

which is exactly the expression stated in (10). This completes the proof. $\qquad\square$

### F.4. Proof of Lemma 5.6

*Proof.* Recall $r(\tau) = \sqrt{(1 + \frac{1}{\tau^2}) \log(1 + \tau^2)}$. By definition in (10), $\tau_n^*$ satisfies $r(\tau_n^*) = w_n^*$ under Assumption 2.3, where $w_n^*$ is

$$w_n^* = \sqrt{\frac{1 + \sigma_n^2}{\sigma_n^2} \left( 2 \log f_n + \log(1 + \sigma_n^2) \right)} \tag{62}$$

Thus, we have

$$\begin{aligned}
(w_n^*)^2 &= \left(1 + \frac{1}{\sigma_n^2}\right)\left(2 \log f_n + \log(1 + \sigma_n^2)\right) \\
&= 2 \log f_n + \log(\sigma_n^2) + \log\left(1 + \frac{1}{\sigma_n^2}\right) + \frac{1}{\sigma_n^2}\left(2 \log f_n + \log(1 + \sigma_n^2)\right) \\
&= 2 \log f_n + \log(\sigma_n^2) + \frac{2}{\sigma_n^2} \log f_n + o(1)
\end{aligned}$$

Under Assumption 2.3, we have $\frac{\log(f_n^2)}{u_n} \to C \in (0, \infty)$ as $n \to \infty$, thus

$$\frac{\log(f_n^2)}{\sigma_n^2} = \frac{2 \log f_n}{\sigma_n^2} \longrightarrow C \in (0, \infty),$$

so

$$\sigma_n^2 = \frac{2 \log f_n}{C}(1 + o(1)), \qquad \frac{1}{\sigma_n^2} = \frac{C}{2 \log f_n}(1 + o(1)), \quad \text{as} \quad n \to \infty. \tag{63}$$

From (63), we obtain

$$\log(\sigma_n^2) = \log\left(\frac{2 \log f_n}{C}(1 + o(1))\right) = \log\left(\tfrac{2}{C}\right) + \log\left(\log f_n\right) + o(1). \tag{64}$$

Hence, combining (63) and (64), we have

$$\begin{aligned}
(w_n^*)^2 &= 2 \log f_n + \log(\sigma_n^2) + \frac{2}{\sigma_n^2} \log f_n + o(1) \\
&= 2 \log f_n + \log\left(\log f_n\right) + \log\left(\tfrac{2}{C}\right) + C + o(1).
\end{aligned} \tag{65}$$

In particular, $(w_n^*)^2 \to \infty$ as $n \to \infty$. The function $r(\tau) = \sqrt{(1 + \frac{1}{\tau^2}) \log(1 + \tau^2)}$ is continuous and strictly increasing for all $\tau \in (0, \infty)$. Since $w_n^* = r(\tau_n^*)$ and $r$ is a strictly increasing function which approaches infinity only as its argument does, it follows that $\tau_n^* \to \infty$ as $n \to \infty$.

The asymptotic behavior of $r(\tau_n^*)$ is as follows.

$$\begin{aligned}
[r(\tau_n^*)]^2 &= \left(1 + \frac{1}{(\tau_n^*)^2}\right) \log(1 + (\tau_n^*)^2) \tag{66} \\
&= \log((\tau_n^*)^2) + \log\left(1 + \frac{1}{(\tau_n^*)^2}\right) + \frac{\log(1 + (\tau_n^*)^2)}{(\tau_n^*)^2} \\
&= \log((\tau_n^*)^2) + o(1) \tag{67}
\end{aligned}$$

Equating the asymptotic expressions for $r(\tau_n^*)$ and $w_n^*$ (i.e., (65) and (67)) gives

$$\begin{aligned}
\log((\tau_n^*)^2) &= (w_n^*)^2 + o(1) \\
&= 2 \log f_n + \log\left(\log f_n\right) + \log\left(\tfrac{2}{C}\right) + C + o(1). \tag{68}
\end{aligned}$$

Exponentiating both sides of (68) gives

$$(\tau_n^*)^2 = \exp\left(2\log f_n + \log(\log f_n) + \log\left(\tfrac{2}{C}\right) + C + o(1)\right)$$

$$= f_n^2 \log f_n \frac{2}{C} e^C \exp(o(1))$$

$$= \frac{2e^C}{C} f_n^2 \log f_n (1 + o(1)).$$

Since $p_n \to 0$ implies $f_n = (1 - p_n)/p_n \sim p_n^{-1}$ and $\log f_n \sim \log(1/p_n)$, we obtain

$$(\tau_n^*)^2 = \frac{2e^C}{C} p_n^{-2} \log\left(\frac{1}{p_n}\right) (1 + o(1)),$$

and therefore

$$\frac{\tau_n^*}{p_n^{-1}\sqrt{\log(1/p_n)}} = \sqrt{\frac{2e^C}{C}} (1 + o(1)).$$

Thus,

$$\lim_{n\to\infty} \frac{\tau_n^*}{p_n^{-1}\sqrt{\log(1/p_n)}} = \sqrt{\frac{2e^C}{C}} \in (0, \infty).$$

$\square$

### F.5. Proof of Theorem 5.7

*Proof.* Suppose Assumption 2.3 holds. The assumption on the tuning parameter $\tau_n$ implies the existence of a constant $\widetilde{\gamma} \in (0, \infty)$ such that

$$\tau_n = \widetilde{\gamma} p_n^{-1} \sqrt{\log(1/p_n)} (1 + o(1)). \tag{69}$$

Let $r(\tau_n) = \sqrt{(1 + \frac{1}{\tau_n^2}) \log(1 + \tau_n^2)}$. Since $\tau_n \to \infty$, we have

$$[r(\tau_n)]^2 = \log(\tau_n^2) + o(1)$$

$$= \log\left(\widetilde{\gamma}^2 p_n^{-2} \log(1/p_n) (1 + o(1))\right) + o(1)$$

$$= 2\log(p_n^{-1}) + \log\left(\log(1/p_n)\right) + \log(\widetilde{\gamma}^2) + o(1). \tag{70}$$

We first analyze the type I error probability

$$t_{1i}^{AsyRB} = 2\left(1 - \Phi(r(\tau_n))\right).$$

Using the standard Gaussian tail bound

$$1 - \Phi(x) \leq \frac{1}{x\sqrt{2\pi}} e^{-x^2/2}, \qquad x > 0,$$

we obtain

$$t_{1i}^{AsyRB} \leq \frac{2}{r(\tau_n)\sqrt{2\pi}} e^{-[r(\tau_n)]^2/2}. \tag{71}$$

From (70),

$$\frac{[r(\tau_n)]^2}{2} = \log(1/p_n) + \tfrac{1}{2}\log\left(\log(1/p_n)\right) + \tfrac{1}{2}\log(\widetilde{\gamma}^2) + o(1),$$

and hence

$$e^{-[r(\tau_n)]^2/2} = \exp\left(-\log(1/p_n)\right) \exp\left(-\tfrac{1}{2}\log\log(1/p_n)\right) \exp\left(-\tfrac{1}{2}\log(\widetilde{\gamma}^2)\right) e^{o(1)}$$

$$= p_n \left(\log(1/p_n)\right)^{-1/2} \widetilde{\gamma}^{-1} (1 + o(1)). \tag{72}$$

Moreover, from (70),

$$\frac{[r(\tau_n)]^2}{2\log(1/p_n)} = 1 + \frac{\log\log(1/p_n) + \log(\widetilde{\gamma}^2) + o(1)}{2\log(1/p_n)} \longrightarrow 1,$$

so $[r(\tau_n)]^2 = 2\log(1/p_n)(1 + o(1))$ and therefore

$$r(\tau_n) = \sqrt{2\log(1/p_n)}\,(1 + o(1)), \qquad \frac{1}{r(\tau_n)} = \frac{1}{\sqrt{2\log(1/p_n)}}\,(1 + o(1)). \tag{73}$$

Combining (71), (72), and (73), we obtain

$$0 \le t_{1i}^{AsyRB} \le \frac{2}{\sqrt{2\pi}}\,\frac{1}{\sqrt{2\log(1/p_n)}}\,p_n\left(\log(1/p_n)\right)^{-1/2}\widetilde{\gamma}^{-1}(1 + o(1))$$

Equivalently, we have

$$0 \le t_{1i}^{AsyRB} \le \widetilde{C}\,p_n\left(\log(1/p_n)\right)^{-1}(1 + o(1)),$$

where $\widetilde{C} = \frac{1}{\sqrt{\pi\widetilde{\gamma}}} \in (0,\infty)$. Thus,

$$0 \le \frac{t_{1i}^{AsyRB}}{p_n} \le \widetilde{C}\left(\log(1/p_n)\right)^{-1}(1 + o(1)).$$

Since $\log(1/p_n) \to \infty$, so $\widetilde{C}\left(\log(1/p_n)\right)^{-1}(1 + o(1)) \to 0$. Hence,

$$\frac{t_{1i}^{AsyRB}}{p_n} \longrightarrow 0, \quad \text{i.e.,} \quad t_{1i}^{AsyRB} = o\left(p_n\right) \tag{74}$$

Next, we analyze the asymptotic type II error probability,

$$t_{2i}^{AsyRB} = 2\Phi\left(\frac{r(\tau_n)}{\sqrt{1+\sigma_n^2}}\right) - 1.$$

From Assumption 2.3, we have

$$\frac{\log(f_n^2)}{\sigma_n^2} \to C \quad \Longrightarrow \quad \sigma_n^2 = \frac{2\log f_n}{C}\,(1 + o(1)), \qquad n \to \infty.$$

Using (70), we obtain

$$\frac{[r(\tau_n)]^2}{1+\sigma_n^2} = \frac{2\log(1/p_n) + \log\log(1/p_n) + \log(\gamma^2) + o(1)}{1 + \frac{2\log f_n}{C}(1 + o(1))}. \tag{75}$$

Dividing both the numerator and denominator of (75) by $2\log(1/p_n)$ yields

$$\frac{[r(\tau_n)]^2}{1+\sigma_n^2} = \frac{1 + \dfrac{\log\log(1/p_n)}{2\log(1/p_n)} + \dfrac{\log(\gamma^2) + o(1)}{2\log(1/p_n)}}{\dfrac{1}{2\log(1/p_n)} + \dfrac{1}{C}\dfrac{\log f_n}{\log(1/p_n)}(1 + o(1))}. \tag{76}$$

As $p_n \to 0$, we have $\log(\log(1/p_n))/\log(1/p_n) \to 0$, $1/\log(1/p_n) \to 0$, and $\log f_n/\log(1/p_n) \to 1$, so the numerator of (76) converges to 1 and the denominator of (76) converges to $1/C$. Hence,

$$\frac{[r(\tau_n)]^2}{1+\sigma_n^2} \longrightarrow C, \qquad \text{and so} \qquad \frac{r(\tau_n)}{\sqrt{1+\sigma_n^2}} \longrightarrow \sqrt{C}.$$

By continuity of $\Phi$,

$$t_{2i}^{AsyRB} = 2\Phi\left(\frac{r(\tau_n)}{\sqrt{1+\sigma_n^2}}\right) - 1 = 2\Phi(\sqrt{C}) - 1 + o(1) = (2\Phi(\sqrt{C}) - 1)(1 + o(1)). \tag{77}$$

Finally, we compute the asymptotic Bayes risk. The asymptotic Bayes risk of the relative belief procedure is

$$R^{AsyRB} = n(1 - p_n)\, t_{1i}^{AsyRB} + np_n\, t_{2i}^{AsyRB}.$$

Using (74) and (77),

$$
\begin{aligned}
R^{AsyRB} &= n(1 - p_n)\, o(p_n) + np_n\big(2\Phi(\sqrt{C}) - 1 + o(1)\big) \\
&= o(np_n) + np_n\big(2\Phi(\sqrt{C}) - 1\big) \\
&= np_n\big(2\Phi(\sqrt{C}) - 1\big)(1 + o(1)).
\end{aligned}
$$

The resulting asymptotic Bayes risk of the relative belief procedure coincides with the asymptotic Bayes risk of the Bayes oracle, so the relative belief multiple testing procedure achieves ABOS. □

### F.6. Proof of Theorem 5.8

The preceding ABOS result relies on a tuning parameter $\tau_n$ whose choice depends on the unknown parameter $p$, and hence it is not directly applicable in practice. We then turn to an empirical Bayes relative belief rule with a data-dependent estimate. Since the associated rejection threshold is random, we first recall the fixed threshold ABOS criterion in Bogdan et al. (2011), and then establish a random threshold version (i.e., Theorem F.2) that will be used below.

**Theorem F.1** (Theorem 3.2 of Bogdan et al. (2011)). *Suppose Assumption 2.3 holds. Consider a fixed threshold multiple testing rule that rejects $H_{0i}$ whenever*

$$X_i^2 > c_n^2,$$

*where*

$$c_n^2 = \log v_n + z_n.$$

*Then, this rule attains ABOS if and only if*

$$z_n = o(\log v_n) \quad and \quad z_n + 2\log\log v_n \to \infty.$$

Parallel to Theorem F.1, we next give a general sufficient condition under which a multiple testing rule whose threshold may depend on the observed data attains ABOS.

**Theorem F.2.** *Suppose Assumption 2.3 holds. Consider some fixed sequence $c_{n,\mathrm{fix}}^2$ satisfying the conditions in Theorem F.1. In other words,*

$$c_{n,\mathrm{fix}}^2 = \log v_n + z_n,$$

*where*

$$z_n = o(\log v_n) \quad and \quad z_n + 2\log\log v_n \to \infty.$$

*Define a multiple testing rule by rejecting $H_{0i}$ whenever*

$$X_i^2 > \widehat{c}_n^2, \qquad i = 1, \dots, n,$$

*where $\widehat{c}_n^2$ is a random threshold. If for all $\varepsilon > 0$,*

$$\mathbb{P}\big(\big|\widehat{c}_n^2 - c_{n,\mathrm{fix}}^2\big| > \varepsilon\big) = o(p_n),$$

*then this multiple testing rule attains ABOS.*

*Proof of Theorem F.2.* To analyze the Bayes risk induced by the random threshold rule, we first define the event

$$\mathcal{A}_n = \Big\{\big|\widehat{c}_n^2 - c_{n,\mathrm{fix}}^2\big| \le \varepsilon\Big\}, \tag{78}$$

where $\varepsilon > 0$ is fixed. Under the additive 0-1 loss, let

$$L_{n,i} = \mathbf{1}\{\mu_i = 0,\ X_i^2 > \widehat{c}_n^2\} + \mathbf{1}\{\mu_i \ne 0,\ X_i^2 < \widehat{c}_n^2\}, \qquad i = 1, \dots, n. \tag{79}$$

Then, the Bayes risk of the random threshold rule admits the decomposition

$$
\begin{aligned}
R_n(\widehat{c}_n^2) &= \mathbb{E}\Big[\sum_{i=1}^n L_{n,i}\Big] \\
&= \mathbb{E}\Big[\sum_{i=1}^n L_{n,i}\mathbf{1}_{\mathcal{A}_n}\Big] + \mathbb{E}\Big[\sum_{i=1}^n L_{n,i}\mathbf{1}_{\mathcal{A}_n^c}\Big] \\
&:= R_{n,\mathcal{A}} + R_{n,\mathcal{A}^c}.
\end{aligned}
\tag{80}
$$

Our goal is to prove that

$$
R_n(\widehat{c}_n^2) \le R_{Opt}^{AsyBO}(1 + o(1)).
\tag{81}
$$

Since the Bayes oracle attains the minimum asymptotic Bayes risk over all multiple testing rules, we also have

$$
R_n(\widehat{c}_n^2) \ge R_{Opt}^{AsyBO}(1 + o(1)).
\tag{82}
$$

Therefore, once (81) is established, (82) yields

$$
1 \le \frac{R_n(\widehat{c}_n^2)}{R_{Opt}^{AsyBO}} \le 1 + o(1),
\tag{83}
$$

and hence,

$$
\frac{R_n(\widehat{c}_n^2)}{R_{Opt}^{AsyBO}} \to 1.
\tag{84}
$$

In the following, we analyze the two terms $R_{n,\mathcal{A}}$ and $R_{n,\mathcal{A}^c}$ in (80) separately in order to prove (81).

*Analysis of $R_{n,\mathcal{A}}$.* Define the following two fixed thresholds:

$$
c_{n,-}^2 := c_{n,\mathrm{fix}}^2 - \varepsilon, \qquad c_{n,+}^2 := c_{n,\mathrm{fix}}^2 + \varepsilon.
\tag{85}
$$

On the event $\mathcal{A}_n$, we have

$$
c_{n,-}^2 \le \widehat{c}_n^2 \le c_{n,+}^2.
\tag{86}
$$

We now explain how (86) yields an upper bound for the loss $L_{n,i}$. Recall from (79) that $L_{n,i} = 1$ can occur only through a type I error or a type II error. Hence:

- if $L_{n,i} = 1$ through a type I error, then we have

$$
\mu_i = 0 \qquad \text{and} \qquad X_i^2 > \widehat{c}_n^2 \ge c_{n,-}^2,
$$

so

$$
\{\mu_i = 0,\ X_i^2 > \widehat{c}_n^2\} \subseteq \{\mu_i = 0,\ X_i^2 > c_{n,-}^2\};
$$

- if $L_{n,i} = 1$ through a type II error, then we have

$$
\mu_i \ne 0 \qquad \text{and} \qquad X_i^2 < \widehat{c}_n^2 \le c_{n,+}^2,
$$

so

$$
\{\mu_i \ne 0,\ X_i^2 < \widehat{c}_n^2\} \subseteq \{\mu_i \ne 0,\ X_i^2 < c_{n,+}^2\}.
$$

Therefore, on $\mathcal{A}_n$,

$$
L_{n,i} \le \mathbf{1}\{\mu_i = 0,\ X_i^2 > c_{n,-}^2\} + \mathbf{1}\{\mu_i \ne 0,\ X_i^2 < c_{n,+}^2\}.
\tag{87}
$$

Multiplying (87) by $\mathbf{1}_{\mathcal{A}_n}$, summing over $i = 1, \ldots, n$, and taking expectations, we obtain

$$
R_{n,\mathcal{A}} = \mathbb{E}\Big[\sum_{i=1}^n L_{n,i}\mathbf{1}_{\mathcal{A}_n}\Big] \le n(1 - p_n)t_{1i}^- + n p_n t_{2i}^+,
\tag{88}
$$

where

$$t_{1i}^- := \mathbb{P}_{H_{0i}}(X_i^2 > c_{n,-}^2), \qquad t_{2i}^+ := \mathbb{P}_{H_{1i}}(X_i^2 < c_{n,+}^2). \tag{89}$$

We now show that the upper bound in the right-hand side of (88) is the asymptotic Bayes risk of the Bayes oracle. To this end, it suffices to verify that the two fixed thresholds in (85) still belong to the class of asymptotically optimal fixed thresholds covered by Theorem F.1. Indeed, since

$$c_{n,\text{fix}}^2 = \log v_n + z_n, \tag{90}$$

we have

$$c_{n,\pm}^2 = \log v_n + (z_n \pm \varepsilon). \tag{91}$$

Because $\varepsilon$ is fixed, the shifted sequences $z_n \pm \varepsilon$ satisfy

$$z_n \pm \varepsilon = o(\log v_n), \qquad (z_n \pm \varepsilon) + 2\log\log v_n \to \infty. \tag{92}$$

Therefore, both $c_{n,-}^2$ and $c_{n,+}^2$ satisfy the assumptions of Theorem F.1.

We now apply the fixed threshold ABOS result to the two terms in (88). Then,

$$n(1 - p_n)t_{1i}^- = o\big(R_{Opt}^{AsyBO}\big) \tag{93}$$

and

$$np_n t_{2i}^+ \le R_{Opt}^{AsyBO}\big(1 + o(1)\big). \tag{94}$$

Substituting (93) and (94) into (88), we obtain

$$R_{n,\mathcal{A}} \le R_{Opt}^{AsyBO}\big(1 + o(1)\big). \tag{95}$$

*Analysis of $R_{n,\mathcal{A}^c}$.* It remains to control the contribution from the complement event $\mathcal{A}_n^c$. We will show that this term is negligible relative to the asymptotic Bayes risk of the Bayes oracle. Since $L_{n,i} \in \{0, 1\}$ for every $i$, we have

$$0 \le \sum_{i=1}^n L_{n,i} \le n. \tag{96}$$

Therefore,

$$R_{n,\mathcal{A}^c} = \mathbb{E}\Big[\sum_{i=1}^n L_{n,i}\mathbf{1}_{\mathcal{A}_n^c}\Big] \le n\,\mathbb{P}(\mathcal{A}_n^c). \tag{97}$$

By the assumption on the random threshold,

$$\mathbb{P}(\mathcal{A}_n^c) = \mathbb{P}\Big(\big|\widehat{c}_n^2 - c_{n,\text{fix}}^2\big| > \varepsilon\Big) = o(p_n). \tag{98}$$

Substituting (98) into (97), we obtain

$$R_{n,\mathcal{A}^c} = o(np_n). \tag{99}$$

On the other hand, the asymptotic Bayes risk of the Bayes oracle is

$$R_{Opt}^{AsyBO} = np_n\big(2\Phi(\sqrt{C}) - 1\big)(1 + o(1)). \tag{100}$$

Since $2\Phi(\sqrt{C}) - 1 > 0$, it follows from (99) and (100) that

$$R_{n,\mathcal{A}^c} = o\big(R_{Opt}^{AsyBO}\big). \tag{101}$$

Finally, combining (80), (95), and (101), we obtain

$$R_n(\widehat{c}_n^2) = R_{n,\mathcal{A}} + R_{n,\mathcal{A}^c} \le R_{Opt}^{AsyBO}(1 + o(1)). \tag{102}$$

This is exactly (81). Together with (82), it follows that (83) holds, and so (84) follows. Thus, the proposed random threshold multiple testing rule achieves ABOS. $\qquad\square$

With the random threshold ABOS criterion proved in Theorem F.2, we now give the formal proof of Theorem 5.8.

We first briefly explain the structure of our proof for Theorem 5.8. We will apply Theorem F.2 with the fixed threshold

$$c_*^2 := 2 \log \frac{1}{p_n} + \log \log n + 2 \log \frac{d_1}{q_*}, \tag{103}$$

where $q_* \in (0, 1)$ is a constant defined later in (112). Therefore, by Theorem F.2, it is enough to prove that for all $\varepsilon > 0$,

$$\mathbb{P}\Big(\big|r(\widehat{\tau}_n)^2 - c_*^2\big| > \varepsilon\Big) = o(p_n). \tag{104}$$

To this end, it suffices to find an event $G_n$ such that, for all sufficiently large $n$,

$$\mathbb{P}\Big(\big|r(\widehat{\tau}_n)^2 - c_*^2\big| > \varepsilon\Big) \overset{(i)}{\leq} \mathbb{P}(G_n^c) \quad \text{and} \quad \mathbb{P}(G_n^c) \overset{(ii)}{=} o(p_n). \tag{105}$$

where

(i) This follows once $G_n$ is chosen so that
$$\big\{\big|r(\widehat{\tau}_n)^2 - c_*^2\big| > \varepsilon\big\} \subseteq G_n^c$$
for all sufficiently large $n$. We will verify this by carrying out the asymptotic analysis on the event $G_n$.

(ii) This follows from a concentration bound for the random variable appearing in the definition of $\widehat{\tau}_n$, which yields
$$\mathbb{P}(G_n^c) = o(p_n).$$

*Proof.* To find $G_n$, we first look at

$$N_n := \sum_{i=1}^n \mathbf{1}\Big(|X_i| > \sqrt{d_2 \log n}\Big). \tag{106}$$

Our task is to identify the order of $\mathbb{E}(N_n)$. Since the $X_i$ are independent and identically distributed,

$$\begin{aligned}
\mathbb{E}(N_n) &= n \, \mathbb{P}\Big(|X_1| > \sqrt{d_2 \log n}\Big) \\
&= \underbrace{n(1 - p_n) \, 2\Big(1 - \Phi(\sqrt{d_2 \log n})\Big)}_{\text{term III}} + \underbrace{np_n \, 2\left(1 - \Phi\left(\frac{\sqrt{d_2 \log n}}{\sqrt{1 + u_n}}\right)\right)}_{\text{term IV}}.
\end{aligned} \tag{107}$$

We examine the two terms separately based on the following analysis. Recall that

$$u_n = \sigma_n^2, \qquad f_n = \frac{1 - p_n}{p_n}.$$

By Assumptions of Theorem 5.8, there exists $\tilde{\zeta} \in (0, \infty)$ such that

$$p_n = \tilde{\zeta} n^{-\beta}, \qquad 0 < \beta < 1. \tag{108}$$

Hence,

$$f_n = \frac{1}{p_n}(1 + o(1)), \qquad \log f_n = \beta \log n - \log \tilde{\zeta} + o(1). \tag{109}$$

Next, Assumption 2.3 states that

$$\frac{\log v_n}{u_n} \to C \in (0, \infty), \qquad \text{where } v_n = u_n f_n^2.$$

Since

$$\log v_n = \log u_n + 2 \log f_n,$$

it follows from (109) that
$$\log v_n = \log u_n + 2\beta \log n + o(\log n).$$

From this relation and $\frac{\log v_n}{u_n} \to C$, we obtain
$$u_n \sim \frac{2\beta}{C} \log n. \tag{110}$$

In particular,
$$\log(1 + u_n) = \log \log n + \log \frac{2\beta}{C} + o(1). \tag{111}$$

We use the above asymptotic analysis in (108)-(111) to analyze the two parts in (107).

- For the term III in (107), since $\sqrt{d_2 \log n} \to \infty$, Mills' ratio gives

$$2\Big(1 - \Phi(\sqrt{d_2 \log n})\Big) \sim 2 \cdot \frac{\varphi(\sqrt{d_2 \log n})}{\sqrt{d_2 \log n}} = 2 \cdot \frac{1}{\sqrt{2\pi}} \frac{\exp\Big(-\frac{(\sqrt{d_2 \log n})^2}{2}\Big)}{\sqrt{d_2 \log n}} = \sqrt{\frac{2}{\pi}} \frac{n^{-d_2/2}}{\sqrt{d_2 \log n}}.$$

  Hence, using (108),
$$\frac{n(1 - p_n)\, 2(1 - \Phi(\sqrt{d_2 \log n}))}{np_n} = O\Big(\frac{n^{\beta - d_2/2}}{\sqrt{\log n}}\Big) \to 0,$$

  because $d_2 \geq 2$ and $0 < \beta < 1$.

- For the term IV in (107), by (110) we have
$$\frac{d_2 \log n}{1 + u_n} \longrightarrow \frac{d_2 C}{2\beta}.$$

  Therefore
$$\frac{\sqrt{d_2 \log n}}{\sqrt{1 + u_n}} \longrightarrow \sqrt{\frac{d_2 C}{2\beta}},$$

  and so
$$2\left(1 - \Phi\left(\frac{\sqrt{d_2 \log n}}{\sqrt{1 + u_n}}\right)\right) \longrightarrow 2\Big(1 - \Phi\Big(\sqrt{\frac{d_2 C}{2\beta}}\Big)\Big).$$

  For convenience, write
$$q_* := 2\Big(1 - \Phi\Big(\sqrt{\frac{d_2 C}{2\beta}}\Big)\Big) \in (0, 1). \tag{112}$$

Combining the term III and term IV, we obtain
$$\mathbb{E}(N_n) \sim q_* np_n. \tag{113}$$

After identifying $\mathbb{E}(N_n)$, we next control the fluctuation of $N_n$ around its mean. Since $N_n$ is a sum of independent Bernoulli random variables, it is binomial with mean $\mathbb{E}(N_n)$. Therefore, for any sequence $\tilde{\delta}_n \in (0, 1)$, the Chernoff bound (see Corollary 2.3.4 of Vershynin (2026)) gives

$$\mathbb{P}\Big(|N_n - \mathbb{E}(N_n)| > \tilde{\delta}_n \mathbb{E}(N_n)\Big) \leq 2\exp\left(-\frac{\tilde{\delta}_n^2}{3}\,\mathbb{E}(N_n)\right). \tag{114}$$

We now choose
$$\tilde{\delta}_n = (\log n)^{-1/4}.$$

This choice tends to zero, so it is strong enough for the asymptotic expansion we need later. At the same time, it is still slow enough for the probability bound in (114) to be negligible on the $p_n$ scale. Indeed, by (113) and (108), we have

$$\mathbb{E}(N_n) \sim q_* np_n \sim q_* \tilde{\zeta}\, n^{1-\beta}.$$

Since $q_* > 0$, $\tilde{\zeta} > 0$, and $0 < \beta < 1$, it follows that

$$\mathbb{E}(N_n) \to \infty.$$

With the choice

$$\tilde{\delta}_n = (\log n)^{-1/4},$$

we have

$$\tilde{\delta}_n^2 \mathbb{E}(N_n) = (\log n)^{-1/2}\mathbb{E}(N_n) \sim q_*\tilde{\zeta}\,\frac{n^{1-\beta}}{\sqrt{\log n}} \to \infty.$$

Applying the Chernoff bound in (114), we obtain

$$\mathbb{P}\Big(|N_n - \mathbb{E}(N_n)| > (\log n)^{-1/4}\mathbb{E}(N_n)\Big) \le 2\exp\Big(-\frac{1}{3}(\log n)^{-1/2}\mathbb{E}(N_n)\Big).$$

It remains to check that the right-hand side is $o(p_n)$. By (108),

$$p_n = \tilde{\zeta}n^{-\beta},$$

and hence we have

$$\log\frac{1}{p_n} = \beta\log n - \log\tilde{\zeta} \sim \beta\log n.$$

Combining this with

$$\tilde{\delta}_n^2 \mathbb{E}(N_n) \sim q_*\tilde{\zeta}\,\frac{n^{1-\beta}}{\sqrt{\log n}},$$

we get

$$\frac{\tilde{\delta}_n^2 \mathbb{E}(N_n)}{\log(1/p_n)} \sim \frac{q_*\tilde{\zeta}\,n^{1-\beta}/\sqrt{\log n}}{\beta\log n} = \frac{q_*\tilde{\zeta}}{\beta}\,\frac{n^{1-\beta}}{(\log n)^{3/2}} \to \infty.$$

Thus,

$$\frac{1}{3}\tilde{\delta}_n^2 \mathbb{E}(N_n) - \log\frac{1}{p_n} \to \infty.$$

Equivalently,

$$\exp\Big(-\frac{\tilde{\delta}_n^2}{3}\mathbb{E}(N_n)\Big) = o(p_n).$$

Consequently,

$$\mathbb{P}\Big(|N_n - \mathbb{E}(N_n)| > (\log n)^{-1/4}\mathbb{E}(N_n)\Big) = o(p_n).$$

Thus, we can define the event $G_n$ by

$$G_n := \Big\{|N_n - \mathbb{E}(N_n)| \le (\log n)^{-1/4}\mathbb{E}(N_n)\Big\}. \tag{115}$$

Then, as desired for $(ii)$ in (105), we have

$$\mathbb{P}(G_n^c) = o(p_n). \tag{116}$$

Now, it suffices to show that, for all sufficiently large $n$,

$$\big\{|r(\widehat{\tau}_n)^2 - c_*^2| > \varepsilon\big\} \subseteq G_n^c. \tag{117}$$

This will establish $(i)$ in (105). We first analyze $r(\widehat{\tau}_n)^2$ on the event $G_n$. Recall that

$$\widehat{\tau}_n = \min\Big\{n\sqrt{\log n}, \frac{d_1 n\sqrt{\log n}}{N_n}\Big\}. \tag{118}$$

On $G_n$, by definition and by (113),

$$N_n = \mathbb{E}(N_n)(1 + o(1)) = q_* \, np_n(1 + o(1)).$$

Thus,

$$N_n = q_* \, np_n(1 + o(1)) \qquad \text{on } G_n. \tag{119}$$

We now use this to simplify the minimum in (118). Since $q_* > 0$ and $np_n \to \infty$, it follows from (119) that, for all sufficiently large $n$,

$$N_n \geq \frac{q_*}{2} \, np_n > d_1 \qquad \text{on } G_n.$$

Therefore, on $G_n$, the second term in the minimum is the active one. In other words,

$$\widehat{\tau}_n = \frac{d_1 n \sqrt{\log n}}{N_n} \qquad \text{on } G_n$$

for all sufficiently large $n$. Substituting (119) gives

$$\widehat{\tau}_n = \frac{d_1}{q_*} \, p_n^{-1} \sqrt{\log n} \, (1 + o(1)) \qquad \text{on } G_n. \tag{120}$$

It now remains to pass from $\widehat{\tau}_n$ to the threshold $r(\widehat{\tau}_n)^2$. Since $\widehat{\tau}_n \to \infty$ on $G_n$, we may use the elementary asymptotic relation

$$r(\tau)^2 = 2\log\tau + o(1) \qquad \text{as } \tau \to \infty.$$

Applying this to (120), we obtain

$$r(\widehat{\tau}_n)^2 = 2\log\frac{1}{p_n} + \log\log n + 2\log\frac{d_1}{q_*} + o(1) \qquad \text{on } G_n. \tag{121}$$

We next compare this with

$$c_*^2 = 2\log\frac{1}{p_n} + \log\log n + 2\log\frac{d_1}{q_*}.$$

By (121), we immediately get

$$r(\widehat{\tau}_n)^2 - c_*^2 = o(1) \qquad \text{on } G_n. \tag{122}$$

Hence, for all $\varepsilon > 0$,

$$|r(\widehat{\tau}_n)^2 - c_*^2| < \varepsilon \qquad \text{on } G_n$$

for all sufficiently large $n$. Equivalently,

$$\left\{ |r(\widehat{\tau}_n)^2 - c_*^2| > \varepsilon \right\} \subseteq G_n^c \qquad \text{for all sufficiently large } n.$$

This is precisely (117).

It remains to verify that the fixed threshold $c_*^2$ satisfies the assumptions of Theorem F.2. By (110),

$$\log u_n = \log\log n + \log\frac{2\beta}{C} + o(1).$$

Combining this with (109), we obtain

$$\log v_n = \log u_n + 2\log f_n = 2\log\frac{1}{p_n} + \log\log n + \log\frac{2\beta}{C} + o(1). \tag{123}$$

Therefore,

$$c_*^2 = \log v_n + z_{n,*}, \qquad z_{n,*} = 2\log\frac{d_1}{q_*} - \log\frac{2\beta}{C} + o(1). \tag{124}$$

Since $\log v_n \to \infty$, we have

$$z_{n,*} = o(\log v_n).$$

Moreover, $z_{n,*}$ converges to a finite constant. Since $\log v_n \to \infty$, it follows that

$$z_{n,*} + 2\log\log v_n \to \infty.$$

Therefore, the fixed sequence $c_*^2$ satisfies the conditions of Theorem F.2.

Finally, by (117) and (116), for all $\varepsilon > 0$ we have

$$\mathbb{P}\Big(|r(\widehat{\tau}_n)^2 - c_*^2| > \varepsilon\Big) \le \mathbb{P}(G_n^c) = o(p_n).$$

This is exactly (104). We then can invoke Theorem F.2. Hence, the empirical Bayes relative belief multiple testing rule that rejects $H_{0i}$ whenever

$$X_i^2 > r(\widehat{\tau}_n)^2, \qquad i = 1, \ldots, n,$$

achieves ABOS. In other words,

$$\frac{R_{EB}^{AsyRB}}{R_{Opt}^{AsyBO}} \to 1.$$

$\square$

# G. Proofs for ABOS via Relative Belief in the Multivariate Normal Means Problem

In this appendix, we provide the detailed proofs for the theoretical results established in Section 6 regarding the multivariate relative belief multiple testing procedure.

## G.1. Proof of Proposition 6.5

*Proof.* The relative belief ratio at the value of interest $\boldsymbol{\mu}_i = \mathbf{0}$ is given by

$$\mathrm{RB}(\boldsymbol{\mu}_i = \mathbf{0}|\boldsymbol{X}_i, \tau) = \frac{\pi(\boldsymbol{\mu}_i = \mathbf{0}|\boldsymbol{X}_i, \tau)}{\pi(\boldsymbol{\mu}_i = \mathbf{0})}.$$

Using the Savage-Dickey ratio result (Dickey, 1971), we have

$$\mathrm{RB}(\boldsymbol{\mu}_i = \mathbf{0}|\boldsymbol{X}_i, \tau) = \frac{m(\boldsymbol{X}_i|\boldsymbol{\mu}_i = \mathbf{0})}{m(\boldsymbol{X}_i|\tau)}. \tag{125}$$

The numerator of (125) is the conditional prior predictive density $m(\boldsymbol{X}_i|\boldsymbol{\mu}_i = \mathbf{0})$, which corresponds to the density of $N_k(\mathbf{0}, \boldsymbol{\Sigma})$. Specifically, its probability density function (pdf) is

$$m(\boldsymbol{X}_i|\boldsymbol{\mu}_i = \mathbf{0}) = (2\pi)^{-k/2}|\boldsymbol{\Sigma}|^{-1/2}\exp\left(-\frac{1}{2}\boldsymbol{X}_i^\top\boldsymbol{\Sigma}^{-1}\boldsymbol{X}_i\right).$$

The denominator of (125) is the prior predictive density $m(\boldsymbol{X}_i|\tau)$, obtained by integrating over $\boldsymbol{\mu}_i$. The hierarchical model specifies that $\boldsymbol{X}_i|\boldsymbol{\mu}_i \sim N_k(\boldsymbol{\mu}_i, \boldsymbol{\Sigma})$ and $\boldsymbol{\mu}_i \sim N_k(\mathbf{0}, \tau\boldsymbol{\Sigma})$. The marginal distribution of $\boldsymbol{X}_i$ is the convolution of these two normal distributions, which results in $\boldsymbol{X}_i \sim N_k(\mathbf{0}, \boldsymbol{\Sigma} + \tau\boldsymbol{\Sigma}) = N_k(\mathbf{0}, (1+\tau)\boldsymbol{\Sigma})$. Thus, the prior predictive density $m(\boldsymbol{X}_i|\tau)$ is

$$m(\boldsymbol{X}_i|\tau) = (2\pi)^{-k/2}|(1+\tau)\boldsymbol{\Sigma}|^{-1/2}\exp\left(-\frac{1}{2}\boldsymbol{X}_i^\top((1+\tau)\boldsymbol{\Sigma})^{-1}\boldsymbol{X}_i\right).$$

Substituting these pdfs into (125) yields

$$\begin{aligned}
&\mathrm{RB}(\boldsymbol{\mu}_i = \mathbf{0}|\boldsymbol{X}_i, \tau)\\
&= \frac{(2\pi)^{-k/2}|\boldsymbol{\Sigma}|^{-1/2}\exp\left(-\frac{1}{2}\boldsymbol{X}_i^\top\boldsymbol{\Sigma}^{-1}\boldsymbol{X}_i\right)}{(2\pi)^{-k/2}|(1+\tau)\boldsymbol{\Sigma}|^{-1/2}\exp\left(-\frac{1}{2}\boldsymbol{X}_i^\top((1+\tau)\boldsymbol{\Sigma})^{-1}\boldsymbol{X}_i\right)}\\
&= \left(\frac{|(1+\tau)\boldsymbol{\Sigma}|}{|\boldsymbol{\Sigma}|}\right)^{1/2}\exp\left(-\frac{1}{2}\boldsymbol{X}_i^\top\left[\boldsymbol{\Sigma}^{-1} - ((1+\tau)\boldsymbol{\Sigma})^{-1}\right]\boldsymbol{X}_i\right).
\end{aligned}$$

The determinant term simplifies using the property $|\tilde{c}A| = \tilde{c}^k|A|$,

$$\left(\frac{(1+\tau)^k|\boldsymbol{\Sigma}|}{|\boldsymbol{\Sigma}|}\right)^{1/2} = ((1+\tau)^k)^{1/2} = (1+\tau)^{k/2}.$$

The term $\boldsymbol{\Sigma}^{-1} - ((1+\tau)\boldsymbol{\Sigma})^{-1}$ in the exponent can be simplified to

$$\boldsymbol{\Sigma}^{-1} - ((1+\tau)\boldsymbol{\Sigma})^{-1} = \boldsymbol{\Sigma}^{-1} - \frac{1}{1+\tau}\boldsymbol{\Sigma}^{-1} = \left(1 - \frac{1}{1+\tau}\right)\boldsymbol{\Sigma}^{-1} = \frac{\tau}{1+\tau}\boldsymbol{\Sigma}^{-1}.$$

Thus, we have

$$\mathrm{RB}(\boldsymbol{\mu}_i = \mathbf{0}|\boldsymbol{X}_i, \tau) = (1+\tau)^{k/2} \exp\left(-\frac{\tau}{2(1+\tau)}\boldsymbol{X}_i^\top\boldsymbol{\Sigma}^{-1}\boldsymbol{X}_i\right).$$

$\square$

## G.2. Proof of Proposition 6.6

*Proof.* The proof begins with the rejection criterion $\mathrm{RB}(\boldsymbol{\mu}_i = \mathbf{0}|\boldsymbol{X}_i, \tau) < 1$ and proceeds by taking the natural logarithm of both sides of the inequality:

$$(1+\tau)^{k/2} \exp\left(-\frac{\tau}{2(1+\tau)}\boldsymbol{X}_i^\top\boldsymbol{\Sigma}^{-1}\boldsymbol{X}_i\right) < 1$$
$$\implies \log\left((1+\tau)^{k/2}\right) - \frac{\tau}{2(1+\tau)}\boldsymbol{X}_i^\top\boldsymbol{\Sigma}^{-1}\boldsymbol{X}_i < \log(1)$$
$$\implies \frac{k}{2}\log(1+\tau) < \frac{\tau}{2(1+\tau)}\boldsymbol{X}_i^\top\boldsymbol{\Sigma}^{-1}\boldsymbol{X}_i$$
$$\implies \boldsymbol{X}_i^\top\boldsymbol{\Sigma}^{-1}\boldsymbol{X}_i > k\frac{1+\tau}{\tau}\log(1+\tau).$$

$\square$

## G.3. Proof of Proposition 6.7

*Proof.* The proof proceeds by standardizing the random vector $\boldsymbol{X}_i$ under each case to derive the distribution of the quadratic form.

**(a) Distribution under $H_{0i}$.** Under the null hypothesis, the data vector is distributed as $\boldsymbol{X}_i \sim N_k(\mathbf{0}, \boldsymbol{\Sigma})$. We define a standardized vector $\boldsymbol{Z}_i = \boldsymbol{\Sigma}^{-1/2}\boldsymbol{X}_i$. The mean and covariance of $\boldsymbol{Z}_i$ are

$$\mathbb{E}[\boldsymbol{Z}_i] = \boldsymbol{\Sigma}^{-1/2}\mathbb{E}[\boldsymbol{X}_i] = \boldsymbol{\Sigma}^{-1/2}\mathbf{0} = \mathbf{0},$$
$$\mathrm{Cov}(\boldsymbol{Z}_i) = \boldsymbol{\Sigma}^{-1/2}\mathrm{Cov}(\boldsymbol{X}_i)(\boldsymbol{\Sigma}^{-1/2})^\top = \boldsymbol{\Sigma}^{-1/2}\boldsymbol{\Sigma}\boldsymbol{\Sigma}^{-1/2} = \boldsymbol{I}.$$

Thus, $\boldsymbol{Z}_i \sim N_k(\mathbf{0}, \boldsymbol{I})$. We now express the quadratic form in terms of $\boldsymbol{Z}_i$ by substituting $\boldsymbol{X}_i = \boldsymbol{\Sigma}^{1/2}\boldsymbol{Z}_i$:

$$\boldsymbol{X}_i^\top\boldsymbol{\Sigma}^{-1}\boldsymbol{X}_i = (\boldsymbol{\Sigma}^{1/2}\boldsymbol{Z}_i)^\top\boldsymbol{\Sigma}^{-1}(\boldsymbol{\Sigma}^{1/2}\boldsymbol{Z}_i) = \boldsymbol{Z}_i^\top\boldsymbol{I}\boldsymbol{Z}_i = \boldsymbol{Z}_i^\top\boldsymbol{Z}_i.$$

By definition, the sum of squares of $k$ independent standard normal random variables $\boldsymbol{Z}_i^\top\boldsymbol{Z}_i$ follows a chi-squared distribution with $k$ degrees of freedom.

**(b) Distribution under $H_{1i}$.** Under the alternative hypothesis, the data vector is distributed as $\boldsymbol{X}_i \sim N_k(\mathbf{0}, (1+g)\boldsymbol{\Sigma})$. We define a different standardized vector $\boldsymbol{Y}_i = ((1+g)\boldsymbol{\Sigma})^{-1/2}\boldsymbol{X}_i$. Following similar logic as in part (a), $\boldsymbol{Y}_i$ is standard normal, $\boldsymbol{Y}_i \sim N_k(\mathbf{0}, \boldsymbol{I})$.

We express $\boldsymbol{X}_i$ in terms of $\boldsymbol{Y}_i$ as $\boldsymbol{X}_i = \sqrt{1+g} \cdot \boldsymbol{\Sigma}^{1/2}\boldsymbol{Y}_i$. Substituting this into the quadratic form:

$$\boldsymbol{X}_i^\top\boldsymbol{\Sigma}^{-1}\boldsymbol{X}_i = (\sqrt{1+g} \cdot \boldsymbol{\Sigma}^{1/2}\boldsymbol{Y}_i)^\top\boldsymbol{\Sigma}^{-1}(\sqrt{1+g} \cdot \boldsymbol{\Sigma}^{1/2}\boldsymbol{Y}_i)$$
$$= (1+g) \cdot \boldsymbol{Y}_i^\top\boldsymbol{\Sigma}^{1/2}\boldsymbol{\Sigma}^{-1}\boldsymbol{\Sigma}^{1/2}\boldsymbol{Y}_i$$
$$= (1+g) \cdot \boldsymbol{Y}_i^\top\boldsymbol{I}\boldsymbol{Y}_i = (1+g)\boldsymbol{Y}_i^\top\boldsymbol{Y}_i.$$

Since $\boldsymbol{Y}_i^\top \boldsymbol{Y}_i$ follows a $\chi_k^2$ distribution, the quadratic form under $H_{1i}$ is distributed as a $(1+g)\chi_k^2$ distribution.

With these distributional facts, it is trivial to obtain type I and type II error probabilities for the relative belief procedure as stated in Proposition 6.7. $\qquad\square$

### G.4. Proof of Proposition 6.8

*Proof.* By Proposition 6.6, the relative belief multiple testing procedure rejects the null hypothesis $H_{0i}$ when

$$\boldsymbol{X}_i^\top \boldsymbol{\Sigma}^{-1} \boldsymbol{X}_i > r_k(\tau), \quad \text{where } r_k(\tau) = k\,\frac{1+\tau}{\tau}\,\log(1+\tau). \tag{126}$$

Write

$$Q(\tau) := \frac{r_k(\tau)}{k} = \frac{1+\tau}{\tau}\,\log(1+\tau) \in (1,\infty). \tag{127}$$

Using the expressions of type I and type II error probabilities for the relative belief procedure in Proposition 6.7, the associated Bayes risk can be expressed as

$$R^{mRB}(\tau) = n(1-p)\Big[1 - F_{\chi_k^2}\big(kQ(\tau)\big)\Big] + np\,F_{\chi_k^2}\Big(\tfrac{kQ(\tau)}{1+g}\Big). \tag{128}$$

By the chain rule, we have

$$\frac{dR^{mRB}}{d\tau} = \frac{dR^{mRB}}{dQ}\cdot\frac{dQ}{d\tau}, \quad \text{and } \frac{dQ}{d\tau} = \frac{1}{\tau^2}\big\{\tau - \log(1+\tau)\big\} > 0, \tag{129}$$

for $\tau > 0$. So stationary point(s) can be obtained by solving $\frac{dR^{mRB}}{dQ} = 0$ for $Q \in (1,\infty)$.

Next we compute the first derivative in $Q$ and factor it in a convenient form. Let $f_{\chi_k^2}$ denote the $\chi_k^2$ density. Differentiating (128) with respect to $Q$ gives

$$\frac{dR^{mRB}}{dQ} = -n(1-p)\,k\,f_{\chi_k^2}(kQ) + np\,\frac{k}{1+g}\,f_{\chi_k^2}\Big(\tfrac{kQ}{1+g}\Big). \tag{130}$$

Solving for $\frac{dR^{mRB}}{dQ} = 0$ gives us the only stationary point

$$Q_* = \frac{1+g}{g}\Big\{\tfrac{2}{k}\log\tfrac{1-p}{p} + \log(1+g)\Big\}. \tag{131}$$

Since $Q \in (1,\infty)$ by (127), the equality admits a feasible and meaningful solution precisely when

$$\frac{1+g}{g}\Big\{\tfrac{2}{k}\log\tfrac{1-p}{p} + \log(1+g)\Big\} > 1. \tag{132}$$

In this regime (132), we show (131) is indeed a unique global minimizer in the sequel. Factoring $nk\,f_{\chi_k^2}(kQ) > 0$ from (130) yields

$$\frac{dR^{mRB}}{dQ} = nk\,f_{\chi_k^2}(kQ)\,\widetilde{h}(Q), \tag{133}$$

where

$$\widetilde{h}(Q) := -(1-p) + \frac{p}{1+g}\,\frac{f_{\chi_k^2}\big(\tfrac{kQ}{1+g}\big)}{f_{\chi_k^2}(kQ)}.$$

Using $f_{\chi_k^2}(x) = \frac{1}{2^{k/2}\Gamma(k/2)}x^{\frac{k}{2}-1}e^{-x/2}$, the density ratio simplifies to

$$\frac{f_{\chi_k^2}\big(\tfrac{kQ}{1+g}\big)}{f_{\chi_k^2}(kQ)} = \Big(\tfrac{1}{1+g}\Big)^{\frac{k}{2}-1}\exp\Big\{\tfrac{kg}{2(1+g)}\,Q\Big\}, \tag{134}$$

and hence

$$\widetilde{h}(Q) = -(1-p) + p\,(1+g)^{-\frac{k}{2}}\,\exp\Big\{\tfrac{kg}{2(1+g)}\,Q\Big\}. \tag{135}$$

Therefore, the sign of $\frac{dR^{mRB}}{dQ}$ would be the same as the sign of $\widetilde{h}(Q)$.

From (135), we have

$$\frac{d\widetilde{h}}{dQ}(Q) = p\,(1+g)^{-\frac{k}{2}}\,\frac{kg}{2(1+g)}\,\exp\Big\{\tfrac{kg}{2(1+g)}\,Q\Big\} > 0, \tag{136}$$

so $\widetilde{h}$ is strictly increasing on $(1,\infty)$. Considering $Q \downarrow 1$ (when $\tau \downarrow 0$), we have

$$\lim_{Q \downarrow 1} \widetilde{h}(Q) = -(1-p) + p\,(1+g)^{-\frac{k}{2}}\,\exp\Big\{\tfrac{kg}{2(1+g)}\Big\} < 0, \tag{137}$$

based on $\frac{1+g}{g}\Big\{\frac{2}{k}\log\frac{1-p}{p} + \log(1+g)\Big\} > 1$ in (132).

Moreover,

$$\lim_{Q \to \infty} \widetilde{h}(Q) = +\infty. \tag{138}$$

By continuity and the strict monotonicity in (136), the $Q_* \in (1,\infty)$ in (131) is unique with

$$\widetilde{h}(Q_*) = 0. \tag{139}$$

Now, we first establish $Q_*$ is a strict local minimizer of $R^{mRB}(Q)$. Differentiating (133) once more and using $\frac{d}{dQ}f_{\chi_k^2}(kQ) = k\,f'_{\chi_k^2}(kQ)$, then

$$\frac{d^2 R^{mRB}}{dQ^2}(Q) = nk\Big\{k\,f'_{\chi_k^2}(kQ)\,\widetilde{h}(Q) + f_{\chi_k^2}(kQ)\,\widetilde{h}'(Q)\Big\}.$$

At $Q_*$, we have $\widetilde{h}(Q_*) = 0$, so

$$\frac{d^2 R^{mRB}}{dQ^2}\Big|_{Q_*} = nk\,f_{\chi_k^2}(kQ_*)\,\widetilde{h}'(Q_*) > 0, \tag{140}$$

by (136). Combining (129) and (140),

$$\frac{d^2 R^{mRB}}{d\tau^2}\Big|_{\tau^*} = \frac{d^2 R^{mRB}}{dQ^2}\Big|_{Q_*}\left(\frac{dQ}{d\tau}\Big|_{\tau^*}\right)^2 > 0,$$

so the corresponding $\tau^*$ is the only strict local minimizer.

Observe from (133) that $\frac{dR^{mRB}}{dQ} = nk\,f_{\chi_k^2}(kQ)\,\widetilde{h}(Q)$ where $nk\,f_{\chi_k^2}(kQ) > 0$. Thus, to determine the sign of $\frac{dR^{mRB}}{dQ}$, we only need to focus on the sign of $\widetilde{h}(Q)$. By (137), (138), (139), and the fact $\widetilde{h}$ is strictly increasing on $(1,\infty)$, we know that $\frac{dR^{mRB}}{dQ} < 0$ on $(1, Q_*)$ and $\frac{dR^{mRB}}{dQ} > 0$ on $(Q_*, \infty)$. Therefore, $Q_*$ is the unique global minimizer of $R^{mRB}(Q)$ over $Q \in (1,\infty)$.

Since $r_k(\tau) = kQ(\tau)$ in (126) is strictly increasing on its feasible domain, it is invertible. Therefore, the corresponding unique global minimizer $\tau^*$ for $R^{mRB}(\tau)$ is

$$\tau^* = r_k^{-1}(kQ_*) = r_k^{-1}\left(\frac{1+g}{g}\Big\{2\log\frac{1-p}{p} + k\log(1+g)\Big\}\right).$$

$\square$

### G.5. Proof of Lemma 6.10

*Proof.* Recall $r_k(\tau) = k\frac{1+\tau}{\tau}\log(1+\tau)$. By definition in (15), $\tau_n^*$ satisfies $r_k(\tau_n^*) = w_n^*$ under Assumption 6.3, where $w_n^*$ is

$$w_n^* = \frac{1+g_n}{g_n}\left[2\log\left(\frac{1-p_n}{p_n}\right) + k\log(1+g_n)\right].$$

We can first rewrite $w_n^*$ as

$$
\begin{aligned}
w_n^* &= \left(1 + \frac{1}{g_n}\right)\left[2\log\left(\frac{1}{p_n}\right) + 2\log(1-p_n) + k\log(1+g_n)\right] \\
&= 2\log\left(\frac{1}{p_n}\right) + 2\log(1-p_n) + k\log g_n + k\log\left(1 + \frac{1}{g_n}\right) + \frac{1}{g_n}\left[2\log\left(\frac{1}{p_n}\right) + 2\log(1-p_n) + k\log(1+g_n)\right] \\
&= 2\log\left(\frac{1}{p_n}\right) + k\log g_n + \frac{2}{g_n}\log\left(\frac{1}{p_n}\right) + o(1).
\end{aligned}
\tag{141}
$$

From Assumption 6.3,

$$\frac{-2\log p_n}{g_n} \to C_0 \in (0,\infty).$$

Hence,

$$g_n = \frac{2\log(1/p_n)}{C_0}(1+o(1)), \qquad \frac{1}{g_n} = \frac{C_0}{2\log(1/p_n)}(1+o(1)). \tag{142}$$

Moreover,

$$\log g_n = \log\left(\frac{2\log(1/p_n)}{C_0}(1+o(1))\right) = \log\left(\frac{2}{C_0}\right) + \log\log\left(\frac{1}{p_n}\right) + o(1). \tag{143}$$

Thus, combining (142) and (143), we have

$$
\begin{aligned}
w_n^* &= 2\log\left(\frac{1}{p_n}\right) + k\log g_n + \frac{2}{g_n}\log\left(\frac{1}{p_n}\right) + o(1) \\
&= 2\log\left(\frac{1}{p_n}\right) + k\log\log\left(\frac{1}{p_n}\right) + k\log\left(\frac{2}{C_0}\right) + C_0 + o(1).
\end{aligned}
$$

It is clear that $w_n^* \to \infty$ as $n \to \infty$. The function $r_k(\tau) = k\frac{1+\tau}{\tau}\log(1+\tau)$ is continuous and strictly increasing for all $\tau \in (0,\infty)$. Since $w_n^* = r_k(\tau_n^*)$ and $r_k(\tau_n^*)$ is a strictly increasing function which approaches infinity only as its argument does, it follows that $\tau_n^* \to \infty$ as $n \to \infty$.

The asymptotic behavior of $r_k(\tau_n^*)$ is as follows.

$$
\begin{aligned}
r_k(\tau_n^*) &= k\left(1 + \frac{1}{\tau_n^*}\right)\log\left(1+\tau_n^*\right) \\
&= k\log\left(\tau_n^*\right) + k\log\left(1 + \frac{1}{\tau_n^*}\right) + k\frac{\log\left(1+\tau_n^*\right)}{\tau_n^*} \\
&= k\log\left(\tau_n^*\right) + o(1).
\end{aligned}
\tag{144}
$$

Equating the asymptotic expressions for $r_k(\tau_n^*)$ and $w_n^*$ (i.e., (141) and (144)) gives

$$k\log\tau_n^* + o(1) = 2\log\left(\frac{1}{p_n}\right) + k\log\log\left(\frac{1}{p_n}\right) + k\log\left(\frac{2}{C_0}\right) + C_0 + o(1).$$

Equivalently, we can write

$$\log(\tau_n^*) = \frac{2}{k}\log\left(\frac{1}{p_n}\right) + \log\log\left(\frac{1}{p_n}\right) + \log\left(\frac{2}{C_0}\right) + \frac{C_0}{k} + o(1).$$

Exponentiating both sides above, we find the asymptotic behavior of $\tau_n^*$

$$
\begin{aligned}
\tau_n^* &= \exp\left(\frac{2}{k}\log\left(\frac{1}{p_n}\right) + \log\log\left(\frac{1}{p_n}\right) + \log\left(\frac{2}{C_0}\right) + \frac{C_0}{k} + o(1)\right) \\
&= \exp\left(\frac{2}{k}\log(1/p_n)\right)\exp\left(\log\log(1/p_n)\right)\exp\left(\log(2/C_0)\right)\exp\left(\frac{C_0}{k}\right)\exp\left(o(1)\right) \\
&= p_n^{-2/k}\log\left(\frac{1}{p_n}\right)\frac{2}{C_0}\exp\left(\frac{C_0}{k}\right)\left(1 + o(1)\right).
\end{aligned}
$$

Thus, we can conclude that

$$
\lim_{n\to\infty}\frac{\tau_n^*}{p_n^{-2/k}\log(1/p_n)} = \frac{2}{C_0}\exp\left(\frac{C_0}{k}\right) \in (0, \infty).
$$

$\square$

### G.6. Proof of Theorem 6.11

Our proof of Theorem 6.11 relies on the following Lemma G.1 for the upper tail of the chi-squared distribution. This bound is a standard result which can be derived directly from the asymptotic expansion of the incomplete gamma function (see Proposition 6.5.32 of Abramowitz & Stegun (1965) for details).

**Lemma G.1.** *Let $\widetilde{X} \sim \chi_k^2$ with $k \geq 2$. Then there exist constants $x_0 = x_0(k) > 0$ and $C_k > 0$ in which both constants depend on $k$, such that, for all $x \geq x_0$,*

$$
\mathbb{P}(\widetilde{X} > x) \leq C_k x^{\frac{k}{2}-1}e^{-x/2}.
$$

Now, the proof of Theorem 6.11 is shown as follows.

*Proof.* Suppose Assumption 6.3 holds. The assumption on the tuning parameter $\tau_n$ implies the existence of a constant $\gamma_1 \in (0, \infty)$ such that

$$
\tau_n = \gamma_1 \cdot p_n^{-2/k}\log(1/p_n)(1 + o(1)). \tag{145}
$$

Let $r_k(\tau_n) = k\frac{1+\tau_n}{\tau_n}\log(1+\tau_n)$. As $\tau_n \to \infty$, we have

$$
\begin{aligned}
r_k(\tau_n) &= k\log\tau_n + o(1) \\
&= k\log\left(\gamma_1 \cdot p_n^{-2/k}\log(1/p_n)(1 + o(1))\right) + o(1) \\
&= k\left[\log\gamma_1 + \frac{2}{k}\log(p_n^{-1}) + \log(\log(p_n^{-1})) + o(1)\right] + o(1) \\
&= 2\log(p_n^{-1}) + k\log(\log(p_n^{-1})) + k\log\gamma_1 + o(1). \tag{146}
\end{aligned}
$$

Recall $\widetilde{X} \sim \chi_k^2$ with $k \geq 2$, we first analyze the type I error probability

$$
t_{1i}^{AsymRB} = \mathbb{P}(\widetilde{X} > r_k(\tau_n)).
$$

By Lemma G.1, there exist constants $x_0 = x_0(k) > 0$ and $C_k > 0$ such that, for all $x \geq x_0$,

$$
\mathbb{P}(\widetilde{X} > x) \leq C_k x^{\frac{k}{2}-1}e^{-x/2}.
$$

Notice that $r_k(\tau_n) \to \infty$ by observing (146). We thus have $r_k(\tau_n) \geq x_0$ for all sufficiently large $n$, and then

$$
t_{1i}^{AsymRB} = \mathbb{P}(\widetilde{X} > r_k(\tau_n)) \leq C_k\left[r_k(\tau_n)\right]^{\frac{k}{2}-1}e^{-r_k(\tau_n)/2}. \tag{147}
$$

From (146),

$$
-\frac{r_k(\tau_n)}{2} = -\log(p_n^{-1}) - \frac{k}{2}\log\left(\log(p_n^{-1})\right) - \frac{k}{2}\log\gamma_1 + o(1),
$$

so

$$e^{-r_k(\tau_n)/2} = \exp\left(-\log(p_n^{-1})\right) \exp\left(-\tfrac{k}{2}\log\left(\log(p_n^{-1})\right)\right) \exp\left(-\tfrac{k}{2}\log\gamma_1\right) e^{o(1)}$$

$$= p_n \left(\log(p_n^{-1})\right)^{-k/2} \gamma_1^{-k/2} (1+o(1)). \tag{148}$$

Moreover, from (146), we also have

$$\frac{r_k(\tau_n)}{2\log(p_n^{-1})} = 1 + \frac{k\log(\log(p_n^{-1})) + k\log\gamma_1 + o(1)}{2\log(p_n^{-1})} \to 1,$$

because $\log(\log(p_n^{-1}))/\log(p_n^{-1}) \to 0$ as $p_n \to 0$. Hence

$$r_k(\tau_n) = 2\log(p_n^{-1})(1+o(1)).$$

Therefore,

$$[r_k(\tau_n)]^{\frac{k}{2}-1} = \left(2\log(p_n^{-1})(1+o(1))\right)^{\frac{k}{2}-1} = 2^{\frac{k}{2}-1}\left(\log(p_n^{-1})\right)^{\frac{k}{2}-1}(1+o(1)). \tag{149}$$

Combining (147), (148) and (149), we obtain

$$0 \le t_{1i}^{AsymRB} \le C_k \cdot 2^{\frac{k}{2}-1}\left(\log(p_n^{-1})\right)^{\frac{k}{2}-1} p_n \left(\log(p_n^{-1})\right)^{-k/2} \gamma_1^{-k/2}(1+o(1))$$

Equivalently, we have

$$0 \le t_{1i}^{AsymRB} \le \gamma_2\, p_n \left(\log(p_n^{-1})\right)^{-1}(1+o(1)),$$

where

$$\gamma_2 := C_k\, 2^{\frac{k}{2}-1}\gamma_1^{-k/2} \in (0,\infty).$$

Thus,

$$0 \le \frac{t_{1i}^{AsymRB}}{p_n} \le \gamma_2 \left(\log(p_n^{-1})\right)^{-1}(1+o(1)).$$

As $n \to \infty$, we have $p_n \to 0$, and hence $\log(p_n^{-1}) \to \infty$. Therefore, $\gamma_2 \left(\log(p_n^{-1})\right)^{-1}(1+o(1))$ converges to 0. This shows that

$$\lim_{n\to\infty} \frac{t_{1i}^{AsymRB}}{p_n} = 0, \qquad \text{i.e.,} \quad t_{1i}^{AsymRB} = o(p_n). \tag{150}$$

We then analyze the asymptotic type II error probability,

$$t_{2i}^{AsymRB} = F_{\chi_k^2}\left(\frac{r_k(\tau_n)}{1+g_n}\right),$$

where $F_{\chi_k^2}$ is the cdf of the chi-squared distribution with $k$ degrees of freedom.

Considering $\frac{r_k(\tau_n)}{1+g_n}$, we substitute the asymptotic form of $r_k(\tau_n)$ from (146) and the form of $g_n$ from Assumption 6.3

$$\lim_{n\to\infty} \frac{r_k(\tau_n)}{1+g_n} = \lim_{n\to\infty} \frac{2\log(p_n^{-1}) + k\log(\log(p_n^{-1})) + k\log\gamma_1 + o(1)}{1 + \frac{2\log(p_n^{-1})}{C_0}(1+o(1))}$$

$$= \lim_{n\to\infty} \frac{1 + \frac{k\log(\log(p_n^{-1}))}{2\log(p_n^{-1})} + \frac{k\log\gamma_1 + o(1)}{2\log(p_n^{-1})}}{\frac{1}{2\log(p_n^{-1})} + \frac{1}{C_0}(1+o(1))}$$

$$= \frac{1+0+0}{0+1/C_0} = C_0.$$

By the continuity of the $F_{\chi_k^2}$ and $0 < F_{\chi_k^2}(C_0) < 1$, it follows that

$$t_{2i}^{AsymRB} = F_{\chi_k^2}(C_0)(1+o(1)). \tag{151}$$

Finally, combining (150) and (151), we obtain the asymptotic Bayes risk

$$
\begin{aligned}
R^{AsymRB} &= n(1 - p_n)t_{1i}^{AsymRB} + np_n t_{2i}^{AsymRB} \\
&= n(1 - p_n) \cdot o(p_n) + np_n \left( F_{\chi_k^2}(C_0) + o(1) \right) \\
&= np_n F_{\chi_k^2}(C_0) + o(np_n) \\
&= np_n F_{\chi_k^2}(C_0)(1 + o(1)),
\end{aligned}
$$

which shows the relative belief multiple testing procedure achieves ABOS by recalling Definition 6.4. $\qquad\square$

