# OpenReview forum: "An Evidential Route to Asymptotic Bayes Optimality under Sparsity"
_ICML.cc/2026/Conference — ICML 2026 regular_

### Official Review · Reviewer_B6fF · 2026-03-08

**Soundness:** 3
**Presentation:** 3
**Significance:** 2
**Originality:** 3
**Overall Recommendation:** 5
**Confidence:** 3

**Summary:**

This paper proposes to use an alternative Bayesian multiple testing framework called relative belief ratio, which relies on whether posterior density at the null hypotheses is greater than the prior density. The paper's main contribution is to show the Bayes risk under one group Gaussian priors (a very simple and common prior) achieves the asymptotically Bayes optimal under sparsity (ABOS). In addition, the paper also extends this conclusion from the univariate normal mean problems to the multivariate normal case.

**Compliance With Llm Reviewing Policy:**

Affirmed.

**Final Justification:**

This is a solid paper, especially in terms of soundness and originality. The rebuttal addressed my main concerns. In particular, I am glad to see the simulation study provided in the rebuttal.

**Key Questions For Authors:**

1. Although the paper has mentioned this in the limitation section, I am still wondering how to estimate the very important hyperparameter $p$ and $\sigma$? I feel this could be critical for whether the propose method can have impact in real applications.
2. In the literature review part, the paper mentions that the more complicated global-local priors such as horseshoe achieves ABOS under the half-thresholding rule. Later, it shows that the simple normal prior also achieves ABOS under the relative belief ratio framework. Does that imply that a normal prior is as good as the horseshoe for the Bayesian multiple testing problems? Could you compare the performance of horseshoe vs normal priors, in finite samples maybe via simulation studies?

**Limitations:**

Yes

**Strengths And Weaknesses:**

### Strengths

1. This paper creatively combines the new framework for Bayesian multiple testing, relative belief ratio, with simple priors such as normal priors, to achieve optimal Bayes risk asymptotically.
2. This paper is well-structured and easy to follow. It provides sufficient background information for readers to understand the settings and the relevant literature.

### Weaknesses

1. While the mathematical derivations appear rigorous, I might still need to be convinced about the validity of the relative belief ratio approach, i.e., whether using 1 as a cutoff for the ratio of posterior vs prior densities to accept/reject the null hypothesis makes sense.
2. This paper contains very thorough literature review and introduction of the background knowledge, which takes more than 4 pages out of the 8 pages in the current version. Could it be a little bit more condensed?
3. This paper is a purely theoretical paper, focusing on asymptotic analysis. I guess it might be ok for this type of topic. But I would like to see some empirical analysis (simulations and/or real data), which could help with establishing the finite sample properties and provide practice comparisons and guidance.

---

> ### Author Rebuttal · Authors · 2026-03-31
>
> Thank you so much for your sincere and constructive feedback, encouragement, and guidance! Your words are truly encouraging and inspiring! We highly recommend you read all of our replies. You would not be disappointed. Please refer to our reply to Reviewers hDQE for how RB can be used in sample level (i.e., empirical Bayes comes to save).
>
> With Bayes oracle as reference (which uses true underlying parameters $p$ and $\\sigma$), we conduct an extensive simulation study to compare the finite sample performance of EBRB\_N, EBRB\_U and EBHS. We adopt the experimental framework in Datta & Ghosh (2013), and Ghosh et al. (2016) as basis, but greatly extend it to a more comprehensive studies. We can compare the simulation averages of the proportion of misclassified hypotheses (as estimates of the misclassification probabilities) of EBRB\_N, EBRB\_U and EBHS with that of the Bayes oracle in the context of the two-groups problem, in order to assess how closely these multiple testing procedures actually perform relative to the Bayes oracle. Note the misclassification probability (MP) is equivalent to the individual Bayes risk $(1 - p) t_{1i} + p t_{2i}$, see Bogdan et al. (2008). Here, the Bayes oracle establishes the lower bound to the MP, whereas the line MP = p represents the case in which all null hypotheses are rejected irrespective of the data. In both van der Pas et al. (2014) and Ghosh et al. (2016), they used $d_3 = 1$ and $d_4 = 2$ as default choices in their simulation study. We adopt their choices for the implementation of EBHS here. EBHS is implemented using the well-recognized horseshoe package (van der Pas et al., 2019). Analogously, we choose $d_1 = 1$ and $d_2 = 2$ as default choices for the implementation of EBRB\_N and EBRB\_U.
>
> The simulated data are generated as follows. We always fix $\\sigma_0^2 = 1$. For each chosen fixed $p \\in (0,1)$, we consider generating $n = \\{200,1000,10000\\}$ independent observations $X_1, ..., X_n$ from the two groups model $(1 - p) N(0, 1) + p N(0, 1 + \\sigma^2)$ where four distinct values of $\\sigma^2$, specifically 3, 5, 7, and 9 are considered. To estimate the misclassification probability, the process is repeated 1000 times, and simulation averages of misclassification proportions across these replications are used as estimates of the misclassification probabilities for each multiple testing procedure under consideration.
>
> Figures 1-3 in the link (https://anonymous.4open.science/r/Figure-1531/Figure.pdf) present the estimated misclassification probabilities corresponding to the EBRB\_N, EBRB\_U and the EBHS multiple testing procedures, alongside those of the Bayes oracle against values of $p$ in the set $\\{0.01,0.05,0.1,0.15,0.2,0.25,0.3,0.35,0.4,0.45,0.5\\}$. As shown in figures, it is clear that the both EBRB\_N and EBRB\_U uniformly outperforms the EBHS approach in every case (often by a large margin) in terms of achieving lower misclassification probabilities across all combinations of $n$'s, $\\sigma^2$'s, and $p$'s. Moreover, for any choice of $n$, as the $\\sigma^2$ increases from 3 to 9, the performances of the EBRB\_N and EBRB\_U progressively approaches that of the Bayes oracle. Notably, when the $\\sigma^2$ gets larger, the both EBRB\_N and EBRB\_U nearly matches the Bayes oracle's performance exactly in a wide range of $p$'s. The EBRB\_U always do slightly better than EBRB\_N as expected by the rate of convergence effects. The computations of EBRB\_N and EBRB\_U are highly efficient.
>
>
> **Reply to**: While the mathematical derivations appear rigorous, I might still need to be convinced about the validity of the relative belief ratio approach, i.e., whether using 1 as a cutoff for the ratio of posterior vs prior densities to accept/reject the null hypothesis makes sense.
>
> In the discrete case, this cut-off at 1 is indicating when there is evidence in favor (or against) by an increase (or decrease) in belief. For the continuous case, one looks at a ratio of the posterior to prior probabilities of neighbourhoods that shrink to the point of interest and then takes a limit. Under weak conditions (prior is positive and continuous at the limiting point), this gives the relative belief ratio in our paper. More elaborated discussion can be found in chapter 4 of Evans (2015).

---

> > ### Author Rebuttal · Reviewer_B6fF · 2026-04-02
> >
> > Thank you for addressing my concerns and adding this throughly designed simulation study. The simulation study significantly complements the paper so please consider including it in the paper.

---

> > > ### Author Response · Authors · 2026-04-03
> > >
> > > We sincerely thank you for the careful reading and constructive suggestions. We greatly appreciate your time and effort during such a busy period of time. We will include the simulation study in the paper.

---

### Official Review · Reviewer_hDQE · 2026-03-13

**Soundness:** 3
**Presentation:** 2
**Significance:** 2
**Originality:** 3
**Overall Recommendation:** 4
**Confidence:** 2

**Summary:**

The paper studies multiple testing rules and their asymptotic Bayesian optimality in a sparse signal regime. Formally, there are $n$ random variables $X_1, \dots , X_n$ drawn from $N(\mu_i, 1)$ respectively where each $\mu_i$ is $0$ with probability $1 - p$ and otherwise drawn from $N(0, \sigma^2)$ with probability $p$ for unknown parameters $p,\sigma$. Now we observe $X_1, \dots , X_n$ and our goal is to predict which of the means $\mu_i$ is nonzero.


When the parameters are known, the Bayes-optimal predictor, which minimizes the expected number of errors, can be obtained via direct computation. However, we want to understand what types of estimators are asymptotically optimal as $n \rightarrow \infty$ without requiring knowledge of $p, \sigma$.


To do this, the paper considers a sequence of parameters depending on $n$ satisfying the following limits $p_n \rightarrow 0, \sigma_n \rightarrow \infty, \frac{\log (1-p_n)^2 \sigma_n^2/p_n^2 }{\sigma_n^2} \rightarrow C$ for some positive constant $C$. This is the natural limit in the sparse regime (with $p_n \rightarrow 0$) that allows for nontrivial prediction and matches the regime studied by an earlier paper Bogdan et. all (2011).

While asymptotic Bayes optimality has been shown for some classes of testing procedures, the main point of this paper is to analyze a different paradigm/family of methods. Previously it was shown that if you just assume a prior on the means $\mu_i$ of some ``horseshoe" form, then as long as the parameters of the horseshoe scale with $p_n,\sigma_n$ in the right way, then Bayesian estimation with such a prior approaches the asymptotically optimal error rate. This paper considers a much simpler prior consisting of just a single Gaussian, but a different estimator based on the relative belief ratio (essentially whether the observation increases the probability density at $0$ or not). They show that if they are given a sequence $\tau_n$ such that $\frac{\tau_n}{p_n^{-1}\sqrt{\log(1/p_n)}}$ approaches a constant, then relative belief ratio with prior $N(0, \tau_n^2)$ approaches the Bayes-optimal rate. The paper also gives a natural generalization to the setting when each $X_i$ is multi-dimensional.

**Compliance With Llm Reviewing Policy:**

Affirmed.

**Key Questions For Authors:**

Admittedly, I am not too familiar with this area, but it seems the proposed methods (and also the previous work discussed involving the horseshoe prior) still need an "oracle" quantity in $\tau_n$. How does this get around the limitation of the Bayes optimal method needing to know $p_n, \sigma_n$ ? Is the main point that it gives a simpler understanding and way of analyzing other methods? Does it propose any new methods that are truly oracle-free?

**Limitations:**

Yes

**Strengths And Weaknesses:**

Strengths:

- The paper addresses a fundamental problem in multiple hypothesis testing.

- The paper presents a new perspective and a clean analysis of an estimator that is both novel and simpler compared to previous analyses.


- The theory and proofs seem sound.


Weaknesses:

- The paper only studies a specific asymptotic regime in the limit as the number of samples goes to infinity.

- I'm not sure if the paper has a practical/perscriptive takeaway. Does it suggest any new methods or conclusions for what statisticians should be doing in practice? Also see questions below.

- The paper is a bit notationally dense and uses quite a bit of statistics jargon. This made it somewhat difficult to understand for an outsider, although maybe this is standard in the literature.

---

> ### Author Rebuttal · Authors · 2026-03-31
>
> Thank you so much for your sincere and constructive feedback, encouragement, and guidance! Your words are truly encouraging and inspiring! We highly recommend you read all of our replies. You would not be disappointed. Please refer to our reply to Reviewers B6fF on an intensive simulation study.
>
> In the paper, we are discussing the ABOS property at the population level (using a tuning parameter, which depends on the underlying true parameter $p$). Inspired by your suggestions, the empirical Bayes construction for $p$ used in van der Pas et al., (2014) and Ghosh et al., (2016), we now really push ourselves to further develop our relative belief (RB) approach to the sample level, so that the RB approach can be used in practice, instead of more purely from a theoretical lens. Specifically, we develop a fully data-driven empirical Bayes relative belief (EBRB\_N) approach using the normal prior $N(0,\\tau^2)$, where we use data to give an estimator (i.e., $\\hat{\\tau} = \\min\\{ n\\sqrt{\\log n}, \\frac{d_1 n \\sqrt{\\log n}}{\\sum_{i=1}^n \\mathbf{1}(|X_i| > \\sqrt{d_2 \\log n})} \\},$ where $d_1 \\ge 1$ and $d_2 \\ge 2$ are some predetermined finite positive constants) for the hyperparameter $\\tau$ in the normal prior, which is asymptotically of the desired order of $p_n^{-1} \\sqrt{\\log(1/p_n)}$, and then carry out the RB rejection rule (i.e., $|X_i| > \\sqrt{(1 + \\frac{1}{\\hat{\\tau}^2}) \\log(1 + \\hat{\\tau}^2)}$). Along with Assumption 2.3 in our paper, we further assume $p \\propto n^{-\\beta}$ for some unknown $0 < \\beta < 1$ as Ghosh et al. (2016) did in their paper. Under those assumptions, we rigorously prove that the EBRB attains ABOS using a general random threshold extension (applying to thresholding rules in the form of  $X^2_i > \\hat{c}^2_{random}$) of a existing general fixed threshold result (i.e., Theorem 3.2 in Bogdan et al. (2011), which applies to thresholding rules in the form of $X_i^2 > c_{fixed}^2$). Again, our proofs are much simpler than such proofs in the previous literature. Under the same assumptions, Ghosh et al. (2016) proved the empirical Bayes procedure using a general
> class of one-group shrinkage priors attains nearly ABOS. Still, under same assumptions, Ghosh & Chakrabarti (2017) then proved empirical Bayes procedure using horseshoe-type priors attains exact ABOS. Specifically, they use $\\hat{\\xi} = \\max\\{ \\frac{1}{n}, \\frac{1}{d_3 n} \\sum_{i=1}^n \\mathbf{1}(|X_i| > \\sqrt{d_4 \\log n}) \\},$ where $d_3 \\ge 1$ and $d_4 \\ge 2$ are some predetermined finite positive constants, and they would reject $H_{0 i}$ if $1 - \\mathbb{E}(\\kappa_i \\mid X_i, \\hat{\\xi}) > \\frac{1}{2}, i = 1, ..., n.$ We would love to provide all the detailed proofs in the next round of discussion because of the word limit in this round. The horseshoe prior is certainly notable among those priors, we denote the empirical Bayes procedure using the horseshoe prior as EBHS.
>
> Beyond the normal prior we discussed in the paper, we keep thinking whether we can be braver, meaning whether we can simply use a bounded uniform prior $U(-\\tau_U, \\tau_U)$, (which carries a lot of historical remarks, e.g., objective Bayes literature) with the RB rule to achieve ABOS. Note $U(-\\tau_U, \\tau_U)$ are not either exponential-tailed or polynomial-tailed priors. Apparently, this type of uniform prior constitutes a new class in the ABOS literature, which is different from the unbounded light-tailed normal prior we proposed or any of the previous unbounded global-local heavy-tailed priors. For the uniform prior $U(-\\tau_U, \\tau_U),$ the RB rule is to reject $H_{0i}$ if $X^2_i > \\tilde{r}\_{U},$ where $\\tilde{r}\_U = y^2\_{\\tau_U}$, and $y\_{\\tau_U} > 0$ is the unique solution to $2 \\tau\_U \\varphi(y\_{\\tau\_U}) = \\Phi(y\_{\\tau\_U} + \\tau\_U) - \\Phi(y\_{\\tau\_U} - \\tau\_U)$. Now, here comes our “double surprises". Applying Theorem 3.2 in Bogdan et al. (2011), we find the RB rule with the uniform prior can achieve ABOS in the population level if the tuning parameter $\\tau_{U,n}$ is asymptotically of the order of $p_n^{-1} \\sqrt{\\log(1/p_n)}$. Thus, the uniform prior (which is in a completely new class) can indeed be used to achieve ABOS. And, notice that $p_n^{-1} \\sqrt{\\log(1/p_n)}$ is also the order required for the tuning $\\tau_n$ in the normal prior to attain ABOS. Since the tuning parameters in the uniform prior and the normal prior share the same desired order $p_n^{-1} \\sqrt{\\log(1/p_n)}$, the same $\\hat{\\tau}$ as in the normal case can also be used for empirical Bayes relative belief approach for the uniform prior (denoted as EBRB\_U). Note that, in terms of estimation, the rate of convergence properties of light-tailed distributions (e.g., normal) are usually better than those of heavy-tailed distributions (e.g., horseshoe). Compared with normal, the rate of convergence properties of bounded uniform are even better (in terms of having a sharper constant).

---

> > ### Author Rebuttal · Reviewer_hDQE · 2026-04-03
> >
> > Thank you for your detailed feedback. The simulation study will improve the paper so please make sure to add it to the paper.

---

> > > ### Author Response · Authors · 2026-04-04
> > >
> > > We sincerely thank you for the careful reading and constructive suggestions. We greatly appreciate your time and effort during such a busy period of time. We will include the simulation study in the paper.

---

### Official Review · Reviewer_ZYTq · 2026-03-16

**Soundness:** 3
**Presentation:** 4
**Significance:** 2
**Originality:** 3
**Overall Recommendation:** 4
**Confidence:** 3

**Summary:**

This paper considers multiple testing problem of (multivariate) normal means under sparsity under a Bayesian framework, with a 0-1 loss and an additive loss function. Bogdan et al. (2011)  and Qin & Ghosh, 2025 have established the Bayes Risk of the oracle Bayes estimator for univariate and multivariate normal means problems. The authors prove asymptotic Bayes optimal under sparsity (ABOS) under the normal means problem with a single parameter, light-tailed one-group normal prior. This is in contrast to complicated hierarchal one-group global–local heavy-tailed priors proposed in the existing literature. The paper also provides review of the existing optimal bayes estimators for both multivariate and univariate means problem.

**Compliance With Llm Reviewing Policy:**

Affirmed.

**Key Questions For Authors:**

1. How is the performance of this prior? In general clear delimitation of what “simplicity of the prior” means here (the ABOS property is sensitive to tuning and does not imply simpler priors are broadly preferable outside this testing regime)
2. Can you discuss how easy/difficult it is easy to extend you result (like those in https://arxiv.org/pdf/1510.01307) to horseshoe priors
3. Is there a empirical estimator that you can propose like in Qin and Ghosh?

**Limitations:**

yes

**Strengths And Weaknesses:**

The paper is written well and easy to follow. I like that the paper does a good job introducing the problem and existing solutions. The paper suggests a single parameter light tailed prior that can achive ABOS. The advantage of global-local prior is that is that they resemble the spike and slab priors very closely, but are much easier to implement. And spike and slab priors have very good finite sample properties as well. While this simpler prior has the right asymptotic properties, its usefulness is hard to judge without any study of the finite sample (or simulation) study. Also the global-local prior cover a wide range of distributions and tail that the proposed prior doesn't cover. Does the Relative Belief Inference lead to ABOS properties under those sophisticated priors? Is this worth studying, can you elaborate why or why not?

---

> ### Author Rebuttal · Authors · 2026-03-31
>
> Thank you so much for your sincere and constructive feedback, encouragement, and guidance! Your words are truly encouraging and inspiring! We highly recommend you read all of our replies. You would not be disappointed.
>
> Due to the word limit, please refer to our replies to Reviewer pyaN for additional references supplemented to the paper.
>
> **Q1.** How is the performance of this prior?
>
> **R1.** Please refer to our replies to Reviewers hDQE and B6fF for how RB can be used in sample level (i.e., empirical Bayes comes to save) and an intensive simulation study, respectively.
>
> **Q2.** Can you discuss how easy/difficult it is easy to extend your result to horseshoe priors？
>
> **R2.** Based on the fact that both uniform prior and normal prior can be used with RB rules to obtain ABOS (see our replies to Reviewer hDQE), it seems that suitable priors can have different tail behaviors, which is quite different from the requirement for the global-local approaches to ABOS (which has a mandatory requirement on heavy tails). To attain ABOS, the same order $p_n^{-1}\sqrt{\log(1/p_n)}$ is required for the tuning parameters for the uniform prior and the normal prior, which is quite intriguing. This seems to suggest that the main requirement for the priors used in the RB approach is that those priors need to be able to place their masses around the parameter values having relatively high likelihood, which can largely avoid the prior-data conflict. Here, the prior-data conflict (Evans \& Moshonov, 2006) refers to situations where the prior places most of its mass on parameter values that are incompatible with the observed data. The importance of prior-data conflict effect is also emphasized in Use of Bayesian Methodology in Clinical Trials of Drug and Biological Products. Guidance for Industry, U.S. Food and Drug Administration (2026). Gelman et al. (2017)  also point out priors need to be understood in the context of the likelihood. Note, the concept of prior-data conflict is deeply connected with the popular idea of weakly informative priors, advocated by Gelman (2006), who creatively first introduced the notion of weakly informative priors, but did not give a precise mathematical characterization of this notion. Evans \& Jang (2011) give a precise definition of weakly informative priors and a corresponding comprehensive methodology in terms of a measure of prior-data conflict. Based on Evans \& Jang (2011), essentially, a prior $\tilde{\pi}_2$ is considered to be weakly informative relative to another prior $\tilde{\pi}_1$ whenever $\tilde{\pi}_2$ leads to fewer prior-data conflicts a priori than $\tilde{\pi}_1$.
>
> Based on the above discussion, our conjecture is that RB can work with the heavy-tailed global-local priors to achieve ABOS. The proofs can be harder than the proofs we did for the uniform prior and the normal prior, but there are abundant and mature techniques for concentration inequalities.  Researchers just need to target their concentration inequalities to the relative belief ratio, and we are optimistic that many capable researchers working on global-local priors can achieve this.
>
> **Q3.** Empirical estimator?
>
> **R3.** Given the observations $X\_1,\ldots,X\_n$, for some predetermined finite positive constants $d\_5 \ge 1$ and $d\_6 \ge 2$, define $\\hat{\\tau}\_{multi} = \\min \\{ n^{2/k} \\log n, d\_5 ( \\frac{n}{\\sum\_{i=1}^n \\mathbf{1}(X^{\\top}\_i \\Sigma^{-1} X\_i > d\_6 \\log n)} )^{2/k} \\log n \\}.$ This estimator is asymptotically of the desired order $p\_n^{-2/k} \\log(1/p\_n)$. We conjecture that combining $\\hat{\\tau}\_{multi}$ with the RB rule in the multivariate normal means setting would lead to ABOS, which is currently work in progress.
>
> Overall, we view RB and global-local purely posterior approaches as complementary.  People are usually familiar with the purely posterior methods, so we hope to draw more attention to the RB thinking mode. While normal means settings (also asymptotically equivalent to nonparametric regression under some regularity conditions, revealed by Brown and Low, 1996) are already rich enough to study really fundamental problems, practitioners always want to go to more complex model settings (such as graphical models and deep neural networks, more details in Bhadra et al., 2019). The global-local priors literature has developed a wide range of computational tools for those more complicated model settings. Currently, RB literature hasn't touched very complicated model settings a lot, so RB can benefit from computation designs in the global-local priors literature, while the nice simplicity of RB may also help a lot in various settings (in terms of accuracy and change of designs), which would be very valuable to explore.

---

> > ### Author Rebuttal · Reviewer_ZYTq · 2026-04-03
> >
> > I will keep my rating. Thanks for your responses.

---

> > > ### Author Response · Authors · 2026-04-04
> > >
> > > We sincerely thank you for the careful reading and constructive suggestions. We greatly appreciate your time and effort during such a busy period of time. We will include the simulation study in the paper.

---

### Official Review · Reviewer_LwkN · 2026-03-16

**Soundness:** 3
**Presentation:** 3
**Significance:** 2
**Originality:** 3
**Overall Recommendation:** 4
**Confidence:** 2

**Summary:**

This work studies the choice of the one-group Gaussian prior for rejecting a global null hypothesis (e.g. means $\mu_i$ all zero is the null) under a two-groups model in the setting of Bayes risk for multiple testing. In contrast to prior work, which leverages a global-local shrinkage prior with multiple parameters, the authors show that a simple, one-group Gaussian prior is sufficient to achieve asymptotically Bayes optimal performance under sparsity (ABOS). The result hinges on the work's choice to use \textit{relative belief} - i.e. the ratio between the posterior and prior density, rather than just the posterior likelihood of the parameter. By leveraging a simple Gaussian prior with multiple testing procedures based on relative belief, the authors prove that their approach is ABOS for both univariate and multivariate means under their specified two-groups model.

**Compliance With Llm Reviewing Policy:**

Affirmed.

**Key Questions For Authors:**

1.  A natural question for this work is the fragility of the result when we deviate from the "two-groups model" - while normality of observations as a modeling assumption may be justified through CLT arguments, normality of the $\mu_i$ parameter seems somewhat unjustified in many settings. Compared to analysis for sparse global null testing in frequentist regimes, how dependent in the ABOS property under the two groups model? Additionally, is the two groups model the "canonical" modeling assumption for global null testing under sparse assumptions in the Bayesian literature?
2. The authors comment on potential applications to sequential inference are particularly interesting: note that the relative belief ratio essentially maps on to a likelihood ratio, which is the key building block for anytime valid inference approaches (frequentist time-uniform inference based on Ville's inequality). Beyond recovering ABOS properties, could one also make frequentist statements using relative belief?

**Limitations:**

Yes.

**Strengths And Weaknesses:**

The strongest points of this submission is (i) clarity of the exposition and (ii) the simplicity of the approach. The authors provide detailed exposition on Bayesian approaches for hypothesis testing (including posterior vs. relative belief based inference), the two-groups model (and its role for inducing sparsity), and comparisons with existing approaches. As someone familiar with statistical testing for the global null but much less familiar with Bayesian approaches for statistical testing, this exposition was immensely helpful for understanding the contributions of the work. The second strength of this work is the relative simplicity of its solution: by considering a testing procedure based on relative belief, rather than posterior likelihoods, the authors show that a one group Gaussian prior is fully sufficient to recover ABOS (asymptotic bayes optimality under sparsity). In contrast to existing work, which leverages at least two prior parameters, the authors recover a simple, computationally tractable solution by switching the design of their test.

This work has one major limitation (in the context of ICML). While the theoretical results of this work stand on their own, even a small set of experiments for global null testing under sparse alternatives (e.g. $F$-tests, higher criticism methods, max tests) + existing Bayesian approaches (e.g. horseshoe prior) could help readers understand the benefits of this method. Even completely synthetic datasets would be sufficient to show how the approaches suggested by the author improves over existing methods. Because many methods (both frequentist and Bayesian) are available to test a similar hypothesis, it would be useful for the authors to provide empirical comparisons (even in the appendix if there is no space in the main body).

---

> ### Author Rebuttal · Authors · 2026-03-31
>
> Thank you so much for your sincere and constructive feedback, encouragement, and guidance! Your words are truly encouraging and inspiring! We highly recommend you read all of our replies. You would not be disappointed.
>
> Please refer to our replies to Reviewers hDQE and B6fF for how RB can be used in sample level (i.e., empirical Bayes comes to save) and an intensive simulation study, respectively.
>
> Due to the word limit, please refer to our replies to Reviewer pyaN for additional references supplemented to the paper.
>
> **Q1.**  A natural question for this work is the fragility of the result when we deviate from the "two-groups model" - while normality of observations as a modeling assumption may be justified through CLT arguments, normality of the $\\mu_i$ parameter seems somewhat unjustified in many settings. Compared to analysis for sparse global null testing in frequentist regimes, how dependent is the ABOS property under the two groups model? Additionally, is the two groups model the "canonical" modeling assumption for global null testing under sparse assumptions in the Bayesian literature?
>
> **R1.**  The two-groups model presented in the paper is canonical at least in the line of research on shrinkage priors. The ABOS framework is already built for normal data (i.e., Bogdan et al., 2011; Qin \& Ghosh, 2025), exponential data (i.e., Li, 2013), and heavy-tailed data (i.e., Tang et al., 2017), although they are usually built parametrically. Thus, it indeed seems worthwhile and interesting to explore the notation of ABOS nonparametrically, in which Ghosal \& van der Vaart (2017) would be potentially really helpful in this respect. The dependence in the ABOS under the two-groups model is not clear at this point. A comprehensive sensitivity analysis on the dependence in the ABOS under the two-groups model would also be worth carefully formulating and conducting.  In practice, though, it seems that whether we use parametric modeling or nonparametric modeling really depends on the applications. This is still a tradeoff on the rate of convergence and uncertainty quantification. I hope this reply is helpful.
>
>
> **Q2.** The authors comment on potential applications to sequential inference are particularly interesting: note that the relative belief ratio essentially maps on to a likelihood ratio, which is the key building block for anytime valid inference approaches (frequentist time-uniform inference based on Ville's inequality). Beyond recovering ABOS properties, could one also make frequentist statements using relative belief?
>
> **R2.** We really enjoy your insights on i) the design switching, ii) the relative belief ratio essentially maps onto a likelihood ratio.  With relation to the likelihood ratio, it seems to us that many things one may try, which certainly include anytime valid inference approaches. It also seems to us that there may be a potential Bayesian perspective on high criticism, using the relative belief. To the best of our knowledge, the previous literature almost always looks at high criticism using a frequentist perspective.
>
> As we pointed out in our replies to reviewer hDQE, we can also use a relative belief rule with the uniform prior $U(-\\tau\_U, \\tau\_U)$.  The uniform prior $U(-\\tau\_U, \\tau\_U)$  is historically considered as an “uninformative" prior, which is intensively discussed in objective Bayes literature (see, e.g., Berger et al., 2024; Kass \& Wasserman, 1996). Objective Bayes is highly connected with frequentism. With such a bridge, there may be more fruitful relations between relative belief and frequentism.

---

> > ### Author Rebuttal · Reviewer_LwkN · 2026-04-03
> >
> > The authors have provided general insights into my questions, and I will maintain my positive score for this submission.

---

> > > ### Author Response · Authors · 2026-04-04
> > >
> > > We sincerely thank you for the careful reading and constructive suggestions. We greatly appreciate your time and effort during such a busy period of time. We will include the simulation study in the paper.

---

### Official Review · Reviewer_pyaN · 2026-03-24

**Soundness:** 3
**Presentation:** 2
**Significance:** 2
**Originality:** 3
**Overall Recommendation:** 4
**Confidence:** 2

**Summary:**

The authors studys asymptotic Bayes optimality under sparsity (ABOS) in high-dimensional multiple testing, and aim to prove that pairing a basic one-group light-tailed prior with relative belief inference is sufficient to achieve this theoretical optimality. Under the two-groups model and Bogdan et al. (2011) additive 0-1 loss framework (Assumption 2.3), they show that the relative-belief rejection rule can get the same asymptotic Bayes risk as the Bayes oracle, for both univariate and multivariate normal means problems. Their proofs are very direct and simple, they do not need the difficult posterior concentration things like other papers. This gives a new way to think about evidence using both prior and posterior together. So, they prove that oracle-level optimality is possible even with the simplest possible prior.

**Compliance With Llm Reviewing Policy:**

Affirmed.

**Final Justification:**

The authors addressed my questions at a high level, and I will keep my positive score for this submission.

**Key Questions For Authors:**

1. Practical tuning of the hyperparameter.
The main results need $\tau_n$ to have a special asymptotic order like $\sqrt{log(1/p)}/p$. But the best $\tau^*$ depends on unknown $p$ and $\sigma$, so it cannot be used in real data. Can the authors tell us what practical way they plan to choose $\tau$  or put a hyperprior on $\tau$?



2. Finite-sample behavior.
The paper is only asymptotic and the conclusion says finite-sample analysis is future work. Could the authors add even a small numerical study? For example, compare the proposed relative-belief rule with the Bayes oracle and horseshoe methods at n=1000 and n=10000. This would help readers see how useful the asymptotic theory is in realistic sample sizes.



3. Unknown variance and dependence.
The analysis assumes known variance ($\sigma_0^2=1$) and independent tests. The conclusion says extensions need nontrivial work. Could the authors clarify which parts of the proof might still work if we estimate the variance or have mild dependence between tests, and which parts would completely break?

**Limitations:**

Yes

**Strengths And Weaknesses:**

Strengths
1. Soundness: The paper is very theoretical, but the main claims are well supported by clear theorems. In the univariate case, the authors derive the relative-belief testing rule, calculate the type I and type II errors and Bayes risk, find the best tuning parameter, and prove exact ABOS under Assumption 2.3. They do the same thing for the multivariate case. The proofs look correct and the assumptions are strong but the same as in the existing ABOS literature. The authors are honest about what is new and what comes from old papers.
2. Presentation: The manuscript is reasonably structured and the writing is clear, but it assumes the reader already knows Bogdan et al. (2011) and all the horseshoe/global-local prior papers very well. I would suggest adding a simple roadmap in the introduction and explaining more clearly what is new and what is inherited from previous work. This would help readers who are not experts in this exact line of research.
3. Significance: The paper addresses an important and relevant problem in sparse multiple testing: can we reach the Bayes oracle risk with a much simpler prior? The significance is moderate and mainly theoretical. It advances understanding by showing a new evidential route, and it could influence future research on simpler Bayesian methods. But it is not yet something that changes practice in applied machine learning today.
4. Originality: The originality is real. The sparse testing problem and ABOS framework are not new, but the paper gives a genuinely new perspective: using relative belief inference with a simple one-group light-tailed prior to get exact ABOS in both univariate and multivariate settings. This creative combination of existing ideas (simple prior + relative belief) is well explained and nicely distinguished from the heavy-tailed prior literature.

Weaknesses
1. The paper depends heavily on the asymptotic framework of Bogdan et al. (2011), so the contribution is more a new route inside an old framework than a completely new way to do sparse testing.
2. The literature review focuses too much on horseshoe and global-local priors; it does not clearly compare the new approach with other Bayesian multiple-testing ideas outside that line.
3. There are no simulations or finite-sample results at all, even though the authors say the tuning depends on unknown parameters and leave practical implementation for future work. A small simulation would make the paper much stronger.
4. The claims are limited by strong assumptions (known variance, independent tests). The conclusion admits extensions to unknown variance or dependent data need more study.

---

> ### Author Rebuttal · Authors · 2026-03-31
>
> Thank you so much for your sincere and constructive feedback, encouragement, and guidance! Your words are truly encouraging and inspiring! We highly recommend you read all of our replies. You would not be disappointed.
>
>  **Q1.** Practical tuning of the hyperparameter. The main results need $\\tau_n$ to have a special asymptotic order like $\\sqrt{\\log(1/p)}/p$. But the best $\\tau^*$ depends on unknown $p$ and $\\sigma$, so it cannot be used in real data. Can the authors tell us what practical way they plan to choose $\\tau$ or put a hyper-prior on $\\tau$?
>
> **R1.**  Please refer to our replies to Reviewers hDQE for how RB can be used in sample level (i.e., empirical Bayes comes to save).
>
> **Q2.** Finite-sample behavior. The paper is only asymptotic and the conclusion says finite-sample analysis is future work. Could the authors add even a small numerical study? For example, compare the proposed relative-belief rule with the Bayes oracle and horseshoe methods at $n=1000$ and $n=10000$. This would help readers see how useful the asymptotic theory is in realistic sample sizes.
>
> **R2.** Please refer to our replies to Reviewer B6fF for the finite-sample simulation study and numerical comparisons.
>
> **Q3.** Unknown variance and dependence. The analysis assumes known variance ($\\sigma_0^2 = 1$) and independent tests. The conclusion says extensions need nontrivial work. Could the authors clarify which parts of the proof might still work if we estimate the variance or have mild dependence between tests, and which parts would completely break?
>
> **R3.** From Bogdan et al. (2011) to Qin and Ghosh (2025), they all assume either $\sigma_0^2$ is known or $\mathbf{\Sigma}$ is known. The unknown variance is well known to be a notoriously difficult problem. We find a recent publication dealing with a relevant problem (i.e., Kotekal, 2024). After a careful read, it seems there are no results we can directly apply to our case of interest. A more thorough study has to be conducted for this issue. We apologize that we can not give more insight into this.
>
> About the dependence extension, our intuition is that the dependency needs to have some structure, such as the well-known positive regression dependency on a subset (PRDS) assumption, which appeared in Benjamini and Yekutieli (2001). The PRDS property is a form of positive dependence and can be loosely interpreted as indicating that all pairwise correlations are non-negative. There is also literature on negative dependence (see, e.g., Chi et al., 2025). Again, we have to do a more comprehensive study to better understand the underlying mechanism.
>
> We are more than happy to know any thoughts you might have on those issues!
>
> **Additional references supplemented to the paper**
>
> [1] U.S. Food and Drug Administration. Draft guidance for industry: Use of Bayesian methodology in clinical trials of drug and biological products, 2026.
>
> [2] Z. Chi, A. Ramdas, and R. Wang. Multiple testing under negative dependence. *Bernoulli*, 2025.
>
> [3] J. O. Berger, J. M. Bernardo, and D. Sun. Objective Bayesian Inference. World Scientific, 2024.
>
> [4] S. Kotekal. Variance estimation in compound decision theory under boundedness. *Advances in Neural Information Processing Systems*, 2024.
>
> [5] S. van der Pas, J. Scott, A. Chakraborty, and A. Bhattacharya. Horseshoe: Implementation of the horseshoe prior, R package version 0.2.0, 2019.
>
> [6] A. Gelman, D. Simpson, and M. Betancourt. The prior can often only be understood in the context of the likelihood. *Entropy*, 2017.
>
> [7] X. Tang, K. Li, and M. Ghosh. Bayesian multiple testing under sparsity for polynomial-tailed distributions. *Statistica Sinica*, 2017.
>
> [8] S. Ghosal and A. W. van der Vaart. Fundamentals of nonparametric Bayesian inference. Cambridge University Press, 2017.
>
> [9] K. Li. Bayesian multiple testing under sparsity for exponential distributions. PhD dissertation, University of Florida, 2013.
>
> [10] M. Evans and G. H. Jang. Weak informativity and the information in one prior relative to another. *Statistical Science*, 2011.
>
> [11] A. Gelman. Prior distributions for variance parameters in hierarchical models (comment on article by Browne and Draper). *Bayesian Analysis*, 2006.
>
> [12] M. Evans and H. Moshonov. Checking for prior-data conflict. *Bayesian Analysis*, 2006.
>
> [13] Y. Benjamini and D. Yekutieli. The control of the false discovery rate in multiple testing under dependency. *Annals of Statistics*, 2001.
>
> [14] L. D. Brown and M. G. Low. Asymptotic equivalence of nonparametric regression and white noise. *Annals of Statistics*, 1996.
>
> [15] R. E. Kass and L. Wasserman. The selection of prior distributions by formal rules. *Journal of the American Statistical Association*, 1996.

---

> > ### Author Rebuttal · Reviewer_pyaN · 2026-04-03
> >
> > The authors addressed my questions at a high level, and I will keep my positive score for this submission.

---

> > > ### Author Response · Authors · 2026-04-04
> > >
> > > We sincerely thank you for the careful reading and constructive suggestions. We greatly appreciate your time and effort during such a busy period of time. We will include the simulation study in the paper.

---

### Decision · Program_Chairs · 2026-04-30

**Decision:**

Accept (regular)

**Comment:**

The paper looks at the problem of multiple testing using relative belief ratio under the Bayesian decision theoretical framework. The multiple testing framework in the paper is presented in comparison to standard sparsity inducing priors. Unlike the typical sparsity inducing priors (like a heavy-tailed, horseshoe, or spike-and-slab prior), the authors show that using a standard Gaussian prior under relative belief yields optimal Bayes risk asymptotically under sparsity (referred to as the ABOS framework in the paper). This discovery is  somewhat surprising given that the more sophisticated priors are the standard choice for sparse Bayesian multiple testing. The perspective of hypothesis testing using relative belief is different from standard Bayesian hypothesis testing methods and a fairly novel contribution in this space. Generally, the writing is quite clear and easy for the reader to follow. Moreover, the authors have satisfactorily replied to the reviewers and therefore I vote to accept the paper.